# M+Adam: Low-Precision Training via Additive–Multiplicative Optimization

Xiaoyuan Liang [1]  Sebastian Loeschcke [2]  Mads Toftrup [3]  Anima Anandkumar [1]

## Abstract

Training with quantized weights can reduce costs but often results in degraded accuracy, especially when optimization is carried out in low precision, without storing high-precision copies. We identify a key failure mode: under low precision, standard optimizers can get stuck and not make progress, especially at large weight magnitudes due to coarse mantissa resolution. To overcome this, multiplicative updates have been previously proposed, in place of additive updates in standard optimizers. While successful under extremely low precision, such as under the logarithmic number system, they suffer from failures near zero and across sign changes. The failure modes of additive and multiplicative updates are therefore complementary. To exploit this, we propose M+Adam, which combines both update types: additive steps handle sign changes and small magnitudes, while multiplicative steps ensure progress at large magnitudes when additive updates are zeroed out under rounding. We prove monotone descent for M+Adam under standard smoothness assumptions. Across LLaMA-style pretraining with 60M–1B models, 1–8× Chinchilla budgets, and using only BF16, FP8, and FP4 master weights, M+Adam consistently improves low-precision training.

## 1. Introduction

Training large language models (LLMs) in full precision (FP32) incurs substantial memory, compute, and energy costs. Modern accelerators are optimized for reduced-precision formats such as FP8 and FP4, via specialized hardware like tensor cores (Micikevicius et al., 2022). As a result, most large-scale training pipelines adopt mixed-precision training, executing forward and backward passes in reduced precision (Micikevicius et al., 2018). However, they still

[1]California Institute of Technology [2]University of Copenhagen [3]Aarhus University. Correspondence to: Xiaoyuan Liang <xliang2@caltech.edu>.

*Proceedings of the 43rd International Conference on Machine Learning*, Seoul, South Korea. PMLR 306, 2026. Copyright 2026 by the author(s).

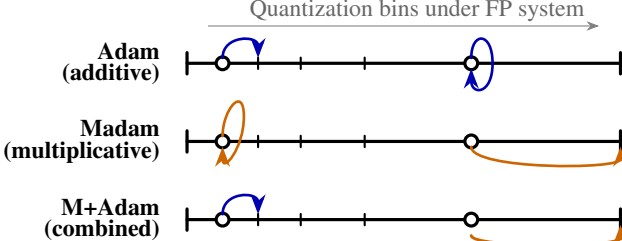

*Figure 1.* **Additive (Adam), multiplicative (Madam), and combined M+Adam updates under low precision.** FP grids under low precision cause additive updates to be lost to rounding at large magnitudes and multiplicative updates to stall near zero, while M+Adam avoids both failure modes.

|  | Adam | Madam | M+ADAM |
|---|---|---|---|
| Additive updates | ✓ | × | ✓ |
| Multiplicative updates | × | ✓ | ✓ |
| Can escape zero | ✓ | × | ✓ |
| Non-zero update at large $|w|$ | × | ✓ | ✓ |

*Table 1.* Comparison of update mechanisms and their failure modes across the studied optimizers.

retain high-precision copies (known as master weights) for optimization, since training directly in low precision results in significant degradation of accuracy and is often unstable (Zamirai et al., 2021).

Recently, the precision of master weights has been reduced, e.g. BF16 instead of FP32, using techniques such as stochastic rounding (SR) (Connolly et al., 2021) to keep them unbiased. While this reduces the memory overhead compared to FP32 master weights, they still exhibit degraded convergence or instability relative to full-precision FP32 training. This is because standard optimizers, such as Adam, fail to make progress at large weight magnitudes under low precision (larger quantization bins in Fig.2) , since updates smaller than the local quantization step are rounded to zero.

To overcome this failure mode, alternative optimizers have been proposed based on multiplicative updates, such as Madam (Bernstein et al., 2020). These are effective at taking larger steps at higher magnitudes (larger quantization bins in Fig.2), where standard optimizers with additive updates fail to make progress. Madam has been shown to be effective for training under extremely low precision, without high-

**Algorithm 1** M+Adam: additive–multiplicative update

---

**Inputs:** learning rates $\eta_a, \eta_m$; moments $\beta_1, \beta_2$; stabilizer $\epsilon$; threshold $\tau$; timestep $t$

**State:** additive moments $(u, v) \leftarrow \mathbf{0}$, multiplicative second moment $v^{(e)} \leftarrow \mathbf{0}$

All operations are performed element-wise.

**repeat**

$g \leftarrow \nabla_w \mathcal{L}(w_t)$

**Additive branch:** ▷*Adam-style methods*

$u \leftarrow \beta_1 u + (1 - \beta_1)g$

$v \leftarrow \beta_2 v + (1 - \beta_2)g^2$

$\hat{u} \leftarrow u/(1 - \beta_1^t), \quad \hat{v} \leftarrow v/(1 - \beta_2^t)$

$u^{(a)} \leftarrow -\eta_a \dfrac{\hat{u}}{\sqrt{\hat{v}} + \epsilon}$

**Multiplicative branch:**

$g^{(e)} \leftarrow (\ln 2)\, w_t g$ ▷*Gradient wrt exponent*

$v^{(e)} \leftarrow \beta_2 v^{(e)} + (1 - \beta_2)\big(g^{(e)}\big)^2$

$\hat{v}^{(e)} \leftarrow v^{(e)}/(1 - \beta_2^t)$

$\tilde{u}^{(m)} \leftarrow -\eta_m \dfrac{g^{(e)}}{\sqrt{\hat{v}^{(e)}} + \epsilon}$

$\rho \leftarrow \text{sign}(w_t) \max(|w_t|, \tau)$

$u^{(m)} \leftarrow \tilde{u}^{(m)}/\rho$

**Combine branches:**

$w_{t+1} \leftarrow w_t + w_t u^{(m)} + u^{(a)}$

**until** converged

---

precision master weights. For instance, Madam can train under the logarithmic number system (LNS) (Zhao et al., 2022), an extreme case of floating-point system where all the bits are allocated to the exponent, and mantissa is not present. While LNS has optimal hardware utilization in theory, it is not available in standard AI hardware.

Multiplicative methods such as Madam overcome the failure mode of Adam at large magnitudes (larger quantization bins in Fig.1), but they have their own failure modes. They struggle near zero, where relative updates produce small absolute changes and cannot move weights through zero when there is a sign change (Fig. 1).

**Our approach:** The additive and multiplicative updates thus suffer from complementary modes of failure. Additive updates get rounded to zero at large magnitudes when mantissa resolution is coarse. Multiplicative updates are able to overcome this and make progress at large magnitudes, but they cannot flip signs or escape exact zero initialization.(Fig. 1 and Table 1).

This motivates us to propose M+Adam. It is a new optimizer that combines Adam-style additive updates with Madam-style multiplicative updates for low-precision training, as outlined in Algorithm 1 and Fig. 2. We prove that M+Adam is a monotone descent method under standard smoothness assumptions, making it a valid optimizer.

We demonstrate stable training with BF16, FP8, and NVFP4 master weights across 60M–1B LLaMA-style models and 1–8× Chinchilla training budgets. M+Adam trains stably without needing any stochastic rounding (SR), and improves over AdamW and AdamW+SR, as seen in Table 2.

Together, these results support our central claim: combined additive–multiplicative updates are especially useful when low-precision master weights becomes a dominant source of optimization error.

## 2. Related Work

We review prior work on multiplicative optimization and low-precision training most closely related to our approach.

### 2.1. Multiplicative Updates

Multiplicative update rules apply relative, scale-invariant parameter changes and have a long history in optimization (Kivinen & Warmuth, 1997). In deep learning, Madam (Bernstein et al., 2020) adapts this principle as a multiplicative analogue of Adam, showing improved robustness and reduced sensitivity to learning-rate scaling. Recent work revisits multiplicative or hybrid updates in modern settings: Kirtas et al. (2024) study additive–multiplicative hybrids; Zhao et al. (2022) combine multiplicative updates with a logarithmic number system to enable low-precision training without FP32 master weights; and Nishida et al. (2025) analyze log-normal multiplicative dynamics under low-precision forward passes. In contrast to log-based Madam approaches, M+Adam remains within the standard floating-point arithmetic.

### 2.2. Low-precision Training

Mixed-precision frameworks such as AMP (PyTorch, 2025) and Transformer Engine (NVIDIA, 2023) accelerate BF16/FP8 arithmetic, but are largely orthogonal to optimizer design and typically rely on FP32 master weights for stable accumulation (Micikevicius et al., 2018). Standard additive optimizers such as AdamW can degrade when master weights are stored in reduced precision, since nearest rounding can cancel small model-weight updates (Zamirai et al., 2021). Stochastic rounding preserves small updates in expectation and can mitigate deterministic rounding bias (Ozkara et al., 2025). Kahan-style compensated updates provide another remedy by accumulating lost low-order update residuals in an auxiliary buffer, and have been shown to recover much of the accuracy lost in pure BF16 training (Zamirai et al., 2021). While both stochastic rounding and Kahan compensation can improve accumulation accuracy, they do not change the underlying additive update geometry or address dynamic-range limitations directly.

Several complementary approaches reduce low-precision

**1) Compute Gradients**  **2) Optimizer step**  **3) Combine & update weights**

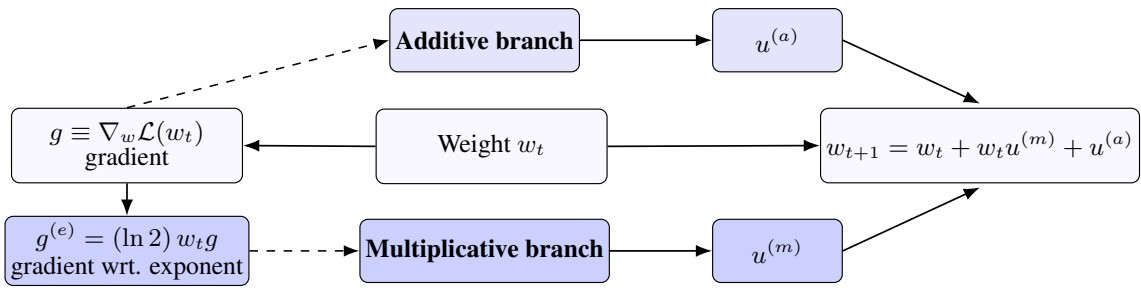

*Figure 2.* Overview of M+Adam. **(1) Compute the two gradients:** the standard weight gradient $g = \nabla_w \mathcal{L}(w_t)$, and gradient with respect to exponent, which evaluates as $g^{(e)} = (\ln 2)\, w_t g$. **(2) Compute the two updates:** use gradient $g$ to compute standard additive update $u^{(a)}$ and $g^{(e)}$ for relative multiplicative update $u^{(m)}$. **(3) Combine the two updates:** apply the combined update $w_{t+1} = w_t + w_t u^{(m)} + u^{(a)}$ to obtain the next weights.

storage costs without changing the optimizer geometry. LoQT (Loeschcke et al., 2024) trains most weights in 4-bit precision while maintaining a low-rank 16-bit subspace. A separate line of work compresses optimizer states through 8-bit or 4-bit moment quantization (Dettmers et al., 2022; Li et al., 2023). Fishman et al. (2025) study stable FP8 LLM pretraining at trillion-token scale with FP32 weights and FP8 forward computation, including quantized optimizer moments, focusing on system-level scaling and mixed-precision pipelines. In contrast, M+Adam modifies the update rule itself through a joint additive–multiplicative formulation, targeting the stability of low-precision weight storage rather than optimizer-state compression.

Recent work (Castro et al., 2025) explores training with 4-bit weights and activations using micro-scaling formats such as MXFP4, where each block shares a common scale factor. These approaches mainly stabilize low-bit matrix multiplication by managing outliers and representational fidelity during computation. Because low-bit quantization is highly discrete, such systems often keep higher-precision weights and round to the target format before GEMMs. Our work addresses a complementary challenge: stable gradient-based parameter updates when the master weights themselves are stored in BF16, FP8, or NVFP4. Thus, outlier-aware quantization and M+Adam attack different bottlenecks: the former improves low-bit computation, while the latter improves numerical stability across optimization steps.

## 3. Background

We briefly review Adam and Madam, the two optimizers that M+Adam builds upon.

**Adam.** Adam (Kingma & Ba, 2015) maintains exponential moving averages of the first and second moments of the gradient and applies adaptive, element-wise additive updates. AdamW (Loshchilov & Hutter, 2019) decouples weight decay from this adaptive gradient step:

$$\text{Adam}(w, g) \;=\; (1 - \eta\lambda)w - \eta\frac{\mu}{\sqrt{\nu} + \epsilon}. \qquad (1)$$

While effective in full precision, these additive updates can be rounded away in low-precision formats when their magnitudes fall below the local quantization step, which grows with $|w|$ in floating-point formats such as BF16 or FP8.

**Madam.** Madam (Bernstein et al., 2020) retains Adam's moment estimates but applies updates multiplicatively:

$$\text{Madam}(w, g) \;=\; w\exp\left(-\eta\frac{\mu}{\sqrt{\nu} + \epsilon}\right). \qquad (2)$$

This yields scale-invariant behavior, with step sizes proportional to $|w|$, but purely multiplicative updates preserve sign and cannot escape $w = 0$. We illustrate these complementary failure modes of AdamW and Madam in a controlled toy example (Sec. 5.1), which motivates combining local additive updates with multiplicative scale control.

## 4. M+Adam: Additive–Multiplicative Optimization

M+Adam is designed for low-precision weight storage, where the optimizer must update weights on a nonuniform floating-point grid. The method combines two complementary update mechanisms: an Adam-style additive branch that acts directly on the weights, and a Madam-style multiplicative branch that applies relative scale changes. This section derives the two branches, explains how they are combined, and presents the corresponding descent result.

### 4.1. Additive–Multiplicative Updates

Low-precision floating-point weights have nonuniform resolution: the spacing between representable values grows

with the exponent. As a result, purely additive updates can become rounded back to zero at large magnitudes. When $|w_t|$ is large, an additive update can be smaller than the local quantization bin and disappear after rounding; when $|w_t|$ is small, however, additive updates remain important because they can update values near zero and change signs. M+Adam is designed around this observation: it combines an Adam-style additive correction with a Madam-style multiplicative scale update.

Let $g_t = \nabla_w \mathcal{L}(w_t)$ be the gradient of weight $w_t$. The additive branch is the standard Adam-style update $u_t^{(a)}$ in the weight space,

$$u_t^{(a)} = -\eta_a \frac{\hat{u}}{\sqrt{\hat{v}} + \epsilon}.$$

A multiplicative update is as follows element-wise:

$$w_{t+1} = w_t \exp(a_t) = w_t 2^{a_t / \ln 2},$$

where $a_t$ is an adaptive preconditioned descent direction. Naturally, the update $a_t$ is based on the gradient $g_t^{(e)}$ with respect to the exponent. To obtain this, we use the mantissa $m_t$ and exponent $e_t$ in floating-point decomposition in the base-2 system:

$$w_t = m_t 2^{e_t},$$

The corresponding gradient simplifies as

$$g_t^{(e)} = \frac{\partial \mathcal{L}}{\partial e} = (\ln 2) \, w_t g_t.$$

Thus the gradient $g_t^{(e)}$ can be expressed without the explicit exponent value, and we use this in our updates.

We then apply adaptive normalization to $g_t^{(e)}$. In our implementation, the multiplicative branch uses a second-moment accumulator,

$$v_t^{(e)} = \beta_2 v_{t-1}^{(e)} + (1 - \beta_2)\big(g_t^{(e)}\big)^2, \tag{3}$$

$$\widetilde{u}_t^{(m)} = -\eta_m \frac{g_t^{(e)}}{\sqrt{\hat{v}_t^{(e)}} + \epsilon}. \tag{4}$$

Thus, $\widetilde{u}_t^{(m)}$ defines the branch's proposed scale movement:

$$\rho_t = \text{sign}(w_t) \max(|w_t|, \tau_t), \qquad u_t^{(m)} = \text{clip}\Big(\widetilde{u}_t^{(m)} / \rho_t\Big)$$

where sign is understood with the convention $\text{sign}(0) = +1$ in the construction of $\rho_t$. The threshold $\tau_t > 0$ prevents division by very small weights, and clipping keeps the relative factor well-defined.

$$w_t \mapsto w_t + w_t u_t^{(m)}.$$

Finally, M+Adam combines the multiplicative rescaling with the additive displacement:

$$w_{t+1} = w_t + w_t u_t^{(m)} + u_t^{(a)}.$$

This structure directly targets the two failure modes shown in Fig. 1. Additive updates can change signs and update small-magnitude weights, but they may be lost to rounding at large magnitudes. Multiplicative updates preserve relative progress at large magnitudes, but by themselves are sign-preserving and cannot recover exact zeros. By combining both branches, M+Adam maintains scale-aware progress where additive updates are rounded away while retaining the local flexibility needed near zero. Algorithm 1 summarizes the core method. Optionally, clipping is added for more stability, which is not listed in Algorithm 1, as outlined in App. A.

### 4.2. Theoretical Analysis

We analyze the deterministic exact-arithmetic core of Algorithm 1. Let $\mathcal{L}(w)$ be the weight-space objective, let $g = \nabla_w \mathcal{L}(w_t)$, and define the base-2 gradient with respect to exponent signal

$$g^{(e)} = (\ln 2) w_t g,$$

obtained by differentiating $z \mapsto \mathcal{L}(w_t 2^z)$ at $z = 0$. Algorithm 1 forms a relative multiplicative update $u^{(m)}$, sets $s = 1 + u^{(m)}$, and updates $w_{t+1} = w_t s + u^{(a)}$. For $s > 0$, writing $z = \log_2 s$ gives the equivalent scale-then-add form $w_{t+1} = w_t 2^z + u^{(a)}$. We assume the following local smoothness bound in these coordinates: for all $(d, z)$ in a neighborhood of the update,

$$\mathcal{L}(w_t 2^z + d) \leq \mathcal{L}(w_t) + \langle g, d \rangle + \langle g^{(e)}, z \rangle$$
$$+ \frac{L_a}{2} \|d\|^2 + \frac{L_e}{2} \|z\|^2 + L_{ae} \|d\| \, \|z\|. \tag{5}$$

Here $L_a$ and $L_e$ control curvature within the additive and exponent directions, while $L_{ae}$ controls their interaction.

**Theorem 4.1** (Idealized scale-then-add descent). *Assume Eq. (5) holds at $w_t$. Consider $w_{t+1} = w_t s + u^{(a)}$, with $s > 0$ element-wise and $z = \log_2 s$. Suppose the branch outputs satisfy, for some $\eta_a, \eta_m > 0$,*

$$\langle g, u^{(a)} \rangle \leq -\eta_a \|g\|^2, \qquad \|u^{(a)}\| \leq \eta_a \|g\|,$$
$$\langle g^{(e)}, z \rangle \leq -\eta_m \|g^{(e)}\|^2, \qquad \|z\| \leq \eta_m \|g^{(e)}\|.$$

*If*

$$L_a \eta_a + L_{ae} \eta_m < 2, \qquad L_e \eta_m + L_{ae} \eta_a < 2,$$

*then*

$$\mathcal{L}(w_{t+1}) \leq \mathcal{L}(w_t)$$
$$- \frac{\eta_a}{2} \left(2 - L_a\eta_a - L_{ae}\eta_m\right) \|g\|^2 \quad (6)$$
$$- \frac{\eta_m}{2} \left(2 - L_e\eta_m - L_{ae}\eta_a\right) \|g^{(e)}\|^2.$$

*In particular, $\mathcal{L}(w_{t+1}) < \mathcal{L}(w_t)$ unless $g = 0$.*

The theorem formalizes the local two-branch geometry of Algorithm 1: an additive weight-space displacement and a multiplicative scale change can be composed while preserving descent under the stated alignment and smoothness conditions. The proof is given in Appendix D.

# 5. Empirical Findings

We evaluate M+Adam in two stages. First, a controlled BF16-stored matrix-fitting diagnostic isolates the update-level failure modes predicted by our analysis. Second, LLaMA-style pretraining on C4/EN tests whether the same additive–multiplicative geometry improves low-precision training across model scale, token budget, and precision regime. Our main LLM study covers 60M, 130M, and 350M models over 1, 2, 4, and 8× Chinchilla budgets, and we additionally evaluate 1B models at the 1× budget across all precision regimes. We consider BF16, FP8, and NVFP4 master-weight storage with BF16 or FP8 compute.

## 5.1. Toy Diagnostic: Isolating Low-Precision Failure Modes

We begin with a minimal convex diagnostic designed to highlight the complementary strengths of additive and multiplicative optimizers. The task is matrix fitting,

$$\mathcal{L}(W) = c\|W - W^\star\|_F^4,$$

where each step samples a minibatch of entries uniformly with replacement and $c$ is a constant. Each optimizer step is computed in FP32, but the updated parameter is immediately rounded through BF16 before the next iteration. Since the objective is otherwise trivial, any failure reflects the interaction between the update geometry and BF16 storage. This directly tests the complementary regimes isolated by the theory. For the evaluation, we perform a gradient step using either Additive or Multiplicative, corresponding to the additive and multiplicative branches of M+Adam, respectively. We disable momentum and adaptive moments and tune the learning rate for simplicity. For M+Adam we have $\eta_a = \eta_m$, and in the figures, the Additive and Multiplicative baselines have their optimal learning rate for each setting, whereas M+Adam reuses the same learning rate across all settings.

Fig. 3 summarizes four diagnostic regimes. In the sign-flip regime, additive steps are needed to change signs: Additive and M+Adam correct the sign, while Multiplicative

remains sign-preserving. In the zero-revival regime, Multiplicative cannot escape from zero because its update is zero-absorbing, whereas Additive and M+Adam recover zero weights through the additive branch. In the small-weight regime, M+Adam retains the additive updates needed to reach the BF16-limited floor. In the large-weight regime, Additive stalls because its updates fall below the local BF16 spacing, while Multiplicative and M+Adam continue to make relative, scale-aware progress. Overall, the toy example isolates the intended complementarity: the additive branch enables sign changes, recovery from zero, and small-magnitude updates, while the multiplicative branch preserves progress when the quantization grid becomes too coarse. Full learning-rate sweeps and exact losses are provided in App. B.

## 5.2. Language-Model Pretraining Setup

We evaluate M+Adam on LLaMA-style autoregressive language-model pretraining on C4/EN. For a model with $P$ million trainable parameters, the $1\times$ Chinchilla budget is $T_{1\times} \approx 20P \times 10^6$ tokens, and the main scaling study evaluates $n\times$ budgets for $n \in \{1, 2, 4, 8\}$. This design is important because optimizer rankings can depend on the data-to-model ratio, training horizon, and tuning protocol (Wen et al., 2026; Semenov et al., 2025). The 60M–350M models are evaluated across the full 1–8× grid. The 1B model is evaluated at $1\times$ Chinchilla across all precision regimes to test whether the observed gains persist beyond the largest model used in our full scaling sweep.

We study four increasingly aggressive low-precision regimes: BF16/BF16, BF16/FP8, FP8/FP8, NVFP4/FP8, where the notation denotes weight storage / compute precision. BF16/BF16 corresponds to end-to-end BF16 training without FP32 master weights. BF16/FP8 keeps BF16 weights but uses Transformer Engine FP8 GEMMs. FP8/FP8 further stores selected trainable weight matrices persistently in FP8 and uses FP8 compute. NVFP4/FP8 uses true NVFP4 master-weight storage with FP8 GEMMs.

For the FP8 master-weight setting, we apply FP8 storage to attention and MLP linear layers, while embeddings, layer norms, and the language-model head remain in BF16. The selected matrices are quantized in blocks with FP16 scales. During training, the optimizer operates under the constraint that these weights are repeatedly quantized to FP8 after updates. In our FP8 experiments, this storage is simulated through explicit quantize/dequantize operations for compatibility with current Transformer Engine kernels.

We compare against AdamW in each regime. In BF16/BF16, we also include two stronger additive baselines: AdamW with stochastic rounding and AdamW with Kahan-style compensated accumulation. Stochastic rounding probabilistically rounds low-precision updates to nearby representable values so that small updates are preserved in expectation (Con-

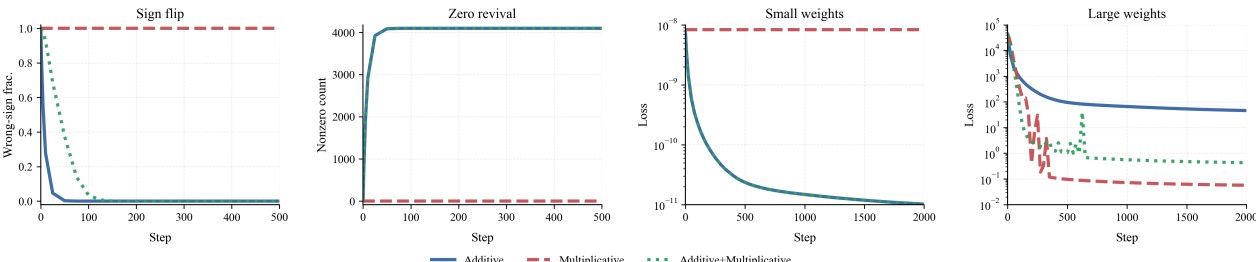

*Figure 3.* **BF16-stored matrix-fitting diagnostics.** Additive and Multiplicative use the optimal learning rate per task and M+Adam uses a shared learning rate across all tasks. (a) Additive and M+Adam can change signs to correct sign errors, while Multiplicative is sign-preserving. (b) Multiplicative is zero-absorbing, while Additive and M+Adam recover zero weights through the additive branch. (c) For small weights, Additive and M+Adam reach the BF16-limited floor, while Multiplicative plateaus higher. (d) For large weights, BF16 spacing grows with $|w|$, so Additive steps can be lost to rounding while Multiplicative updates remain active.

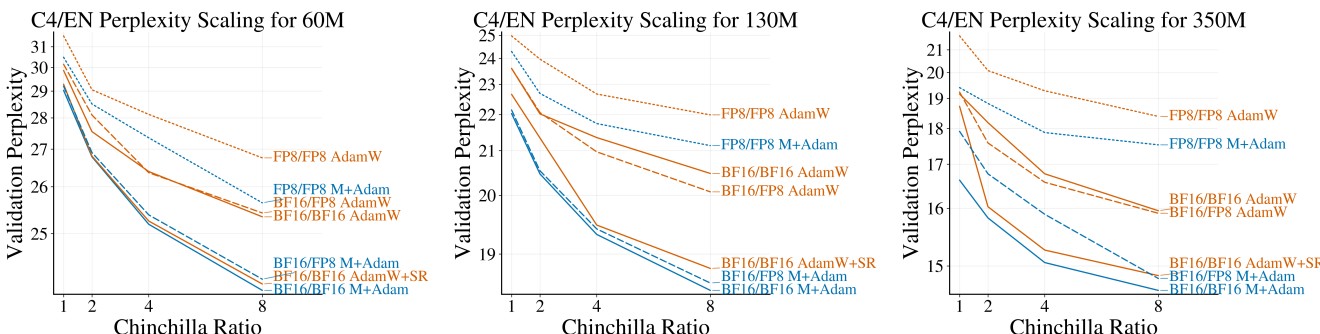

*Figure 4.* **C4/EN perplexity scaling across model sizes and precision regimes.** Validation perplexity as a function of Chinchilla ratio for 60M, 130M, and 350M LLaMA-style models. Each panel overlays tuned AdamW and M+Adam runs across BF16-weight/BF16-compute, BF16-weight/FP8-compute, and FP8-weight/FP8-compute regimes. AdamW with stochastic rounding is included in the BF16/BF16 setting. Lower perplexity is better.

nolly et al., 2021). Kahan-style compensated accumulation keeps an auxiliary correction buffer that accumulates low-order update residuals lost to rounding, thereby improving additive update accumulation under BF16 storage (Zamirai et al., 2021). We additionally report an FP32-weight/TF32-compute AdamW reference as a higher-precision point of comparison.

**Hyperparameter tuning.** We use a coordinate-descent protocol (Wen et al., 2026; Semenov et al., 2025) for each optimizer and precision regime. At $1\times$ Chinchilla, we sweep learning rate, warmup length, $\beta_1$, $\beta_2$, $\epsilon$, gradient clipping, and decoupled weight decay where applicable. For M+Adam, we additionally sweep the additive and multiplicative learning rates. To control cost at larger budgets and model sizes, we identify the hyperparameters that are scaling-sensitive and re-sweep only those in the remaining regimes, while holding the others fixed. We accept a hyperparameter change only when it improves final validation loss by at least $\Delta_{\text{loss}} \geq 3 \times 10^{-3}$. Full grids, sweep traces, and best configurations are reported in App. I.

### 5.3. Scaling Across Budgets and Precision Regimes

Fig. 4 reports C4/EN validation perplexity for 60M, 130M, and 350M models across $1$–$8\times$ Chinchilla budgets. Each panel compares tuned AdamW and M+Adam across the evaluated low-precision regimes.

Across model sizes, token budgets, and precision regimes, M+Adam consistently improves over the corresponding AdamW baseline. The gap persists as the Chinchilla ratio increases, showing that the improvement is not merely an early-training effect. The same trend also holds as the precision regime becomes more aggressive. FP8 compute degrades absolute perplexity for both optimizers, and FP8 weights make the problem harder still, but M+Adam retains a clear advantage.

### 5.4. Main $1\times$ Chinchilla Results

Table 2 summarizes the main $1\times$ Chinchilla results across 60M, 130M, 350M, and 1B models. M+Adam improves over AdamW in every low-precision regime and at every model size. In BF16/BF16, M+Adam reduces perplexity relative to AdamW by $2.8\%$, $6.6\%$, $13.3\%$, and $9.6\%$ for 60M, 130M, 350M, and 1B models, respectively. It also improves over

*Table 2.* Comparison of AdamW and M+Adam pre-training for LLaMA2-style language models on C4 at the $1\times$ Chinchilla budget. We report validation perplexity for each model size, optimizer, and precision regime. The NVFP4/FP8 setting uses true NVFP4 master-weight storage with FP8 GEMMs and is included as an aggressive low-precision stress test. Each entry is the best value achieved over the corresponding hyperparameter sweep unless otherwise noted.

| Weights / compute | Optimizer | 60M | 130M | 350M | 1B |
|---|---|---|---|---|---|
| *FP32 weights with TF32 compute* | | | | | |
| FP32 / TF32 | AdamW | 29.047 | 22.615 | 15.955 | 14.011 |
| *BF16 weights with BF16 compute* | | | | | |
| BF16 / BF16 | AdamW | 29.865 | 23.587 | 19.156 | 15.642 |
| BF16 / BF16 | AdamW + Kahan | 29.275 | 22.859 | 18.758 | 14.854 |
| BF16 / BF16 | AdamW + SR | 29.277 | 22.646 | 18.720 | 14.346 |
| BF16 / BF16 | M+Adam | **29.035** | **22.032** | **16.607** | **14.139** |
| *BF16 weights with FP8 compute* | | | | | |
| BF16 / FP8 | AdamW | 30.151 | 23.594 | 19.234 | 16.346 |
| BF16 / FP8 | M+Adam | **29.201** | **22.136** | **17.905** | **15.690** |
| *FP8 weights with FP8 compute* | | | | | |
| FP8 / FP8 | AdamW | 31.560 | 24.966 | 21.674 | 17.235 |
| FP8 / FP8 | M+Adam | **30.484** | **24.284** | **19.395** | **16.712** |
| *NVFP4 weights with FP8 compute* | | | | | |
| NVFP4 / FP8 | AdamW | 33.884 | 27.517 | 24.941 | 19.066 |
| NVFP4 / FP8 | M+Adam | **30.948** | **25.511** | **20.796** | **17.692** |

*Table 3.* **Scaling-sensitive hyperparameters under low-precision regimes.** We list hyperparameters whose approximate-optimal values shift with model scale and/or token budget under the restricted re-tuning protocol. Hyperparameters not listed were found to transfer across the evaluated regimes. We report results for the two precision settings used throughout the paper: BF16 weights with BF16 compute, and BF16 weights with FP8 compute.

| Setting | Optimizer | Scaling-sensitive hyperparameters |
|---|---|---|
| BF16 weights + BF16 compute | AdamW | learning rate, warmup, $\beta_1$, $\beta_2$, gradient-norm clipping |
| | M+Adam | multiplicative learning rate, warmup, multiplicative weight decay, $\epsilon$ |
| BF16 weights + FP8 compute | AdamW | learning rate, warmup, $\beta_1$, $\beta_2$, gradient-norm clipping |
| | M+Adam | multiplicative learning rate, warmup, $\beta_1$, multiplicative weight decay, $\epsilon$ |

AdamW+Kahan and AdamW+SR. Kahan compensation and stochastic rounding both help additive updates survive low-precision accumulation, but neither changes the underlying additive update geometry. This suggests that explicitly separating additive and multiplicative update mechanisms may provide benefits beyond improvements to the additive update alone, as in SR or Kahan compensation.

When compute is reduced to FP8 while weights remain BF16, M+Adam continues to improve over AdamW, with relative perplexity reductions of $3.2\%$, $6.2\%$, $6.9\%$, and $4.0\%$. The FP8-weight/FP8-compute regime is more challenging because the optimizer must repeatedly update weights that are quantized to FP8. Even in this setting, M+Adam improves over AdamW by $3.4\%$, $2.7\%$, $10.5\%$, and $3.0\%$. The 350M result is especially notable: M+Adam with FP8 weights and FP8 compute reaches 19.395 perplexity, narrowing the gap to BF16-weight AdamW despite using a coarser weight format.

The NVFP4/FP8 regime amplifies the failure mode of purely additive updates because the representable bins are even coarser. M+Adam gives its largest gains in this setting, re-

ducing perplexity over AdamW by $8.7\%$, $7.3\%$, $16.6\%$, and $7.2\%$ for 60M, 130M, 350M, and 1B models. These results directly support our central claim: as weight precision becomes more restrictive, additive–multiplicative updates become increasingly valuable.

Finally, the 1B results suggest that the gains extend beyond the 60M–350M sweep: even at $1\times$ Chinchilla, M+Adam improves over AdamW across all evaluated precision regimes.

### 5.5. Hyperparameter Transfer and Scaling Sensitivity

M+Adam introduces additional hyperparameters relative to AdamW, especially separate learning rates for the additive and multiplicative branches. A natural concern is whether these require exhaustive re-tuning across model sizes and token budgets. Our coordinate-descent results suggest that they do not: after tuning a base configuration, only a small subset of hyperparameters is sufficiently scaling-sensitive to warrant re-sweeping, while the remaining settings transfer across regimes.

*Table 4.* **Optimizer-state footprint for M+Adam with Apollo-E.** Bytes/param counts optimizer states only, not measured peak memory. Apollo-E compresses the additional exponent/multiplicative second-moment state introduced by M+Adam using a rank-4 approximation.

| Optimizer / state | State precision | 130M | | 350M | |
|---|---|---|---|---|---|
| | | Bytes/param | PPL | Bytes/param | PPL |
| AdamW, | BF16 | 4.000 | 23.587 | 4.000 | 19.156 |
| AdamW, | FP32 | 8.000 | 22.615 | 8.000 | 15.955 |
| M+Adam, Apollo-E rank 4 | BF16 | 4.754 | 21.979 | 4.377 | 16.550 |

This transferability is consistent with the optimizer design. The multiplicative branch controls relative scale changes, while the additive branch provides local weight-space updates, so a single learning rate does not need to simultaneously handle both scale adaptation and local corrections. Under a matched restricted re-tuning protocol, in which both the baselines and M+Adam re-sweep only the hyperparameters identified as scaling-sensitive (Table 3), M+Adam continues to achieve lower validation perplexity across the regimes reported in App. I. This suggests that the improvements are not simply an artifact of giving M+Adam a larger hyperparameter search space, but persist under a comparable tuning budget.

### 5.6. Optimizer-State Overhead and Apollo-E Compression

M+Adam removes FP32 master weights but adds one optimizer state: a second-moment accumulator for the multiplicative branch. Since prior work shows that optimizer states can often be compressed with low-rank approximations (Zhu et al., 2025; Zhao et al., 2024; Loeschcke et al., 2025), we test whether this added state can be compressed independently.

Apollo (Zhu et al., 2025) compresses optimizer states using low-rank gradient projections, storing and updating the optimizer statistics in a reduced-dimensional subspace. We apply Apollo only to the exponent/multiplicative second-moment accumulator; we refer to this variant as Apollo-E, where "E" denotes the exponent branch. We use rank $r = 4$ for both 130M and 350M models. The additive AdamW-style branch and the additive–multiplicative update rule are otherwise unchanged. Table 4 shows that Apollo-E keeps the optimizer-state footprint close to BF16-state AdamW while retaining most of M+Adam's perplexity gains. Detailed Apollo-E variants and rank sweeps are given in App. H.

### 6. Limitations and Future Work

M+Adam focuses on optimization geometry for low-precision weights rather than optimizer-state compression or systems-level acceleration. The current prototype introduces one additional optimizer state for the multiplicative branch and incurs runtime overhead relative to AdamW, e.g. roughly

$6\%-9\%$ at 1B scale (App. G). At lower precision, overhead also comes from simulating FP8/FP4 weight storage through explicit quantize/dequantize steps, since fully native low-bit parameter-update support is still limited in current training stacks. Our Apollo-E results suggest that the additional state can be compressed independently, while fused kernels and native low-precision update paths could further reduce the cost of the multiplicative branch.

Our theoretical analysis studies an idealized additive–multiplicative update and does not fully model practical ingredients such as stochastic gradients, Adam-style moments, clipping, or distributed training. Finally, while we evaluate up to 1B-parameter LLaMA-style models across BF16, FP8, and FP4 regimes, larger-scale studies, longer-horizon FP8/FP8 training, and broader downstream evaluations remain important future work. Another natural direction is to study whether M+Adam can be combined with stochastic rounding or Kahan-style compensated accumulation.

### 7. Conclusion

We present M+Adam, an optimizer that combines additive Adam-style updates with multiplicative Madam-style updates. This is effective under low-precision arithmetic: the additive branch provides local updates, while the multiplicative branch makes scale-aware relative updates in cases when the additive updates are rounded off to zero. Our analysis provides an idealized descent guarantee, and our experiments show that this geometry targets a concrete failure mode of low-precision training. Additive updates can get lost to rounding under coarse quantization, while multiplicative updates struggle with zeros and sign changes; combining them mitigates both failure modes. Across 60M-1B parameter LLaMA-style models, multiple Chinchilla budgets, and BF16/FP8/FP4 regimes, M+Adam consistently improves over AdamW, with the largest gains in the most aggressive low-precision settings. Overall, M+Adam shows that low-precision training benefits from optimizers aligned with floating-point structure. By directly addressing quantization-induced update failures, it provides a principled step toward end-to-end low-precision training without relying on FP32 master weights.

## Impact Statement

This work aims to advance the field of machine learning by enabling more efficient and stable low-precision training of large neural networks. While improved training efficiency may have downstream environmental and economic implications, we do not foresee any direct negative societal impacts arising specifically from this work.

## Acknowledgements

Sebastian Loeschcke is supported by the Danish Data Science Academy, which is funded by the Novo Nordisk Foundation (NNF21SA0069429) and VILLUM FONDEN (40516). Anima Anandkumar is supported in part by the Bren endowed chair, and the AI2050 Senior Fellow program at Schmidt Sciences. We acknowledge TACC and Caltech HPC for computing resources.

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

## A. Detailed Algorithm

We provide pseudocode for our method in Algorithm 2. Our presentation treats M+Adam as a *weight-space* optimizer with two coupled update components: an additive component and a multiplicative component. All operations are performed elementwise.

At each step, we first compute the stochastic gradient

$$g = \nabla_w \mathcal{L}(w).$$

The additive component follows an Adam-style update in weight space, using first and second moments to form a preconditioned direction and an additive update $u^{(a)}$. The multiplicative component is designed to capture relative changes in the weights: it starts from the scale-sensitive quantity

$$g^{(e)} = (\ln 2) \, wg,$$

applies variance normalization, and converts the resulting proposal into a relative multiplicative update $u^{(m)}$. For optional exponent-step clipping, this relative change can be mapped into a log-base-2 multiplicative increment

$$\Delta e = \log_2(1 + u^{(m)}).$$

The two components are then composed directly in weight space as

$$w^+ = w + wu^{(m)} + u^{(a)}.$$

The additive component keeps Adam-style first and second moments, whereas the multiplicative component uses second-moment normalization only. This design preserves the intended additive–multiplicative behavior while improving numerical stability in finite precision. We also allow optional multiplicative-step clipping, multiplicative weight decay, additive weight decay, and optional final weight clipping.

$$\text{clamp}_a(x) = \max(-a, \min(x, a))$$

where needed. This differs from standard global gradient-norm clipping: our cap acts on the multiplicative update component itself, rather than on the raw gradient. A small $\epsilon$ stabilizes divisions in both components, and a small threshold $\tau > 0$ is used when converting multiplicative weight proposals into relative changes near zero.

---

**Algorithm 2** M+Adam: additive–multiplicative updates

---

1: **Inputs:** additive and multiplicative step sizes $\eta_a, \eta_m$; moments $\beta_1, \beta_2$; stabilizer $\epsilon$; small constant $\tau > 0$; optional clipping thresholds $g_{\max}, d_{\mathrm{mult,max}}, w_{\max}$
2: **State:** additive moments $(\mu, \nu) \leftarrow \mathbf{0}$, multiplicative second moment $\nu^{(e)} \leftarrow \mathbf{0}$
3: **Helper:** $\mathrm{clamp}_a(x) \triangleq \max(-a, \min(x, a))$              (elementwise)
4: **repeat**
5:   $g \leftarrow \nabla_w \mathcal{L}(w)$              ▷*stochastic gradient in weight space*
6:   $\mu \leftarrow \beta_1 \mu + (1 - \beta_1) g$
7:   $\nu \leftarrow \beta_2 \nu + (1 - \beta_2) g^2$
8:   $u_{\mathrm{add}} \leftarrow \mu / (\sqrt{\nu} + \epsilon)$        ▷*Adam-style preconditioned additive direction*
9:   $u^{(a)} \leftarrow -\eta_a\, u_{\mathrm{add}}$
10:   $g^{(e)} \leftarrow (\ln 2)\, wg$       ▷*coordinate-wise gradient with respect to exponent signal*
11:   $\nu^{(e)} \leftarrow \beta_2 \nu^{(e)} + (1 - \beta_2)\big(g^{(e)}\big)^2$
12:   $u_{\mathrm{mult}} \leftarrow \mathrm{clamp}_{g_{\max}}\big(g^{(e)} / (\sqrt{\nu^{(e)}} + \epsilon)\big)$
13:   $\widetilde{u}^{(m)} \leftarrow -\eta_m\, u_{\mathrm{mult}}$
14:   $\rho \leftarrow \mathrm{sign}(w) \max(|w|, \tau)$
15:   $u^{(m)} \leftarrow \widetilde{u}^{(m)} / \rho$       ▷*convert multiplicative proposal to relative change*
16:   $u^{(m)} \leftarrow \max(-(0.75 - \varepsilon_0), \min(0.75, u^{(m)}))$      ▷*keep* $\log(1 + u^{(m)})$ *stable*
17:   $\Delta e \leftarrow \log_2(1 + u^{(m)})$
18:   $\Delta e \leftarrow \mathrm{clamp}_{d_{\mathrm{mult,max}}}(\Delta e)$          ▷*optional per-step cap*
19:   $u^{(m)} \leftarrow 2^{\Delta e} - 1$         ▷*relative multiplicative update after clipping*
20:   $w \leftarrow w + wu^{(m)} + u^{(a)}$   ▷*compose additive shift and multiplicative rescaling directly in weight space*
21:   **if** using weight clipping **then**
22:    $w \leftarrow \mathrm{clamp}_{w_{\max}}(w)$         ▷$w \in [-w_{\max}, w_{\max}]$
23:   **end if**
24: **until** converged

---

## B. BF16-Stored Matrix Fitting to Isolate Low-Precision Failure Modes

We describe the minimal convex diagnostic used to isolate how update geometry interacts with low-precision weight storage. The goal is explanatory rather than competitive: we remove data and architecture and study only whether an optimizer update survives repeated BF16 storage.

**Setup.** We minimize

$$\min_{W \in \mathbb{R}^{n \times n}} \mathcal{L}(W) = c\|W - W^\star\|_F^4, \tag{7}$$

We adopt this loss rather than the typical squared error, to prevent gradient descent from converging in a single step. For this reason we also introduce stochastic samples of the entries and calculate the loss as the average of the sampled entries. We pick $B$ samples uniformly at random with replacement. Each step is computed in FP32, but the parameter is immediately rounded through BF16 before the next iteration:

$$\widetilde{W}_{t+1} = \mathrm{Update}(W_t, G_t), \qquad W_{t+1} = \mathrm{RoundBF16}(\widetilde{W}_{t+1}). \tag{8}$$

Thus, failures in this experiment come only from the interaction between the update rule and BF16 storage.

**Update rules.** Additive is a simple additive gradient step:

$$u_t^{(a)} = -\eta_a g_t. \tag{9}$$

For Multiplicative, we first form the coordinate-wise gradient with respect to exponent signal

$$g_t^{(e)} = (\ln 2)\, w_t g_t.$$

We then apply the multiplicative update

$$\widetilde{u}_t^{(m)} = -\eta_m g_t^{(e)},$$

$$\rho_t = \text{sign}_+(w_t) \max(|w_t|, \tau),$$

$$u_t^{(m)} = \text{clip}\left(\frac{\widetilde{u}_t^{(m)}}{\rho_t}, -r_{\max}, r_{\max}\right),$$

$$w_{t+1} = \text{bf16}\left(w_t + w_t u_t^{(m)}\right).$$

We set $r_{\max} = 0.75$ and $\tau = 10^{-8}$.

For M+Adam, we follow the same steps as above but apply both branches:

$$w_{t+1} = \text{bf16}\left(w_t + w_t u_t^{(m)} + u_t^{(a)}\right).$$

**Diagnostic regimes.** We use five regimes, each chosen to expose a specific low-precision failure mode:

1. **Sign flip:** success requires changing signs.

2. **Zero revival:** success requires escaping exact zero initialization.

3. **Small weights:** success requires fine local updates near zero.

4. **Large weights:** success requires updates that remain effective under coarse BF16 spacing.

5. **Mixed scales:** success requires handling small and large coordinates simultaneously.

- **Sign flip.**
$$n = 4096, \qquad c = 1.$$

  Targets are random signs:
  $$w_i^\star \in \{-1, +1\},$$

  and the initialization has the opposite sign:
  $$w_{0,i} = -w_i^\star.$$

- **Zero revival.**
$$n = 4096, \qquad c = 1.$$

  Targets are random signs:
  $$w_i^\star \in \{-1, +1\},$$

  and all weights start exactly at zero:
  $$w_{0,i} = 0.$$

- **Small weights.**
$$n = 4096, \qquad c = 10^8.$$

  Targets are small Gaussian values:
  $$w_i^\star = 10^{-4} z_i, \qquad z_i \sim \mathcal{N}(0, 1),$$

  and all weights start at zero:
  $$w_{0,i} = 0.$$

- **Large weights.**
$$n = 4096, \qquad c = 10^{-7}.$$

  Targets are positive large-scale values:
  $$w_i^\star = 900 \max(|z_i|, 1/900), \qquad z_i \sim \mathcal{N}(0, 1),$$

  so $w_i^\star \geq 1$. All weights start at
  $$w_{0,i} = 100.$$

- **Mixed scales.**

$$n = 4096.$$

The first half uses the small-weights regime:

$$i \leq 2048: \qquad w_i^\star = 10^{-4} z_i, \quad w_{0,i} = 0, \quad c_i = 10^8.$$

The second half uses the large-weights regime:

$$i > 2048: \qquad w_i^\star = 900 \max(|z_i|, 1/900), \quad w_{0,i} = 100, \quad c_i = 10^{-7}.$$

For fairness, we tune learning rates separately for each optimizer and regime. For the main diagnostic figure, M+Adam sets $\eta_a = \eta_m$ and uses one fixed shared learning rate across all regimes. The additive component handles sign changes, recovery from zero, and small-magnitude updates, while the multiplicative component controls scale-aware progress at large magnitudes. Fig. 5 shows the best sweep curves.

**Main observations.** The results support four qualitative conclusions. First, multiplicative-only updates fail when the solution requires sign changes or escaping exact zero initialization; these are structural limitations, not learning-rate artifacts. Second, additive updates are effective for sign changes, zero revival, and local updates, but can be rounded away at large magnitudes because BF16 spacing grows with $|W_t|$. Third, mixed-scale problems expose a conflict for purely additive updates: a step size that affects large entries can be too coarse for small entries, while a step size that updates small entries may not survive rounding for large entries. Finally, M+Adam avoids the main weakness of either branch alone: the additive component handles local updates, sign changes, and zero revival, while the multiplicative component gives scale-aware progress at large magnitudes.

**Summary.** This toy problem is intentionally simple, but it provides a plausible mechanism behind the main empirical LLM results. Purely additive updates fail when quantization bins become too wide; purely multiplicative updates fail near zero and across sign changes. M+Adam combines the two geometries, giving relative progress at large magnitudes while retaining the flexibility of additive optimization near zero.

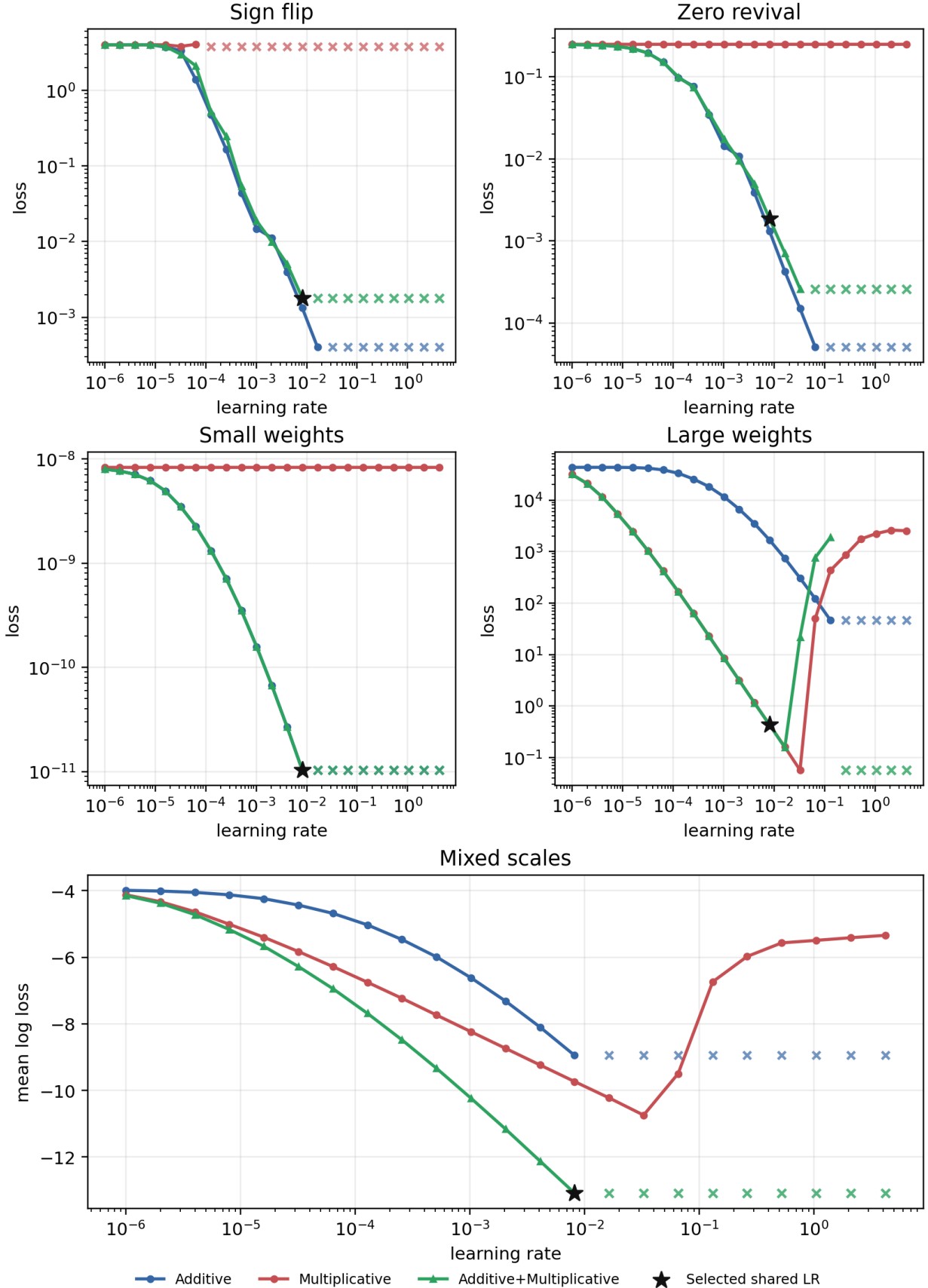

*Figure 5.* Learning-rate sweep curves for the BF16-stored matrix-fitting diagnostic.

## C. Hyperparameter Sensitivity

We claim reduced hyperparameter sensitivity in BF16. As direct evidence, we report learning-rate sensitivity for optimizers AdamW and M+Adam under the same BF16 training setting and token budget ($1\times$Chinchilla), across three model sizes (60M/130M/350M). For AdamW we sweep the global learning rate. For M+Adam we separately sweep the mantissa step size $\text{lr}_m$ and exponent step size $\text{lr}_e$ around the tuned configuration (holding all other hyperparameters fixed as in the sweep tables). We plot the resulting validation losses; for readability under occasional divergence/outliers, the visualization uses a log-scaled view of loss deviations so that stable regions remain visually comparable. These 1D sweeps provide a minimal, direct sensitivity check supporting the claim of improved robustness.

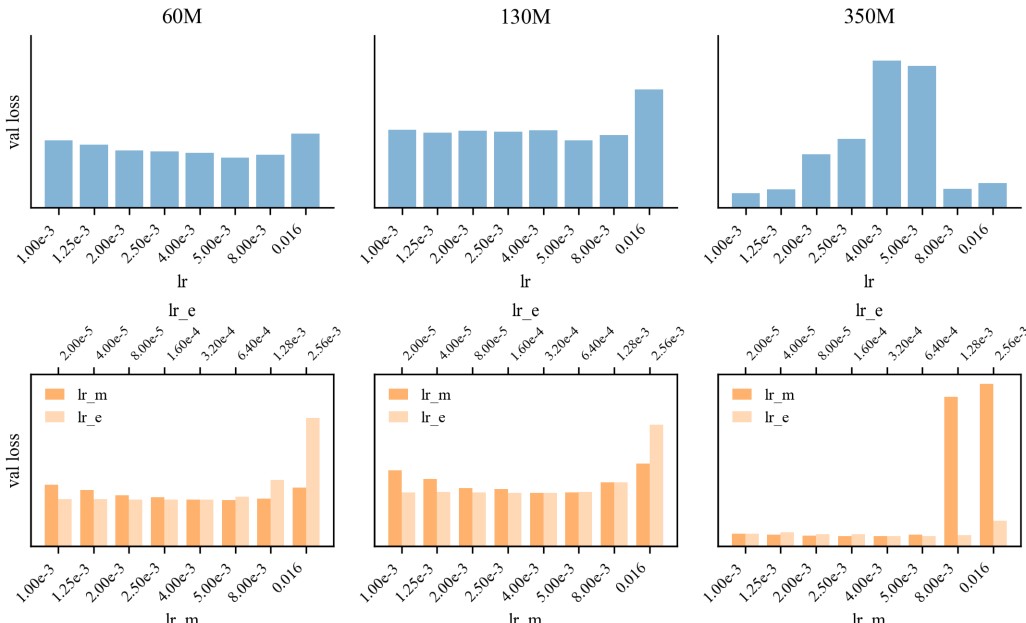

*Figure 6.* **Learning-rate sensitivity in BF16 (AdamW vs. M+Adam).** Columns correspond to model sizes (60M, 130M, 350M). The top row shows AdamW validation loss versus learning rate. The bottom row shows M+Adam, where the bottom x-axis sweeps $\text{lr}_m$ and the top x-axis sweeps $\text{lr}_e$; bars correspond to the same sweep index for the paired rates. Overall, M+Adam exhibits a stable region across learning-rate choices.

## D. Proofs

### D.1. Proof of the Scale-Then-Add Descent Theorem

*Proof of Theorem 4.1.* Since $z = \log_2 s$, the update can be written as

$$w_{t+1} = w_t 2^z + u^{(a)}.$$

Applying the local smoothness bound in Eq. (5) with $d = u^{(a)}$ gives

$$\mathcal{L}(w_{t+1}) \leq \mathcal{L}(w_t) + \langle g, u^{(a)} \rangle + \langle g^{(e)}, z \rangle$$
$$+ \frac{L_a}{2}\|u^{(a)}\|^2 + \frac{L_e}{2}\|z\|^2 + L_{ae}\|u^{(a)}\|\,\|z\|.$$

Using the descent-alignment and size assumptions yields

$$\mathcal{L}(w_{t+1}) \leq \mathcal{L}(w_t) - \eta_a\|g\|^2 - \eta_m\|g^{(e)}\|^2$$
$$+ \frac{L_a}{2}\eta_a^2\|g\|^2 + \frac{L_e}{2}\eta_m^2\|g^{(e)}\|^2$$
$$+ L_{ae}\eta_a\eta_m\|g\|\,\|g^{(e)}\|.$$

By Young's inequality,

$$\|g\|\,\|g^{(e)}\| \leq \frac{1}{2}\left(\|g\|^2 + \|g^{(e)}\|^2\right).$$

Therefore,

$$\mathcal{L}(w_{t+1}) \leq \mathcal{L}(w_t) - \eta_a \|g\|^2 - \eta_m \|g^{(e)}\|^2$$
$$+ \frac{L_a}{2} \eta_a^2 \|g\|^2 + \frac{L_e}{2} \eta_m^2 \|g^{(e)}\|^2$$
$$+ \frac{L_{ae}}{2} \eta_a \eta_m \left( \|g\|^2 + \|g^{(e)}\|^2 \right).$$

Collecting the coefficients of $\|g\|^2$ and $\|g^{(e)}\|^2$ gives

$$\mathcal{L}(w_{t+1}) \leq \mathcal{L}(w_t)$$
$$- \frac{\eta_a}{2} \left( 2 - L_a \eta_a - L_{ae} \eta_m \right) \|g\|^2$$
$$- \frac{\eta_m}{2} \left( 2 - L_e \eta_m - L_{ae} \eta_a \right) \|g^{(e)}\|^2.$$

The learning-rate conditions make both bracketed coefficients positive. If $g \neq 0$, then the additive descent term is strictly negative, so $\mathcal{L}(w_{t+1}) < \mathcal{L}(w_t)$. This proves the claim. $\qquad\square$

**Relation to Algorithm 1.** Algorithm 1 computes an additive update $u^{(a)}$ and a relative multiplicative update $u^{(m)}$. With

$$s = 1 + u^{(m)},$$

its core composition is

$$w_{t+1} = w_t + w_t u^{(m)} + u^{(a)}$$
$$= w_t s + u^{(a)}.$$

When $s > 0$ elementwise, this is equivalently

$$w_{t+1} = w_t 2^z + u^{(a)}, \qquad z = \log_2 s.$$

The theorem analyzes this same scale-then-add map in the local coordinates $(d, z)$, with $d = u^{(a)}$. The adaptive moments, normalization, thresholding, and clipping in Algorithm 1 determine the concrete branch outputs $u^{(a)}$ and $u^{(m)}$; the descent statement only requires these outputs to satisfy the stated alignment and size conditions. Thus the theorem captures the local additive–multiplicative geometry of the algorithm without committing to a particular implementation of the branch preconditioners.

# E. Model Architectures

| Model | Params | Max Seq Len | Hidden Dim | Inter Dim | # Layers | # Heads |
|-------|--------|-------------|------------|-----------|----------|---------|
| LLaMA-60M | 60M | 1024 | 512 | 1376 | 8 | 8 |
| LLaMA-130M | 130M | 1024 | 768 | 2048 | 12 | 12 |
| LLaMA-350M | 350M | 1024 | 1024 | 2736 | 24 | 16 |
| LLaMA-1B | 1B | 1024 | 2048 | 5461 | 24 | 32 |

*Table 5.* Detailed architecture hyperparameters for each model size we studied.

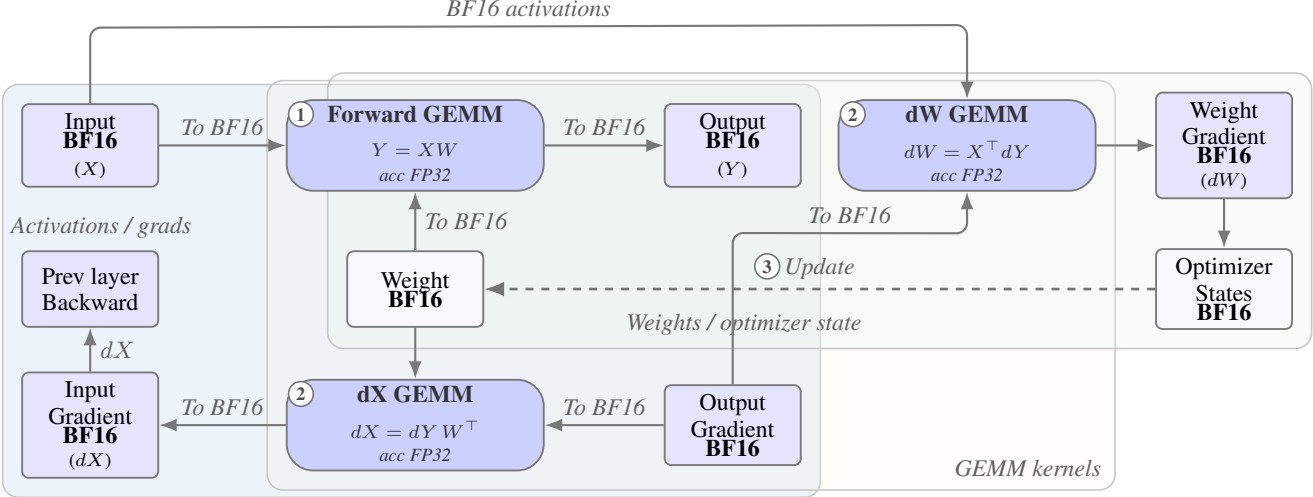

*Figure 7.* **BF16 End-to-End Training Dataflow with BF16 GEMMs and BF16 Optimizer States** Forward/backward use BF16 activations and BF16 weights; GEMM partial sums accumulate in FP32; gradients and optimizer states are maintained in BF16.

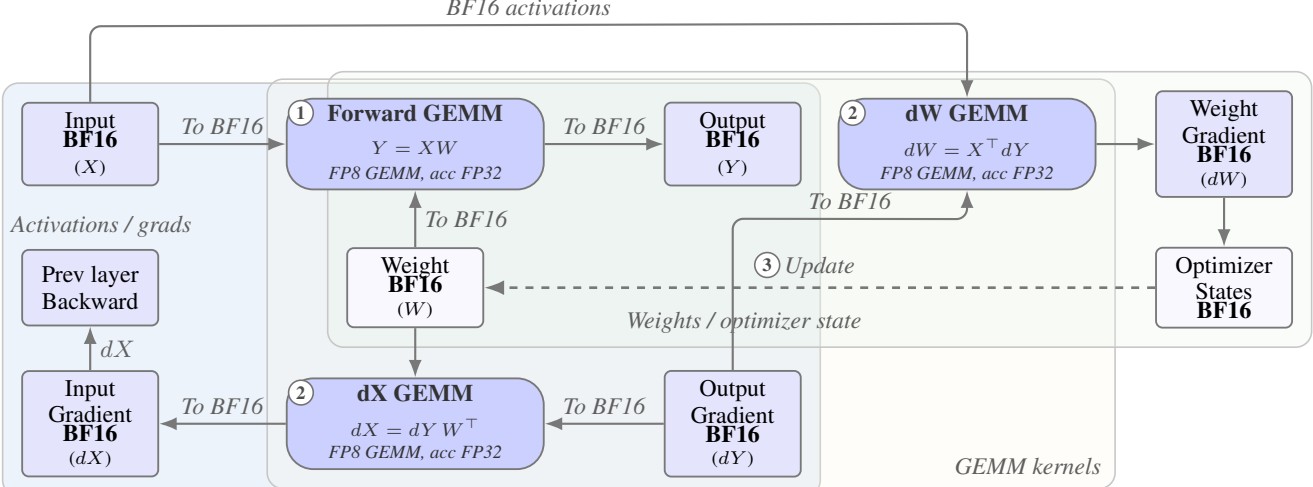

*Figure 8.* **BF16 Training Dataflow with TE FP8 GEMMs and BF16 Optimizer States** Activations, outputs, and gradients are stored in BF16. Transformer Engine casts GEMM operands to FP8 (delayed scaling) and accumulates partial sums in FP32 before returning BF16 results; optimizer states are maintained in BF16 and applied to update BF16 weights.

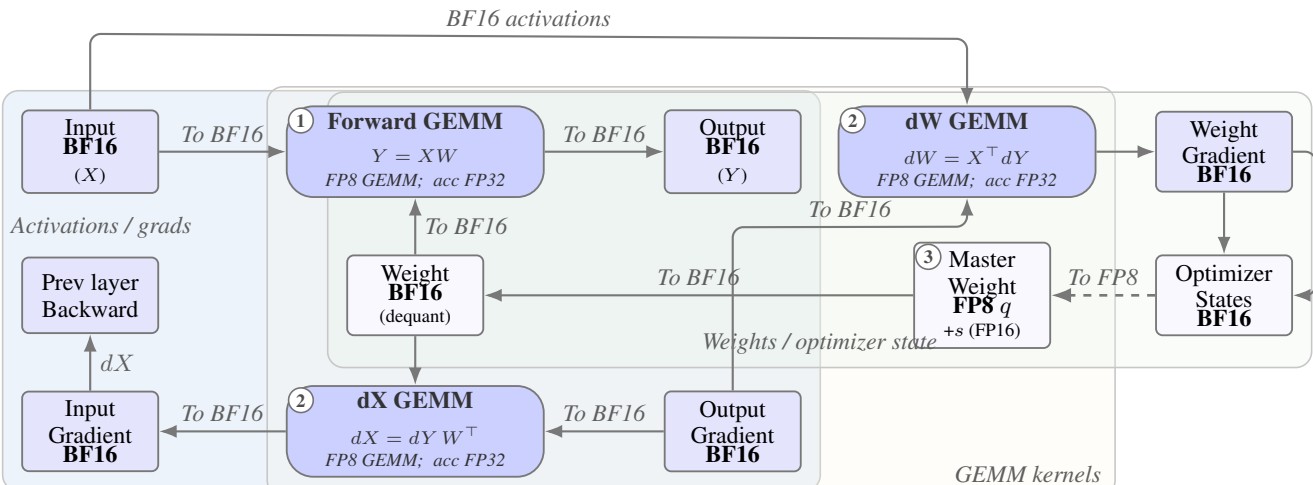

*Figure 9.* **BF16 Training with TE FP8 GEMMs and True FP8-Master Weights** Activations, outputs, and backpropagated gradients are stored in BF16. For the GEMM kernels, Transformer Engine casts BF16 operands to FP8 and accumulates partial sums in FP32 before returning BF16 outputs/gradients. For selected Linear/MLP weights, the only weight value is FP8 ($q$) with FP16 block scales ($s$); a BF16 compute copy is refreshed by dequantization each step.

## F. Memory Comparison

*Table 6.* Per-parameter memory accounting in bytes. BF16/FP32/FP8 use 2/4/1 bytes. **Persistent** counts master weights and optimizer states; **With gradient** additionally includes a BF16 gradient buffer. SR denotes stochastic rounding when writing updated weights to BF16.

| Method | Master weights | Optimizer states | Persistent | With gradient |
|---|---|---|---|---|
| AdamW | FP32 master $(4)$ | $(m, v)$ BF16 $(2 \times 2)$ | $4 + 4 = 8$ | $8 + $ BF16 $g$ $(2) = 10$ |
| M+Adam (BF16 master) | BF16 $(2)$ | $(\mu_m, \nu_m, \nu_e)$ BF16 $(3 \times 2)$ | $2 + 6 = 8$ | $8 + $ BF16 $g$ $(2) = 10$ |
| M+Adam (FP8 master) | FP8 $(1)$ | $(\mu_m, \nu_m, \nu_e)$ BF16 $(3 \times 2)$ | $1 + 6 = 7$ | $7 + $ BF16 $g$ $(2) = 9$ |
| AdamW + SR (BF16) | BF16 $(2)$ | $(m, v)$ BF16 $(2 \times 2)$ | $2 + 4 = 6$ | $6 + $ BF16 $g$ $(2) = 8$ |

Table 6 summarizes the per-parameter memory accounting. In its uncompressed version, M+Adam replaces FP32 master weights with lower-precision weights, but introduces an additional optimizer state for the multiplicative branch. As a result, the BF16-master version has the same memory footprint as AdamW with FP32 master weights in this accounting, while the FP8-master version is slightly smaller. Thus, M+Adam should not be interpreted as an optimizer-state memory-reduction method by itself. Its contribution is instead to improve the numerical stability of low-precision weight updates and reduce reliance on FP32 master weights. Our Apollo-E results in App. H further suggest that the added multiplicative state can be compressed independently, making optimizer-state compression complementary to the proposed additive–multiplicative update geometry.

## G. Compute Resources and Runtime Overhead

We report representative compute resources and wall-clock runtimes for the main $1\times$ Chinchilla experiments in Tables 7 and 8. All runs were executed on NVIDIA H200 GPUs using distributed training. The reported wall-clock time measures end-to-end training time for the corresponding final configuration, including forward and backward passes, optimizer updates, evaluation overhead, and logging overhead as used in our training pipeline. Per-update time is computed as wall-clock time divided by the number of optimizer steps.

The overhead numbers compare our current M+Adam implementation against AdamW within the same training stack and kernel configuration. They should therefore be interpreted as implementation-level overheads rather than intrinsic lower bounds for the additive–multiplicative update rule. In particular, our current implementation simulates some low-precision storage paths, such as FP8/FP4 master weights, through explicit quantize/dequantize operations for compatibility with existing Transformer Engine kernels. These software-simulation steps can introduce additional overhead that would not necessarily remain in a fused or fully native low-precision implementation.

These measurements are intended to quantify the practical overhead of M+Adam and AdamW+SR relative to AdamW in our implementation. They do not include the full research compute used for preliminary experiments, failed runs, debugging, or hyperparameter sweeps. Thus, the total project compute was larger than the compute required to reproduce the final reported configurations.

*Table 7.* **End-to-end runtime and per-update overhead of M+Adam relative to AdamW at $1\times$ Chinchilla.** We report wall-clock time, seconds per optimizer update, relative overhead, and hardware for the final training configurations.

| Model | Precision | Steps | AdamW wall-clock | M+Adam wall-clock | AdamW s/upd | M+Adam s/upd | Overhead | Hardware |
|---|---|---|---|---|---|---|---|---|
| 350M | FP8/FP8 | 60,000 | 5h 58m 39s | 6h 23m 41s | 0.359 | 0.384 | 6.98% | 4×H200 |
| 130M | FP8/FP8 | 20,000 | 1h 10m 30s | 1h 15m 28s | 0.212 | 0.226 | 7.04% | 2×H200 |
| 60M | FP8/FP8 | 10,000 | 22m 17s | 25m 03s | 0.134 | 0.150 | 12.42% | 2×H200 |
| 350M | BF16/FP8 | 60,000 | 3h 35m 10s | 4h 48m 26s | 0.215 | 0.288 | 34.05% | 4×H200 |
| 130M | BF16/FP8 | 20,000 | 1h 29m 55s | 1h 41m 10s | 0.270 | 0.304 | 12.51% | 1×H200 |
| 60M | BF16/FP8 | 10,000 | 25m 51s | 30m 50s | 0.155 | 0.185 | 19.28% | 1×H200 |
| 350M | BF16/BF16 | 60,000 | 3h 31m 38s | 4h 44m 50s | 0.212 | 0.285 | 34.59% | 4×H200 |
| 130M | BF16/BF16 | 20,000 | 1h 30m 11s | 1h 44m 20s | 0.271 | 0.313 | 15.69% | 1×H200 |
| 60M | BF16/BF16 | 10,000 | 22m 38s | 27m 16s | 0.136 | 0.164 | 20.47% | 1×H200 |
| 1B | BF16/FP8 | 100,000 | 108h 07m 42s | 117h 33m 16s | 3.893 | 4.232 | 8.72% | 4×H200 |
| 1B | BF16/BF16 | 100,000 | 103h 29m 15s | 109h 55m 08s | 3.726 | 3.957 | 6.21% | 4×H200 |

Table 7 shows that the overhead of M+Adam depends on the precision regime and model size. The overhead is smallest in

the FP8/FP8 and 1B settings, where the cost of the additional multiplicative branch is relatively small compared to the total training step. In contrast, the BF16/BF16 and BF16/FP8 settings at smaller scales show larger relative overheads, reflecting that the optimizer computation and extra state operations occupy a larger fraction of total runtime when the model is smaller or compute kernels are less dominant.

*Table 8.* **End-to-end runtime and per-update overhead of AdamW+SR relative to AdamW for BF16/BF16 training at** $1\times$ **Chinchilla.** We report wall-clock time, seconds per optimizer update, relative overhead, and hardware for the stochastic-rounding baseline.

| Model | Steps | AdamW wall-clock | AdamW+SR wall-clock | AdamW s/upd | AdamW+SR s/upd | Overhead | Hardware |
|---|---|---|---|---|---|---|---|
| 350M | 60,000 | 3h 31m 38s | 5h 03m 57s | 0.212 | 0.304 | 43.62% | $4\times$H200 |
| 130M | 20,000 | 1h 30m 11s | 1h 46m 25s | 0.271 | 0.319 | 18.00% | $1\times$H200 |
| 60M | 10,000 | 22m 38s | 28m 09s | 0.136 | 0.169 | 24.37% | $1\times$H200 |

Table 8 provides the corresponding runtime comparison for AdamW with stochastic rounding in the BF16/BF16 regime. In our implementation, AdamW+SR also introduces nontrivial overhead relative to standard AdamW. This comparison is useful because AdamW+SR is a strong additive baseline for mitigating BF16 round-to-nearest effects, but its runtime cost is not negligible. Together, Tables 7 and 8 show that M+Adam improves low-precision optimization stability at the cost of additional runtime, while also making clear that stochastic-rounding baselines carry their own systems overhead.

## H. Apollo-E Compression for the Multiplicative State

M+Adam introduces one additional optimizer state beyond AdamW: the second-moment accumulator used by the multiplicative branch. This state improves the stability of exponent-style updates, but also increases optimizer-state memory. To test whether this overhead is intrinsic to the method, we evaluate Apollo-E, a compressed variant that replaces only this extra multiplicative second-moment state with an Apollo-style approximation. The additive AdamW-style branch is unchanged.

Apollo-E compresses the multiplicative-state normalizer rather than changing the additive–multiplicative update rule. We evaluate two modes: mini, which uses the most aggressive tensor-wise compression, and channel, which uses a channel-wise compressed representation with rank controlling the auxiliary dimension. Larger ranks store slightly more state but can better approximate the dense multiplicative normalizer.

Table 9 reports optimizer-state memory in bytes per trainable parameter. These numbers count tensors stored in the optimizer state only; they do not include model weights, gradients, activations, temporary buffers, CUDA allocator effects, or communication buffers. Thus, the table should be interpreted as optimizer-state accounting rather than measured peak GPU memory.

*Table 9.* **Apollo-E compression of the additional multiplicative state.** Apollo-E compresses only the extra multiplicative second-moment state introduced by M+Adam. Bytes/param counts optimizer-state tensors only and is not measured peak memory.

| Model | Optimizer / state | Bytes/param | PPL |
|---|---|---|---|
| 130M | AdamW, BF16 state | 4.000 | 23.587 |
| 130M | AdamW, FP32 state | 8.000 | 22.615 |
| 130M | Apollo-E M+Adam, mini, rank 1 | 4.739 | 22.126 |
| 130M | Apollo-E M+Adam, channel, rank 1 | 4.739 | 22.085 |
| 130M | Apollo-E M+Adam, channel, rank 4 | 4.754 | 21.979 |
| 130M | Apollo-E M+Adam, channel, rank 8 | 4.775 | 21.997 |
| 130M | Apollo-E M+Adam, channel, rank 16 | 4.817 | 21.995 |
| 350M | AdamW, BF16 state | 4.000 | 19.156 |
| 350M | AdamW, FP32 state | 8.000 | 15.955 |
| 350M | Apollo-E M+Adam, mini, rank 1 | 4.362 | 16.752 |
| 350M | Apollo-E M+Adam, channel, rank 1 | 4.362 | 16.683 |
| 350M | Apollo-E M+Adam, channel, rank 4 | 4.377 | 16.550 |
| 350M | Apollo-E M+Adam, channel, rank 8 | 4.397 | 16.529 |
| 350M | Apollo-E M+Adam, channel, rank 16 | 4.438 | 16.576 |

Apollo-E reduces the optimizer-state overhead of M+Adam to a small increase over BF16-state AdamW while preserving most of the optimization gain. For 130M models, Apollo-E uses roughly $4.74$–$4.82$ bytes per parameter, substantially below FP32-state AdamW, while improving perplexity over both BF16-state and FP32-state AdamW. For 350M models, Apollo-E uses only $4.36$–$4.44$ bytes per parameter and remains much closer to full M+Adam performance than to BF16 AdamW.

These results show that the additional multiplicative state is not fundamental memory overhead: it can be compressed independently of the additive–multiplicative update geometry.

# I. Hyperparameter Ablation

We report hyperparameter sweep results for the optimizers studied in the main text. Each table is organized as follows: the first row lists the approximately best configuration found in the sweep, and the subsequent rows report one-dimensional ablations that vary a single hyperparameter while holding all others fixed at this reference setting. Reported losses are the final C4/EN validation losses. We consider four precision regimes throughout: **(i) BF16 weights + BF16 compute**, where model parameters, activations, and optimizer state are stored in BF16; **(ii) BF16 weights + FP8 compute**, where parameters are maintained in BF16 master weights while Transformer Engine executes GEMMs in FP8; **(iii) FP8 weights + FP8 compute**, where parameters are stored in FP8 master weights and Transformer Engine executes GEMMs in FP8; and **(iv) FP32 weights + TE32 compute**, where parameters are stored in FP32 and Transformer Engine executes GEMMs in TE32.

## I.1. Sweeping Results for AdamW (BF16 weights + BF16 compute)

*Table 10.* Hyperparameter ablation for AdamW on 60M on $1\times$ Chinchilla Data (BF16 weights + BF16 compute)

| $\beta_1$ | $\beta_2$ | $\epsilon$ | $\eta$ | $g_{\text{norm}}$ | warmup | wd | Loss | Link |
|------|------|------|------|------|------|------|------|------|
| 0.90 | 0.95 | $10^{-10}$ | 0.004 | 1.0 | 1000 | 0.5 | 3.4155 | 1 |
| **0.87** | 0.95 | $10^{-10}$ | 0.004 | 1.0 | 1000 | 0.5 | 3.4054 | 2 |
| **0.90** | 0.95 | $10^{-10}$ | 0.004 | 1.0 | 1000 | 0.5 | 3.4155 | 3 |
| **0.95** | 0.95 | $10^{-10}$ | 0.004 | 1.0 | 1000 | 0.5 | 3.4617 | 4 |
| **0.98** | 0.95 | $10^{-10}$ | 0.004 | 1.0 | 1000 | 0.5 | 3.7057 | 5 |
| 0.87 | **0.90** | $10^{-10}$ | 0.004 | 1.0 | 1000 | 0.5 | 3.4364 | 6 |
| 0.87 | **0.95** | $10^{-10}$ | 0.004 | 1.0 | 1000 | 0.5 | 3.4054 | 7 |
| 0.87 | **0.98** | $10^{-10}$ | 0.004 | 1.0 | 1000 | 0.5 | 3.4041 | 8 |
| 0.87 | 0.95 | $10^{-10}$ | 0.004 | 1.0 | 1000 | 0.5 | 3.4054 | 9 |
| 0.87 | 0.95 | $10^{-20}$ | 0.004 | 1.0 | 1000 | 0.5 | 3.4066 | 10 |
| 0.87 | 0.95 | $10^{-10}$ | **0.004** | 1.0 | 1000 | 0.5 | 3.4054 | 11 |
| 0.87 | 0.95 | $10^{-10}$ | **0.008** | 1.0 | 1000 | 0.5 | 3.3967 | 12 |
| 0.87 | 0.95 | $10^{-10}$ | **0.016** | 1.0 | 1000 | 0.5 | 3.4932 | 13 |
| 0.87 | 0.95 | $10^{-10}$ | 0.008 | **0.0** | 1000 | 0.5 | 3.3983 | 14 |
| 0.87 | 0.95 | $10^{-10}$ | 0.008 | **1.0** | 1000 | 0.5 | 3.3967 | 15 |
| 0.87 | 0.95 | $10^{-10}$ | 0.008 | **2.0** | 1000 | 0.5 | 3.3960 | 16 |
| 0.87 | 0.95 | $10^{-10}$ | 0.008 | 1.0 | **500** | 0.5 | 3.4084 | 17 |
| 0.87 | 0.95 | $10^{-10}$ | 0.008 | 1.0 | **1000** | 0.5 | 3.3967 | 18 |
| 0.87 | 0.95 | $10^{-10}$ | 0.008 | 1.0 | **2000** | 0.5 | 3.4039 | 19 |
| 0.87 | 0.95 | $10^{-10}$ | 0.008 | 1.0 | 1000 | **0.0** | 3.4160 | 20 |
| 0.87 | 0.95 | $10^{-10}$ | 0.008 | 1.0 | 1000 | **0.5** | 3.3967 | 21 |
| 0.87 | 0.95 | $10^{-10}$ | 0.008 | 1.0 | 1000 | **1.0** | 3.5651 | 22 |

*Table 11.* Hyperparameter ablation for AdamW on 60M on $2\times$ Chinchilla Data (BF16 weights + BF16 compute)

| $\beta_1$ | $\beta_2$ | $\epsilon$ | $\eta$ | $g_{\text{norm}}$ | warmup | wd | Loss | Link |
|---|---|---|---|---|---|---|---|---|
| 0.90 | 0.95 | $10^{-10}$ | 0.004 | 1.0 | 1000 | 0.5 | 3.3486 | 1 |
| **0.87** | 0.95 | $10^{-10}$ | 0.004 | 1.0 | 1000 | 0.5 | 3.3358 | 2 |
| **0.90** | 0.95 | $10^{-10}$ | 0.004 | 1.0 | 1000 | 0.5 | 3.3486 | 3 |
| **0.95** | 0.95 | $10^{-10}$ | 0.004 | 1.0 | 1000 | 0.5 | 3.4005 | 4 |
| **0.98** | 0.95 | $10^{-10}$ | 0.004 | 1.0 | 1000 | 0.5 | 3.5797 | 5 |
| 0.87 | **0.90** | $10^{-10}$ | 0.004 | 1.0 | 1000 | 0.5 | 3.3740 | 6 |
| 0.87 | **0.95** | $10^{-10}$ | 0.004 | 1.0 | 1000 | 0.5 | 3.3358 | 7 |
| 0.87 | **0.98** | $10^{-10}$ | 0.004 | 1.0 | 1000 | 0.5 | 3.3340 | 8 |
| 0.87 | 0.95 | $10^{-10}$ | 0.004 | 1.0 | 1000 | 0.5 | 3.3358 | 9 |
| 0.87 | 0.95 | $10^{-20}$ | 0.004 | 1.0 | 1000 | 0.5 | 3.3371 | 10 |
| 0.87 | 0.95 | $10^{-10}$ | **0.004** | 1.0 | 1000 | 0.5 | 3.3358 | 11 |
| 0.87 | 0.95 | $10^{-10}$ | **0.008** | 1.0 | 1000 | 0.5 | 3.3163 | 12 |
| 0.87 | 0.95 | $10^{-10}$ | **0.016** | 1.0 | 1000 | 0.5 | 3.4010 | 13 |
| 0.87 | 0.95 | $10^{-10}$ | 0.008 | **0.0** | 1000 | 0.5 | 3.3335 | 14 |
| 0.87 | 0.95 | $10^{-10}$ | 0.008 | **1.0** | 1000 | 0.5 | 3.3163 | 15 |
| 0.87 | 0.95 | $10^{-10}$ | 0.008 | **2.0** | 1000 | 0.5 | 3.3156 | 16 |
| 0.87 | 0.95 | $10^{-10}$ | 0.008 | 1.0 | **500** | 0.5 | 3.3250 | 17 |
| 0.87 | 0.95 | $10^{-10}$ | 0.008 | 1.0 | **1000** | 0.5 | 3.3163 | 18 |
| 0.87 | 0.95 | $10^{-10}$ | 0.008 | 1.0 | **2000** | 0.5 | 3.3163 | 19 |
| 0.87 | 0.95 | $10^{-10}$ | 0.008 | 1.0 | 1000 | **0.0** | 3.3454 | 20 |
| 0.87 | 0.95 | $10^{-10}$ | 0.008 | 1.0 | 1000 | **0.5** | 3.3163 | 21 |
| 0.87 | 0.95 | $10^{-10}$ | 0.008 | 1.0 | 1000 | **1.0** | 3.4692 | 22 |

*Table 12.* Hyperparameter ablation for AdamW on 60M on $4\times$ Chinchilla Data (BF16 weights + BF16 compute)

| $\beta_1$ | $\beta_2$ | $\epsilon$ | $\eta$ | $g_{\text{norm}}$ | warmup | wd | Loss | Link |
|---|---|---|---|---|---|---|---|---|
| 0.90 | 0.95 | $10^{-10}$ | 0.004 | 1.0 | 1000 | 0.5 | 3.2994 | 1 |
| **0.87** | 0.95 | $10^{-10}$ | 0.004 | 1.0 | 1000 | 0.5 | 3.2842 | 2 |
| **0.90** | 0.95 | $10^{-10}$ | 0.004 | 1.0 | 1000 | 0.5 | 3.2994 | 3 |
| **0.95** | 0.95 | $10^{-10}$ | 0.004 | 1.0 | 1000 | 0.5 | 3.3591 | 4 |
| **0.98** | 0.95 | $10^{-10}$ | 0.004 | 1.0 | 1000 | 0.5 | 3.4959 | 5 |
| 0.87 | **0.90** | $10^{-10}$ | 0.004 | 1.0 | 1000 | 0.5 | 3.3335 | 6 |
| 0.87 | **0.95** | $10^{-10}$ | 0.004 | 1.0 | 1000 | 0.5 | 3.2842 | 7 |
| 0.87 | **0.98** | $10^{-10}$ | 0.004 | 1.0 | 1000 | 0.5 | 3.2781 | 8 |
| 0.87 | 0.98 | $10^{-10}$ | 0.004 | 1.0 | 1000 | 0.5 | 3.2781 | 9 |
| 0.87 | 0.98 | $10^{-20}$ | 0.004 | 1.0 | 1000 | 0.5 | 3.2781 | 10 |
| 0.87 | 0.98 | $10^{-10}$ | **0.004** | 1.0 | 1000 | 0.5 | 3.2781 | 11 |
| 0.87 | 0.98 | $10^{-10}$ | **0.008** | 1.0 | 1000 | 0.5 | 3.2862 | 12 |
| 0.87 | 0.98 | $10^{-10}$ | **0.016** | 1.0 | 1000 | 0.5 | 3.3536 | 13 |
| 0.87 | 0.98 | $10^{-10}$ | 0.004 | **0.0** | 1000 | 0.5 | 3.2773 | 14 |
| 0.87 | 0.98 | $10^{-10}$ | 0.004 | **1.0** | 1000 | 0.5 | 3.2781 | 15 |
| 0.87 | 0.98 | $10^{-10}$ | 0.004 | **2.0** | 1000 | 0.5 | 3.2792 | 16 |
| 0.87 | 0.98 | $10^{-10}$ | 0.004 | 1.0 | **500** | 0.5 | 3.2890 | 17 |
| 0.87 | 0.98 | $10^{-10}$ | 0.004 | 1.0 | **1000** | 0.5 | 3.2781 | 18 |
| 0.87 | 0.98 | $10^{-10}$ | 0.004 | 1.0 | **2000** | 0.5 | 3.2726 | 19 |
| 0.87 | 0.98 | $10^{-10}$ | 0.004 | 1.0 | 2000 | **0.0** | 3.2741 | 20 |
| 0.87 | 0.98 | $10^{-10}$ | 0.004 | 1.0 | 2000 | **0.5** | 3.2726 | 21 |
| 0.87 | 0.98 | $10^{-10}$ | 0.004 | 1.0 | 2000 | **1.0** | 3.2901 | 22 |

*Table 13.* Hyperparameter ablation for AdamW on 60M on $8\times$ Chinchilla Data (BF16 weights + BF16 compute)

| $\beta_1$ | $\beta_2$ | $\epsilon$ | $\eta$ | $g_{\text{norm}}$ | warmup | wd | Loss | Link |
|---|---|---|---|---|---|---|---|---|
| 0.87 | 0.98 | $10^{-10}$ | 0.004 | 1.0 | 2000 | 0.5 | 3.2321 | 1 |
| **0.87** | 0.95 | $10^{-10}$ | 0.004 | 1.0 | 1000 | 0.5 | 3.2475 | 2 |
| **0.90** | 0.95 | $10^{-10}$ | 0.004 | 1.0 | 1000 | 0.5 | 3.2641 | 3 |
| **0.95** | 0.95 | $10^{-10}$ | 0.004 | 1.0 | 1000 | 0.5 | 3.3308 | 4 |
| **0.98** | 0.95 | $10^{-10}$ | 0.004 | 1.0 | 1000 | 0.5 | 3.4480 | 5 |
| 0.87 | **0.90** | $10^{-10}$ | 0.004 | 1.0 | 1000 | 0.5 | 3.3071 | 6 |
| 0.87 | **0.95** | $10^{-10}$ | 0.004 | 1.0 | 1000 | 0.5 | 3.2475 | 7 |
| 0.87 | **0.98** | $10^{-10}$ | 0.004 | 1.0 | 1000 | 0.5 | 3.2364 | 8 |
| 0.87 | 0.98 | $10^{-10}$ | 0.004 | 1.0 | 1000 | 0.5 | 3.2364 | 9 |
| 0.87 | 0.98 | $10^{-20}$ | 0.004 | 1.0 | 1000 | 0.5 | 3.2364 | 10 |
| 0.87 | 0.98 | $10^{-10}$ | **0.004** | 1.0 | 1000 | 0.5 | 3.2364 | 11 |
| 0.87 | 0.98 | $10^{-10}$ | **0.008** | 1.0 | 1000 | 0.5 | 3.2397 | 12 |
| 0.87 | 0.98 | $10^{-10}$ | **0.016** | 1.0 | 1000 | 0.5 | 3.2929 | 13 |
| 0.87 | 0.98 | $10^{-10}$ | 0.004 | **0.0** | 1000 | 0.5 | 3.2356 | 14 |
| 0.87 | 0.98 | $10^{-10}$ | 0.004 | **1.0** | 1000 | 0.5 | 3.2364 | 15 |
| 0.87 | 0.98 | $10^{-10}$ | 0.004 | **2.0** | 1000 | 0.5 | 3.2356 | 16 |
| 0.87 | 0.98 | $10^{-10}$ | 0.004 | 1.0 | **500** | 0.5 | 3.2449 | 17 |
| 0.87 | 0.98 | $10^{-10}$ | 0.004 | 1.0 | **1000** | 0.5 | 3.2364 | 18 |
| 0.87 | 0.98 | $10^{-10}$ | 0.004 | 1.0 | **2000** | 0.5 | 3.2321 | 19 |
| 0.87 | 0.98 | $10^{-10}$ | 0.004 | 1.0 | 2000 | **0.0** | 3.2353 | 20 |
| 0.87 | 0.98 | $10^{-10}$ | 0.004 | 1.0 | 2000 | **0.5** | 3.2321 | 21 |
| 0.87 | 0.98 | $10^{-10}$ | 0.004 | 1.0 | 2000 | **1.0** | 3.2545 | 22 |

*Table 14.* Hyperparameter ablation for AdamW on 130M on $1\times$ Chinchilla Data (BF16 weights + BF16 compute)

| $\beta_1$ | $\beta_2$ | $\epsilon$ | $\eta$ | $g_{\text{norm}}$ | warmup | wd | Loss | Link |
|---|---|---|---|---|---|---|---|---|
| 0.95 | 0.98 | $10^{-10}$ | 0.008 | 1.0 | 2000 | 0.5 | 3.1607 | 1 |
| **0.87** | 0.95 | $10^{-10}$ | 0.004 | 1.0 | 1000 | 0.5 | 4.3310 | 2 |
| **0.90** | 0.95 | $10^{-10}$ | 0.004 | 1.0 | 1000 | 0.5 | 3.8910 | 3 |
| **0.95** | 0.95 | $10^{-10}$ | 0.004 | 1.0 | 1000 | 0.5 | 3.2357 | 4 |
| **0.98** | 0.95 | $10^{-10}$ | 0.004 | 1.0 | 1000 | 0.5 | 4.7147 | 5 |
| 0.95 | **0.90** | $10^{-10}$ | 0.004 | 1.0 | 1000 | 0.5 | 3.2923 | 6 |
| 0.95 | **0.95** | $10^{-10}$ | 0.004 | 1.0 | 1000 | 0.5 | 3.2357 | 7 |
| 0.95 | **0.98** | $10^{-10}$ | 0.004 | 1.0 | 1000 | 0.5 | 3.2171 | 8 |
| 0.95 | 0.98 | $10^{-10}$ | 0.004 | 1.0 | 1000 | 0.5 | 3.2171 | 9 |
| 0.95 | 0.98 | $10^{-20}$ | 0.004 | 1.0 | 1000 | 0.5 | 3.2184 | 10 |
| 0.95 | 0.98 | $10^{-10}$ | **0.004** | 1.0 | 1000 | 0.5 | 3.2171 | 11 |
| 0.95 | 0.98 | $10^{-10}$ | **0.008** | 1.0 | 1000 | 0.5 | 3.2019 | 12 |
| 0.95 | 0.98 | $10^{-10}$ | **0.016** | 1.0 | 1000 | 0.5 | 3.3035 | 13 |
| 0.95 | 0.98 | $10^{-10}$ | 0.008 | **0.0** | 1000 | 0.5 | 3.2736 | 14 |
| 0.95 | 0.98 | $10^{-10}$ | 0.008 | **1.0** | 1000 | 0.5 | 3.2019 | 15 |
| 0.95 | 0.98 | $10^{-10}$ | 0.008 | **2.0** | 1000 | 0.5 | 3.2663 | 16 |
| 0.95 | 0.98 | $10^{-10}$ | 0.008 | 1.0 | **500** | 0.5 | 3.2084 | 17 |
| 0.95 | 0.98 | $10^{-10}$ | 0.008 | 1.0 | **1000** | 0.5 | 3.2019 | 18 |
| 0.95 | 0.98 | $10^{-10}$ | 0.008 | 1.0 | **2000** | 0.5 | 3.1607 | 19 |
| 0.95 | 0.98 | $10^{-10}$ | 0.008 | 1.0 | 2000 | **0.0** | 3.1942 | 20 |
| 0.95 | 0.98 | $10^{-10}$ | 0.008 | 1.0 | 2000 | **0.5** | 3.1607 | 21 |
| 0.95 | 0.98 | $10^{-10}$ | 0.008 | 1.0 | 2000 | **1.0** | 3.2797 | 22 |

*Table 15.* Hyperparameter ablation for AdamW on 350M on $1\times$ Chinchilla Data (BF16 weights + BF16 compute)

| $\beta_1$ | $\beta_2$ | $\epsilon$ | $\eta$ | $g_{norm}$ | warmup | wd | Loss | Link |
|---|---|---|---|---|---|---|---|---|
| 0.90 | 0.98 | $10^{-10}$ | 0.008 | 1.0 | 1000 | 0.5 | 2.9526 | 1 |
| **0.87** | 0.98 | $10^{-10}$ | 0.008 | 1.0 | 1000 | 0.5 | 2.9746 | 2 |
| **0.90** | 0.98 | $10^{-10}$ | 0.008 | 1.0 | 1000 | 0.5 | 2.9526 | 3 |
| **0.95** | 0.98 | $10^{-10}$ | 0.008 | 1.0 | 1000 | 0.5 | 3.0524 | 4 |
| **0.98** | 0.98 | $10^{-10}$ | 0.008 | 1.0 | 1000 | 0.5 | 3.0516 | 5 |
| 0.90 | **0.90** | $10^{-10}$ | 0.008 | 1.0 | 1000 | 0.5 | 3.0848 | 6 |
| 0.90 | **0.95** | $10^{-10}$ | 0.008 | 1.0 | 1000 | 0.5 | 3.0572 | 7 |
| 0.90 | **0.98** | $10^{-10}$ | 0.008 | 1.0 | 1000 | 0.5 | 2.9526 | 8 |
| 0.90 | 0.98 | $10^{-10}$ | 0.008 | 1.0 | 1000 | 0.5 | 2.9526 | 9 |
| 0.90 | 0.98 | $10^{-20}$ | 0.008 | 1.0 | 1000 | 0.5 | 3.0534 | 10 |
| 0.90 | 0.98 | $10^{-10}$ | **0.004** | 1.0 | 1000 | 0.5 | 6.4219 | 11 |
| 0.90 | 0.98 | $10^{-10}$ | **0.008** | 1.0 | 1000 | 0.5 | 2.9526 | 12 |
| 0.90 | 0.98 | $10^{-10}$ | **0.016** | 1.0 | 1000 | 0.5 | 3.0662 | 13 |
| 0.90 | 0.98 | $10^{-10}$ | 0.008 | **0.0** | 1000 | 0.5 | 3.0574 | 14 |
| 0.90 | 0.98 | $10^{-10}$ | 0.008 | **1.0** | 1000 | 0.5 | 2.9526 | 15 |
| 0.90 | 0.98 | $10^{-10}$ | 0.008 | **2.0** | 1000 | 0.5 | 3.0353 | 16 |
| 0.90 | 0.98 | $10^{-10}$ | 0.008 | 1.0 | **1000** | 0.5 | 2.9526 | 17 |
| 0.90 | 0.98 | $10^{-10}$ | 0.008 | 1.0 | **2000** | 0.5 | 3.0118 | 18 |
| 0.90 | 0.98 | $10^{-10}$ | 0.008 | 1.0 | 1000 | **0.0** | 7.1275 | 19 |
| 0.90 | 0.98 | $10^{-10}$ | 0.008 | 1.0 | 1000 | **0.5** | 2.9526 | 20 |
| 0.90 | 0.98 | $10^{-10}$ | 0.008 | 1.0 | 1000 | **1.0** | 3.0813 | 21 |

## I.2. Sweeping Results for M+Adam (BF16 weights + BF16 compute)

*Table 16.* Hyperparameter ablation for M+Adam on 60M on $1\times$ Chinchilla Data (BF16 weights + BF16 compute)

| $\beta_1$ | $\beta_2$ | $\epsilon$ | $\eta_m$ | $\eta_a$ | $g_{norm}$ | warmup | $\lambda_m$ | $\lambda_e$ | Loss | Link |
|---|---|---|---|---|---|---|---|---|---|---|
| 0.87 | 0.98 | $10^{-20}$ | 0.004 | $8 \times 10^{-5}$ | 1.0 | 1000 | 0.1 | 0.01 | 3.3685 | 1 |
| **0.87** | 0.98 | $10^{-20}$ | 0.004 | $8 \times 10^{-5}$ | 1.0 | 1000 | 0.1 | 0.01 | 3.3685 | 2 |
| **0.90** | 0.98 | $10^{-20}$ | 0.004 | $8 \times 10^{-5}$ | 1.0 | 1000 | 0.1 | 0.01 | 3.3761 | 3 |
| **0.95** | 0.98 | $10^{-20}$ | 0.004 | $8 \times 10^{-5}$ | 1.0 | 1000 | 0.1 | 0.01 | 3.4061 | 4 |
| **0.98** | 0.98 | $10^{-20}$ | 0.004 | $8 \times 10^{-5}$ | 1.0 | 1000 | 0.1 | 0.01 | 3.5602 | 5 |
| 0.87 | **0.90** | $10^{-20}$ | 0.004 | $8 \times 10^{-5}$ | 1.0 | 1000 | 0.1 | 0.01 | 3.4013 | 6 |
| 0.87 | **0.95** | $10^{-20}$ | 0.004 | $8 \times 10^{-5}$ | 1.0 | 1000 | 0.1 | 0.01 | 3.3800 | 7 |
| 0.87 | **0.98** | $10^{-20}$ | 0.004 | $8 \times 10^{-5}$ | 1.0 | 1000 | 0.1 | 0.01 | 3.3685 | 8 |
| 0.87 | 0.98 | $10^{-10}$ | 0.004 | $8 \times 10^{-5}$ | 1.0 | 1000 | 0.1 | 0.01 | 3.3681 | 9 |
| 0.87 | 0.98 | $10^{-20}$ | 0.004 | $8 \times 10^{-5}$ | 1.0 | 1000 | 0.1 | 0.01 | 3.3685 | 10 |
| 0.87 | 0.98 | $10^{-20}$ | **0.004** | $8 \times 10^{-5}$ | 1.0 | 1000 | 0.1 | 0.01 | 3.3685 | 11 |
| 0.87 | 0.98 | $10^{-20}$ | **0.008** | $8 \times 10^{-5}$ | 1.0 | 1000 | 0.1 | 0.01 | 3.3688 | 12 |
| 0.87 | 0.98 | $10^{-20}$ | **0.016** | $8 \times 10^{-5}$ | 1.0 | 1000 | 0.1 | 0.01 | 3.4139 | 13 |
| 0.87 | 0.98 | $10^{-20}$ | 0.004 | $8 \times 10^{-5}$ | 1.0 | 1000 | 0.1 | 0.01 | 3.3685 | 14 |
| 0.87 | 0.98 | $10^{-20}$ | 0.004 | $1.6 \times 10^{-4}$ | 1.0 | 1000 | 0.1 | 0.01 | 3.3683 | 15 |
| 0.87 | 0.98 | $10^{-20}$ | 0.004 | $3.2 \times 10^{-4}$ | 1.0 | 1000 | 0.1 | 0.01 | 3.3686 | 16 |
| 0.87 | 0.98 | $10^{-20}$ | 0.004 | $8 \times 10^{-5}$ | 1.0 | **500** | 0.1 | 0.01 | 3.3765 | 17 |
| 0.87 | 0.98 | $10^{-20}$ | 0.004 | $8 \times 10^{-5}$ | 1.0 | **1000** | 0.1 | 0.01 | 3.3685 | 18 |
| 0.87 | 0.98 | $10^{-20}$ | 0.004 | $8 \times 10^{-5}$ | 1.0 | **2000** | 0.1 | 0.01 | 3.3666 | 19 |
| 0.87 | 0.98 | $10^{-20}$ | 0.004 | $8 \times 10^{-5}$ | 1.0 | 1000 | **0.0** | 0.01 | 3.4083 | 20 |
| 0.87 | 0.98 | $10^{-20}$ | 0.004 | $8 \times 10^{-5}$ | 1.0 | 1000 | **0.1** | 0.01 | 3.3685 | 21 |
| 0.87 | 0.98 | $10^{-20}$ | 0.004 | $8 \times 10^{-5}$ | 1.0 | 1000 | **0.2** | 0.01 | 3.3755 | 22 |
| 0.87 | 0.98 | $10^{-20}$ | 0.004 | $8 \times 10^{-5}$ | 1.0 | 1000 | 0.1 | **0.0** | 3.3699 | 23 |
| 0.87 | 0.98 | $10^{-20}$ | 0.004 | $8 \times 10^{-5}$ | 1.0 | 1000 | 0.1 | **0.01** | 3.3685 | 24 |
| 0.87 | 0.98 | $10^{-20}$ | 0.004 | $8 \times 10^{-5}$ | 1.0 | 1000 | 0.1 | **0.02** | 3.3685 | 25 |

*Table 17.* Hyperparameter ablation for M+Adam on 60M on $2\times$ Chinchilla Data (BF16 weights + BF16 compute)

| $\beta_1$ | $\beta_2$ | $\epsilon$ | $\eta_m$ | $\eta_a$ | $g_{\text{norm}}$ | warmup | $\lambda_m$ | $\lambda_e$ | Loss | Link |
|---|---|---|---|---|---|---|---|---|---|---|
| 0.87 | 0.98 | $10^{-10}$ | 0.004 | $8 \times 10^{-5}$ | 1.0 | 1000 | 0.1 | 0.01 | 3.2878 | 1 |
| **0.87** | 0.95 | $10^{-10}$ | 0.004 | $8 \times 10^{-5}$ | 1.0 | 1000 | 0.1 | 0.01 | 3.3009 | 2 |
| **0.90** | 0.95 | $10^{-10}$ | 0.004 | $8 \times 10^{-5}$ | 1.0 | 1000 | 0.1 | 0.01 | 3.3060 | 3 |
| **0.95** | 0.95 | $10^{-10}$ | 0.004 | $8 \times 10^{-5}$ | 1.0 | 1000 | 0.1 | 0.01 | 3.3516 | 4 |
| **0.98** | 0.95 | $10^{-10}$ | 0.004 | $8 \times 10^{-5}$ | 1.0 | 1000 | 0.1 | 0.01 | 3.5857 | 5 |
| 0.87 | **0.90** | $10^{-10}$ | 0.004 | $8 \times 10^{-5}$ | 1.0 | 1000 | 0.1 | 0.01 | 3.3254 | 6 |
| 0.87 | **0.95** | $10^{-10}$ | 0.004 | $8 \times 10^{-5}$ | 1.0 | 1000 | 0.1 | 0.01 | 3.3009 | 7 |
| 0.87 | **0.98** | $10^{-10}$ | 0.004 | $8 \times 10^{-5}$ | 1.0 | 1000 | 0.1 | 0.01 | 3.2878 | 8 |
| 0.87 | 0.98 | $10^{-10}$ | 0.004 | $8 \times 10^{-5}$ | 1.0 | 1000 | 0.1 | 0.01 | 3.2878 | 9 |
| 0.87 | 0.98 | $10^{-20}$ | 0.004 | $8 \times 10^{-5}$ | 1.0 | 1000 | 0.1 | 0.01 | 3.2870 | 10 |
| 0.87 | 0.98 | $10^{-10}$ | **0.004** | $8 \times 10^{-5}$ | 1.0 | 1000 | 0.1 | 0.01 | 3.2878 | 11 |
| 0.87 | 0.98 | $10^{-10}$ | **0.008** | $8 \times 10^{-5}$ | 1.0 | 1000 | 0.1 | 0.01 | 3.2929 | 12 |
| 0.87 | 0.98 | $10^{-10}$ | **0.016** | $8 \times 10^{-5}$ | 1.0 | 1000 | 0.1 | 0.01 | 3.3414 | 13 |
| 0.87 | 0.98 | $10^{-10}$ | 0.004 | $8 \times 10^{-5}$ | 1.0 | 1000 | 0.1 | 0.01 | 3.2878 | 14 |
| 0.87 | 0.98 | $10^{-10}$ | 0.004 | $1.6 \times 10^{-4}$ | 1.0 | 1000 | 0.1 | 0.01 | 3.2872 | 15 |
| 0.87 | 0.98 | $10^{-10}$ | 0.004 | $3.2 \times 10^{-4}$ | 1.0 | 1000 | 0.1 | 0.01 | 3.2881 | 16 |
| 0.87 | 0.98 | $10^{-10}$ | 0.004 | $8 \times 10^{-5}$ | 1.0 | **500** | 0.1 | 0.01 | 3.2953 | 17 |
| 0.87 | 0.98 | $10^{-10}$ | 0.004 | $8 \times 10^{-5}$ | 1.0 | **1000** | 0.1 | 0.01 | 3.2878 | 18 |
| 0.87 | 0.98 | $10^{-10}$ | 0.004 | $8 \times 10^{-5}$ | 1.0 | **2000** | 0.1 | 0.01 | 3.2867 | 19 |
| 0.87 | 0.98 | $10^{-10}$ | 0.004 | $8 \times 10^{-5}$ | 1.0 | 1000 | **0.0** | 0.01 | 3.3321 | 20 |
| 0.87 | 0.98 | $10^{-10}$ | 0.004 | $8 \times 10^{-5}$ | 1.0 | 1000 | **0.1** | 0.01 | 3.2878 | 21 |
| 0.87 | 0.98 | $10^{-10}$ | 0.004 | $8 \times 10^{-5}$ | 1.0 | 1000 | **0.2** | 0.01 | 3.2997 | 22 |
| 0.87 | 0.98 | $10^{-10}$ | 0.004 | $8 \times 10^{-5}$ | 1.0 | 1000 | 0.1 | **0.0** | 3.2882 | 23 |
| 0.87 | 0.98 | $10^{-10}$ | 0.004 | $8 \times 10^{-5}$ | 1.0 | 1000 | 0.1 | **0.01** | 3.2878 | 24 |
| 0.87 | 0.98 | $10^{-10}$ | 0.004 | $8 \times 10^{-5}$ | 1.0 | 1000 | 0.1 | **0.02** | 3.2877 | 25 |

*Table 18.* Hyperparameter ablation for M+Adam on 60M on $4\times$ Chinchilla Data (BF16 weights + BF16 compute)

| $\beta_1$ | $\beta_2$ | $\epsilon$ | $\eta_m$ | $\eta_a$ | $g_{\text{norm}}$ | warmup | $\lambda_m$ | $\lambda_e$ | Loss | Link |
|---|---|---|---|---|---|---|---|---|---|---|
| 0.87 | 0.98 | $10^{-10}$ | 0.004 | $8 \times 10^{-5}$ | 1.0 | 1000 | 0.1 | 0.01 | 3.2260 | 1 |
| **0.87** | 0.95 | $10^{-10}$ | 0.004 | $8 \times 10^{-5}$ | 1.0 | 1000 | 0.1 | 0.01 | 3.2403 | 2 |
| **0.90** | 0.95 | $10^{-10}$ | 0.004 | $8 \times 10^{-5}$ | 1.0 | 1000 | 0.1 | 0.01 | 3.2482 | 3 |
| **0.95** | 0.95 | $10^{-10}$ | 0.004 | $8 \times 10^{-5}$ | 1.0 | 1000 | 0.1 | 0.01 | 3.2921 | 4 |
| **0.98** | 0.95 | $10^{-10}$ | 0.004 | $8 \times 10^{-5}$ | 1.0 | 1000 | 0.1 | 0.01 | 3.4824 | 5 |
| 0.87 | **0.90** | $10^{-10}$ | 0.004 | $8 \times 10^{-5}$ | 1.0 | 1000 | 0.1 | 0.01 | 3.2697 | 6 |
| 0.87 | **0.95** | $10^{-10}$ | 0.004 | $8 \times 10^{-5}$ | 1.0 | 1000 | 0.1 | 0.01 | 3.2403 | 7 |
| 0.87 | **0.98** | $10^{-10}$ | 0.004 | $8 \times 10^{-5}$ | 1.0 | 1000 | 0.1 | 0.01 | 3.2260 | 8 |
| 0.87 | 0.98 | $10^{-10}$ | 0.004 | $8 \times 10^{-5}$ | 1.0 | 1000 | 0.1 | 0.01 | 3.2260 | 9 |
| 0.87 | 0.98 | $10^{-20}$ | 0.004 | $8 \times 10^{-5}$ | 1.0 | 1000 | 0.1 | 0.01 | 3.2255 | 10 |
| 0.87 | 0.98 | $10^{-10}$ | **0.004** | $8 \times 10^{-5}$ | 1.0 | 1000 | 0.1 | 0.01 | 3.2260 | 11 |
| 0.87 | 0.98 | $10^{-10}$ | **0.008** | $8 \times 10^{-5}$ | 1.0 | 1000 | 0.1 | 0.01 | 3.2327 | 12 |
| 0.87 | 0.98 | $10^{-10}$ | **0.016** | $8 \times 10^{-5}$ | 1.0 | 1000 | 0.1 | 0.01 | 3.2797 | 13 |
| 0.87 | 0.98 | $10^{-10}$ | 0.004 | $8 \times 10^{-5}$ | 1.0 | 1000 | 0.1 | 0.01 | 3.2260 | 14 |
| 0.87 | 0.98 | $10^{-10}$ | 0.004 | $1.6 \times 10^{-4}$ | 1.0 | 1000 | 0.1 | 0.01 | 3.2263 | 15 |
| 0.87 | 0.98 | $10^{-10}$ | 0.004 | $3.2 \times 10^{-4}$ | 1.0 | 1000 | 0.1 | 0.01 | 3.2260 | 16 |
| 0.87 | 0.98 | $10^{-10}$ | 0.004 | $8 \times 10^{-5}$ | 1.0 | **500** | 0.1 | 0.01 | 3.2310 | 17 |
| 0.87 | 0.98 | $10^{-10}$ | 0.004 | $8 \times 10^{-5}$ | 1.0 | **1000** | 0.1 | 0.01 | 3.2260 | 18 |
| 0.87 | 0.98 | $10^{-10}$ | 0.004 | $8 \times 10^{-5}$ | 1.0 | **2000** | 0.1 | 0.01 | 3.2260 | 19 |
| 0.87 | 0.98 | $10^{-10}$ | 0.004 | $8 \times 10^{-5}$ | 1.0 | 1000 | **0.0** | 0.01 | 3.2770 | 20 |
| 0.87 | 0.98 | $10^{-10}$ | 0.004 | $8 \times 10^{-5}$ | 1.0 | 1000 | **0.1** | 0.01 | 3.2260 | 21 |
| 0.87 | 0.98 | $10^{-10}$ | 0.004 | $8 \times 10^{-5}$ | 1.0 | 1000 | **0.2** | 0.01 | 3.2430 | 22 |
| 0.87 | 0.98 | $10^{-10}$ | 0.004 | $8 \times 10^{-5}$ | 1.0 | 1000 | 0.1 | **0.0** | 3.2263 | 23 |
| 0.87 | 0.98 | $10^{-10}$ | 0.004 | $8 \times 10^{-5}$ | 1.0 | 1000 | 0.1 | **0.01** | 3.2260 | 24 |
| 0.87 | 0.98 | $10^{-10}$ | 0.004 | $8 \times 10^{-5}$ | 1.0 | 1000 | 0.1 | **0.02** | 3.2251 | 25 |

*Table 19.* Hyperparameter ablation for M+Adam on 60M on $8\times$ Chinchilla Data (BF16 weights + BF16 compute)

| $\beta_1$ | $\beta_2$ | $\epsilon$ | $\eta_m$ | $\eta_a$ | $g_{\text{norm}}$ | warmup | $\lambda_m$ | $\lambda_e$ | Loss | Link |
|---|---|---|---|---|---|---|---|---|---|---|
| 0.87 | 0.98 | $10^{-10}$ | 0.004 | $8 \times 10^{-5}$ | 1.0 | 1000 | 0.1 | 0.01 | 3.1813 | 1 |
| **0.87** | 0.95 | $10^{-10}$ | 0.004 | $8 \times 10^{-5}$ | 1.0 | 1000 | 0.1 | 0.01 | 3.1947 | 2 |
| **0.90** | 0.95 | $10^{-10}$ | 0.004 | $8 \times 10^{-5}$ | 1.0 | 1000 | 0.1 | 0.01 | 3.2058 | 3 |
| **0.95** | 0.95 | $10^{-10}$ | 0.004 | $8 \times 10^{-5}$ | 1.0 | 1000 | 0.1 | 0.01 | 3.2489 | 4 |
| **0.98** | 0.95 | $10^{-10}$ | 0.004 | $8 \times 10^{-5}$ | 1.0 | 1000 | 0.1 | 0.01 | 3.3928 | 5 |
| 0.87 | **0.90** | $10^{-10}$ | 0.004 | $8 \times 10^{-5}$ | 1.0 | 1000 | 0.1 | 0.01 | 3.2315 | 6 |
| 0.87 | **0.95** | $10^{-10}$ | 0.004 | $8 \times 10^{-5}$ | 1.0 | 1000 | 0.1 | 0.01 | 3.1947 | 7 |
| 0.87 | **0.98** | $10^{-10}$ | 0.004 | $8 \times 10^{-5}$ | 1.0 | 1000 | 0.1 | 0.01 | 3.1813 | 8 |
| 0.87 | 0.98 | $10^{-10}$ | 0.004 | $8 \times 10^{-5}$ | 1.0 | 1000 | 0.1 | 0.01 | 3.1813 | 9 |
| 0.87 | 0.98 | $10^{-20}$ | 0.004 | $8 \times 10^{-5}$ | 1.0 | 1000 | 0.1 | 0.01 | 3.1811 | 10 |
| 0.87 | 0.98 | $10^{-10}$ | **0.004** | $8 \times 10^{-5}$ | 1.0 | 1000 | 0.1 | 0.01 | 3.1813 | 11 |
| 0.87 | 0.98 | $10^{-10}$ | **0.008** | $8 \times 10^{-5}$ | 1.0 | 1000 | 0.1 | 0.01 | 3.1898 | 12 |
| 0.87 | 0.98 | $10^{-10}$ | **0.016** | $8 \times 10^{-5}$ | 1.0 | 1000 | 0.1 | 0.01 | 3.2317 | 13 |
| 0.87 | 0.98 | $10^{-10}$ | 0.004 | $8 \times 10^{-5}$ | 1.0 | 1000 | 0.1 | 0.01 | 3.1813 | 14 |
| 0.87 | 0.98 | $10^{-10}$ | 0.004 | $1.6 \times 10^{-4}$ | 1.0 | 1000 | 0.1 | 0.01 | 3.1817 | 15 |
| 0.87 | 0.98 | $10^{-10}$ | 0.004 | $3.2 \times 10^{-4}$ | 1.0 | 1000 | 0.1 | 0.01 | 3.1803 | 16 |
| 0.87 | 0.98 | $10^{-10}$ | 0.004 | $8 \times 10^{-5}$ | 1.0 | **500** | 0.1 | 0.01 | 3.1834 | 17 |
| 0.87 | 0.98 | $10^{-10}$ | 0.004 | $8 \times 10^{-5}$ | 1.0 | **1000** | 0.1 | 0.01 | 3.1813 | 18 |
| 0.87 | 0.98 | $10^{-10}$ | 0.004 | $8 \times 10^{-5}$ | 1.0 | **2000** | 0.1 | 0.01 | 3.1806 | 19 |
| 0.87 | 0.98 | $10^{-10}$ | 0.004 | $8 \times 10^{-5}$ | 1.0 | 1000 | **0.0** | 0.01 | 3.2383 | 20 |
| 0.87 | 0.98 | $10^{-10}$ | 0.004 | $8 \times 10^{-5}$ | 1.0 | 1000 | **0.1** | 0.01 | 3.1813 | 21 |
| 0.87 | 0.98 | $10^{-10}$ | 0.004 | $8 \times 10^{-5}$ | 1.0 | 1000 | **0.2** | 0.01 | 3.2027 | 22 |
| 0.87 | 0.98 | $10^{-10}$ | 0.004 | $8 \times 10^{-5}$ | 1.0 | 1000 | 0.1 | **0.0** | 3.1815 | 23 |
| 0.87 | 0.98 | $10^{-10}$ | 0.004 | $8 \times 10^{-5}$ | 1.0 | 1000 | 0.1 | **0.01** | 3.1813 | 24 |
| 0.87 | 0.98 | $10^{-10}$ | 0.004 | $8 \times 10^{-5}$ | 1.0 | 1000 | 0.1 | **0.02** | 3.1803 | 25 |

*Table 20.* Hyperparameter ablation for M+Adam on 130M on $1\times$ Chinchilla Data (BF16 weights + BF16 compute)

| $\beta_1$ | $\beta_2$ | $\epsilon$ | $\eta_m$ | $\eta_a$ | $g_{\text{norm}}$ | warmup | $\lambda_m$ | $\lambda_e$ | Loss | Link |
|---|---|---|---|---|---|---|---|---|---|---|
| 0.87 | 0.98 | $10^{-20}$ | 0.004 | $3.2 \times 10^{-4}$ | 1.0 | 2000 | 0.1 | 0.0 | 3.0925 | 1 |
| **0.87** | 0.98 | $10^{-20}$ | 0.004 | $8 \times 10^{-5}$ | 1.0 | 1000 | 0.1 | 0.01 | 3.1358 | 2 |
| **0.90** | 0.98 | $10^{-20}$ | 0.004 | $8 \times 10^{-5}$ | 1.0 | 1000 | 0.1 | 0.01 | 3.1711 | 3 |
| **0.95** | 0.98 | $10^{-20}$ | 0.004 | $8 \times 10^{-5}$ | 1.0 | 1000 | 0.1 | 0.01 | 3.1592 | 4 |
| **0.98** | 0.98 | $10^{-20}$ | 0.004 | $8 \times 10^{-5}$ | 1.0 | 1000 | 0.1 | 0.01 | 4.0847 | 5 |
| 0.87 | **0.90** | $10^{-20}$ | 0.004 | $8 \times 10^{-5}$ | 1.0 | 1000 | 0.1 | 0.01 | 3.1565 | 6 |
| 0.87 | **0.95** | $10^{-20}$ | 0.004 | $8 \times 10^{-5}$ | 1.0 | 1000 | 0.1 | 0.01 | 3.1465 | 7 |
| 0.87 | **0.98** | $10^{-20}$ | 0.004 | $8 \times 10^{-5}$ | 1.0 | 1000 | 0.1 | 0.01 | 3.1358 | 8 |
| 0.87 | 0.98 | $10^{-20}$ | 0.004 | $8 \times 10^{-5}$ | 1.0 | 1000 | 0.1 | 0.01 | 3.1358 | 9 |
| 0.87 | 0.98 | $10^{-10}$ | 0.004 | $8 \times 10^{-5}$ | 1.0 | 1000 | 0.1 | 0.01 | 3.1381 | 10 |
| 0.87 | 0.98 | $10^{-20}$ | **0.004** | $8 \times 10^{-5}$ | 1.0 | 1000 | 0.1 | 0.01 | 3.1358 | 11 |
| 0.87 | 0.98 | $10^{-20}$ | **0.008** | $8 \times 10^{-5}$ | 1.0 | 1000 | 0.1 | 0.01 | 3.1924 | 12 |
| 0.87 | 0.98 | $10^{-20}$ | **0.016** | $8 \times 10^{-5}$ | 1.0 | 1000 | 0.1 | 0.01 | 7.0499 | 13 |
| 0.87 | 0.98 | $10^{-20}$ | 0.004 | $8 \times 10^{-5}$ | 1.0 | 1000 | 0.1 | 0.01 | 3.1358 | 14 |
| 0.87 | 0.98 | $10^{-20}$ | 0.004 | $1.6 \times 10^{-4}$ | 1.0 | 1000 | 0.1 | 0.01 | 3.1263 | 15 |
| 0.87 | 0.98 | $10^{-20}$ | 0.004 | $3.2 \times 10^{-4}$ | 1.0 | 1000 | 0.1 | 0.01 | 3.1156 | 16 |
| 0.87 | 0.98 | $10^{-20}$ | 0.004 | $3.2 \times 10^{-4}$ | 1.0 | **500** | 0.1 | 0.01 | 3.1470 | 17 |
| 0.87 | 0.98 | $10^{-20}$ | 0.004 | $3.2 \times 10^{-4}$ | 1.0 | **1000** | 0.1 | 0.01 | 3.1156 | 18 |
| 0.87 | 0.98 | $10^{-20}$ | 0.004 | $3.2 \times 10^{-4}$ | 1.0 | **2000** | 0.1 | 0.01 | 3.0932 | 19 |
| 0.87 | 0.98 | $10^{-20}$ | 0.004 | $3.2 \times 10^{-4}$ | 1.0 | 2000 | **0.0** | 0.01 | 3.2527 | 20 |
| 0.87 | 0.98 | $10^{-20}$ | 0.004 | $3.2 \times 10^{-4}$ | 1.0 | 2000 | **0.2** | 0.01 | 3.1023 | 21 |
| 0.87 | 0.98 | $10^{-20}$ | 0.004 | $3.2 \times 10^{-4}$ | 1.0 | 2000 | 0.1 | **0.0** | 3.0925 | 22 |
| 0.87 | 0.98 | $10^{-20}$ | 0.004 | $3.2 \times 10^{-4}$ | 1.0 | 2000 | 0.1 | **0.01** | 3.0932 | 23 |
| 0.87 | 0.98 | $10^{-20}$ | 0.004 | $3.2 \times 10^{-4}$ | 1.0 | 2000 | 0.1 | **0.02** | 3.0929 | 24 |

*Table 21.* Hyperparameter ablation for M+Adam on 350M on $1\times$ Chinchilla Data (BF16 weights + BF16 compute)

| $\beta_1$ | $\beta_2$ | $\epsilon$ | $\eta_m$ | $\eta_a$ | $g_{\text{norm}}$ | warmup | $\lambda_m$ | $\lambda_e$ | Loss | Link |
|---|---|---|---|---|---|---|---|---|---|---|
| 0.87 | 0.98 | $10^{-20}$ | 0.004 | $3.2 \times 10^{-4}$ | 1.0 | 2000 | 0.1 | 0.0 | 2.8098 | 1 |
| **0.87** | 0.98 | $10^{-20}$ | 0.004 | $8 \times 10^{-5}$ | 1.0 | 1000 | 0.1 | 0.01 | 2.9334 | 2 |
| **0.90** | 0.98 | $10^{-20}$ | 0.004 | $8 \times 10^{-5}$ | 1.0 | 1000 | 0.1 | 0.01 | 2.9703 | 3 |
| **0.95** | 0.98 | $10^{-20}$ | 0.004 | $8 \times 10^{-5}$ | 1.0 | 1000 | 0.1 | 0.01 | 4.2758 | 4 |
| **0.98** | 0.98 | $10^{-20}$ | 0.004 | $8 \times 10^{-5}$ | 1.0 | 1000 | 0.1 | 0.01 | 4.4370 | 5 |
| 0.87 | **0.90** | $10^{-20}$ | 0.004 | $8 \times 10^{-5}$ | 1.0 | 1000 | 0.1 | 0.01 | 3.0165 | 6 |
| 0.87 | **0.95** | $10^{-20}$ | 0.004 | $8 \times 10^{-5}$ | 1.0 | 1000 | 0.1 | 0.01 | 2.9435 | 7 |
| 0.87 | **0.98** | $10^{-20}$ | 0.004 | $8 \times 10^{-5}$ | 1.0 | 1000 | 0.1 | 0.01 | 2.9334 | 8 |
| 0.87 | 0.98 | $10^{-20}$ | 0.004 | $8 \times 10^{-5}$ | 1.0 | 1000 | 0.1 | 0.01 | 2.9334 | 9 |
| 0.87 | 0.98 | $10^{-10}$ | 0.004 | $8 \times 10^{-5}$ | 1.0 | 1000 | 0.1 | 0.01 | 2.9480 | 10 |
| 0.87 | 0.98 | $10^{-20}$ | **0.004** | $8 \times 10^{-5}$ | 1.0 | 1000 | 0.1 | 0.01 | 2.9334 | 11 |
| 0.87 | 0.98 | $10^{-20}$ | **0.008** | $8 \times 10^{-5}$ | 1.0 | 1000 | 0.1 | 0.01 | 6.2851 | 12 |
| 0.87 | 0.98 | $10^{-20}$ | **0.016** | $8 \times 10^{-5}$ | 1.0 | 1000 | 0.1 | 0.01 | 6.5733 | 13 |
| 0.87 | 0.98 | $10^{-20}$ | 0.004 | $8 \times 10^{-5}$ | 1.0 | 1000 | 0.1 | 0.01 | 2.9334 | 14 |
| 0.87 | 0.98 | $10^{-20}$ | 0.004 | $1.6 \times 10^{-4}$ | 1.0 | 1000 | 0.1 | 0.01 | 2.9693 | 15 |
| 0.87 | 0.98 | $10^{-20}$ | 0.004 | $3.2 \times 10^{-4}$ | 1.0 | 1000 | 0.1 | 0.01 | 2.9100 | 16 |
| 0.87 | 0.98 | $10^{-20}$ | 0.004 | $3.2 \times 10^{-4}$ | 1.0 | **1000** | 0.1 | 0.01 | 2.9100 | 17 |
| 0.87 | 0.98 | $10^{-20}$ | 0.004 | $3.2 \times 10^{-4}$ | 1.0 | **2000** | 0.1 | 0.01 | 2.8184 | 18 |
| 0.87 | 0.98 | $10^{-20}$ | 0.004 | $3.2 \times 10^{-4}$ | 1.0 | 2000 | **0.0** | 0.01 | 7.0109 | 19 |
| 0.87 | 0.98 | $10^{-20}$ | 0.004 | $3.2 \times 10^{-4}$ | 1.0 | 2000 | **0.1** | 0.01 | 2.8184 | 20 |
| 0.87 | 0.98 | $10^{-20}$ | 0.004 | $3.2 \times 10^{-4}$ | 1.0 | 2000 | **0.2** | 0.01 | 2.8297 | 21 |
| 0.87 | 0.98 | $10^{-20}$ | 0.004 | $3.2 \times 10^{-4}$ | 1.0 | 2000 | 0.1 | **0.0** | 2.8098 | 22 |
| 0.87 | 0.98 | $10^{-20}$ | 0.004 | $3.2 \times 10^{-4}$ | 1.0 | 2000 | 0.1 | **0.01** | 2.8184 | 23 |
| 0.87 | 0.98 | $10^{-20}$ | 0.004 | $3.2 \times 10^{-4}$ | 1.0 | 2000 | 0.1 | **0.02** | 2.8152 | 24 |

## I.3. Sweeping Results for AdamW (BF16 weights + FP8 compute)

*Table 22.* Hyperparameter ablation for AdamW on 60M on $1\times$ Chinchilla Data (BF16 weights + FP8 compute)

| $\beta_1$ | $\beta_2$ | $\epsilon$ | $\eta$ | $g_{\text{norm}}$ | warmup | wd | Loss | Link |
|---|---|---|---|---|---|---|---|---|
| 0.87 | 0.98 | $10^{-10}$ | 0.004 | 1.0 | 2000 | 0.5 | 3.4062 | 1 |
| **0.87** | 0.95 | $10^{-10}$ | 0.004 | 1.0 | 1000 | 0.5 | 3.4184 | 2 |
| **0.90** | 0.95 | $10^{-10}$ | 0.004 | 1.0 | 1000 | 0.5 | 3.4278 | 3 |
| **0.95** | 0.95 | $10^{-10}$ | 0.004 | 1.0 | 1000 | 0.5 | 3.4695 | 4 |
| **0.98** | 0.95 | $10^{-10}$ | 0.004 | 1.0 | 1000 | 0.5 | 3.7007 | 5 |
| 0.87 | **0.90** | $10^{-10}$ | 0.004 | 1.0 | 1000 | 0.5 | 3.4432 | 6 |
| 0.87 | **0.95** | $10^{-10}$ | 0.004 | 1.0 | 1000 | 0.5 | 3.4184 | 7 |
| 0.87 | **0.98** | $10^{-10}$ | 0.004 | 1.0 | 1000 | 0.5 | 3.4142 | 8 |
| 0.87 | 0.98 | $10^{-10}$ | 0.004 | 1.0 | 1000 | 0.5 | 3.4142 | 9 |
| 0.87 | 0.98 | $10^{-20}$ | 0.004 | 1.0 | 1000 | 0.5 | 3.4114 | 10 |
| 0.87 | 0.98 | $10^{-10}$ | **0.004** | 1.0 | 1000 | 0.5 | 3.4142 | 11 |
| 0.87 | 0.98 | $10^{-10}$ | **0.008** | 1.0 | 1000 | 0.5 | 3.4294 | 12 |
| 0.87 | 0.98 | $10^{-10}$ | **0.016** | 1.0 | 1000 | 0.5 | 3.5628 | 13 |
| 0.87 | 0.98 | $10^{-10}$ | 0.004 | **0.0** | 1000 | 0.5 | 3.4124 | 14 |
| 0.87 | 0.98 | $10^{-10}$ | 0.004 | **1.0** | 1000 | 0.5 | 3.4142 | 15 |
| 0.87 | 0.98 | $10^{-10}$ | 0.004 | **2.0** | 1000 | 0.5 | 3.4134 | 16 |
| 0.87 | 0.98 | $10^{-10}$ | 0.004 | 1.0 | **500** | 0.5 | 3.4303 | 17 |
| 0.87 | 0.98 | $10^{-10}$ | 0.004 | 1.0 | **1000** | 0.5 | 3.4142 | 18 |
| 0.87 | 0.98 | $10^{-10}$ | 0.004 | 1.0 | **2000** | 0.5 | 3.4062 | 19 |
| 0.87 | 0.98 | $10^{-10}$ | 0.004 | 1.0 | 2000 | **0.0** | 3.4075 | 20 |
| 0.87 | 0.98 | $10^{-10}$ | 0.004 | 1.0 | 2000 | **0.5** | 3.4062 | 21 |
| 0.87 | 0.98 | $10^{-10}$ | 0.004 | 1.0 | 2000 | **1.0** | 3.4637 | 22 |

*Table 23.* Hyperparameter ablation for AdamW on 60M on $2\times$ Chinchilla Data (BF16 weights + FP8 compute)

| $\beta_1$ | $\beta_2$ | $\epsilon$ | $\eta$ | $g_{\mathrm{norm}}$ | warmup | wd | Loss | Link |
|---|---|---|---|---|---|---|---|---|
| 0.87 | 0.98 | $10^{-10}$ | 0.004 | 1.0 | 2000 | 0.5 | 3.3359 | 1 |
| **0.87** | 0.95 | $10^{-10}$ | 0.004 | 1.0 | 1000 | 0.5 | 3.3511 | 2 |
| **0.90** | 0.95 | $10^{-10}$ | 0.004 | 1.0 | 1000 | 0.5 | 3.3648 | 3 |
| **0.95** | 0.95 | $10^{-10}$ | 0.004 | 1.0 | 1000 | 0.5 | 3.4084 | 4 |
| **0.98** | 0.95 | $10^{-10}$ | 0.004 | 1.0 | 1000 | 0.5 | 3.5746 | 5 |
| 0.87 | **0.90** | $10^{-10}$ | 0.004 | 1.0 | 1000 | 0.5 | 3.3833 | 6 |
| 0.87 | **0.95** | $10^{-10}$ | 0.004 | 1.0 | 1000 | 0.5 | 3.3511 | 7 |
| 0.87 | **0.98** | $10^{-10}$ | 0.004 | 1.0 | 1000 | 0.5 | 3.3441 | 8 |
| 0.87 | 0.98 | $10^{-10}$ | 0.004 | 1.0 | 1000 | 0.5 | 3.3441 | 9 |
| 0.87 | 0.98 | $10^{-20}$ | 0.004 | 1.0 | 1000 | 0.5 | 3.3415 | 10 |
| 0.87 | 0.98 | $10^{-10}$ | **0.004** | 1.0 | 1000 | 0.5 | 3.3441 | 11 |
| 0.87 | 0.98 | $10^{-10}$ | **0.008** | 1.0 | 1000 | 0.5 | 3.3415 | 12 |
| 0.87 | 0.98 | $10^{-10}$ | **0.016** | 1.0 | 1000 | 0.5 | 3.4447 | 13 |
| 0.87 | 0.98 | $10^{-10}$ | 0.004 | **0.0** | 1000 | 0.5 | 3.3413 | 14 |
| 0.87 | 0.98 | $10^{-10}$ | 0.004 | **1.0** | 1000 | 0.5 | 3.3441 | 15 |
| 0.87 | 0.98 | $10^{-10}$ | 0.004 | **2.0** | 1000 | 0.5 | 3.3427 | 16 |
| 0.87 | 0.98 | $10^{-10}$ | 0.004 | 1.0 | **500** | 0.5 | 3.3557 | 17 |
| 0.87 | 0.98 | $10^{-10}$ | 0.004 | 1.0 | **1000** | 0.5 | 3.3441 | 18 |
| 0.87 | 0.98 | $10^{-10}$ | 0.004 | 1.0 | **2000** | 0.5 | 3.3359 | 19 |
| 0.87 | 0.98 | $10^{-10}$ | 0.004 | 1.0 | 2000 | **0.0** | 3.3369 | 20 |
| 0.87 | 0.98 | $10^{-10}$ | 0.004 | 1.0 | 2000 | **0.5** | 3.3359 | 21 |
| 0.87 | 0.98 | $10^{-10}$ | 0.004 | 1.0 | 2000 | **1.0** | 3.3653 | 22 |

*Table 24.* Hyperparameter ablation for AdamW on 60M on $4\times$ Chinchilla Data (BF16 weights + FP8 compute)

| $\beta_1$ | $\beta_2$ | $\epsilon$ | $\eta$ | $g_{\mathrm{norm}}$ | warmup | wd | Loss | Link |
|---|---|---|---|---|---|---|---|---|
| 0.87 | 0.98 | $10^{-10}$ | 0.008 | 1.0 | 2000 | 0.5 | 3.2714 | 1 |
| **0.87** | 0.95 | $10^{-10}$ | 0.004 | 1.0 | 1000 | 0.5 | 3.3014 | 2 |
| **0.90** | 0.95 | $10^{-10}$ | 0.004 | 1.0 | 1000 | 0.5 | 3.3197 | 3 |
| **0.95** | 0.95 | $10^{-10}$ | 0.004 | 1.0 | 1000 | 0.5 | 3.3730 | 4 |
| **0.98** | 0.95 | $10^{-10}$ | 0.004 | 1.0 | 1000 | 0.5 | 3.4989 | 5 |
| 0.87 | **0.90** | $10^{-10}$ | 0.004 | 1.0 | 1000 | 0.5 | 3.3438 | 6 |
| 0.87 | **0.95** | $10^{-10}$ | 0.004 | 1.0 | 1000 | 0.5 | 3.3014 | 7 |
| 0.87 | **0.98** | $10^{-10}$ | 0.004 | 1.0 | 1000 | 0.5 | 3.2902 | 8 |
| 0.87 | 0.98 | $10^{-10}$ | 0.004 | 1.0 | 1000 | 0.5 | 3.2902 | 9 |
| 0.87 | 0.98 | $10^{-20}$ | 0.004 | 1.0 | 1000 | 0.5 | 3.2884 | 10 |
| 0.87 | 0.98 | $10^{-10}$ | **0.004** | 1.0 | 1000 | 0.5 | 3.2902 | 11 |
| 0.87 | 0.98 | $10^{-10}$ | **0.008** | 1.0 | 1000 | 0.5 | 3.2816 | 12 |
| 0.87 | 0.98 | $10^{-10}$ | **0.016** | 1.0 | 1000 | 0.5 | 3.3514 | 13 |
| 0.87 | 0.98 | $10^{-10}$ | 0.008 | **0.0** | 1000 | 0.5 | 3.2856 | 14 |
| 0.87 | 0.98 | $10^{-10}$ | 0.008 | **1.0** | 1000 | 0.5 | 3.2816 | 15 |
| 0.87 | 0.98 | $10^{-10}$ | 0.008 | **2.0** | 1000 | 0.5 | 3.2967 | 16 |
| 0.87 | 0.98 | $10^{-10}$ | 0.008 | 1.0 | **500** | 0.5 | 3.2785 | 17 |
| 0.87 | 0.98 | $10^{-10}$ | 0.008 | 1.0 | **1000** | 0.5 | 3.2816 | 18 |
| 0.87 | 0.98 | $10^{-10}$ | 0.008 | 1.0 | **2000** | 0.5 | 3.2714 | 19 |
| 0.87 | 0.98 | $10^{-10}$ | 0.008 | 1.0 | 2000 | **0.0** | 3.2754 | 20 |
| 0.87 | 0.98 | $10^{-10}$ | 0.008 | 1.0 | 2000 | **0.5** | 3.2714 | 21 |
| 0.87 | 0.98 | $10^{-10}$ | 0.008 | 1.0 | 2000 | **1.0** | 3.3896 | 22 |

*Table 25.* Hyperparameter ablation for AdamW on 60M on $8\times$ Chinchilla Data (BF16 weights + FP8 compute)

| $\beta_1$ | $\beta_2$ | $\epsilon$ | $\eta$ | $g_{\text{norm}}$ | warmup | wd | Loss | Link |
|---|---|---|---|---|---|---|---|---|
| 0.87 | 0.98 | $10^{-10}$ | 0.008 | 0.0 | 1000 | 0.5 | 3.2350 | 1 |
| **0.87** | 0.95 | $10^{-10}$ | 0.004 | 1.0 | 1000 | 0.5 | 3.2664 | 2 |
| **0.90** | 0.95 | $10^{-10}$ | 0.004 | 1.0 | 1000 | 0.5 | 3.2886 | 3 |
| **0.95** | 0.95 | $10^{-10}$ | 0.004 | 1.0 | 1000 | 0.5 | 3.3466 | 4 |
| **0.98** | 0.95 | $10^{-10}$ | 0.004 | 1.0 | 1000 | 0.5 | 3.4553 | 5 |
| 0.87 | **0.90** | $10^{-10}$ | 0.004 | 1.0 | 1000 | 0.5 | 3.3219 | 6 |
| 0.87 | **0.95** | $10^{-10}$ | 0.004 | 1.0 | 1000 | 0.5 | 3.2664 | 7 |
| 0.87 | **0.98** | $10^{-10}$ | 0.004 | 1.0 | 1000 | 0.5 | 3.2470 | 8 |
| 0.87 | 0.98 | $10^{-10}$ | 0.004 | 1.0 | 1000 | 0.5 | 3.2470 | 9 |
| 0.87 | 0.98 | $10^{-20}$ | 0.004 | 1.0 | 1000 | 0.5 | 3.2455 | 10 |
| 0.87 | 0.98 | $10^{-10}$ | **0.004** | 1.0 | 1000 | 0.5 | 3.2470 | 11 |
| 0.87 | 0.98 | $10^{-10}$ | **0.008** | 1.0 | 1000 | 0.5 | 3.2404 | 12 |
| 0.87 | 0.98 | $10^{-10}$ | **0.016** | 1.0 | 1000 | 0.5 | 3.2883 | 13 |
| 0.87 | 0.98 | $10^{-10}$ | 0.008 | **0.0** | 1000 | 0.5 | 3.2350 | 14 |
| 0.87 | 0.98 | $10^{-10}$ | 0.008 | **1.0** | 1000 | 0.5 | 3.2404 | 15 |
| 0.87 | 0.98 | $10^{-10}$ | 0.008 | **2.0** | 1000 | 0.5 | 3.2427 | 16 |
| 0.87 | 0.98 | $10^{-10}$ | 0.008 | 0.0 | **500** | 0.5 | 3.2361 | 17 |
| 0.87 | 0.98 | $10^{-10}$ | 0.008 | 0.0 | **1000** | 0.5 | 3.2350 | 18 |
| 0.87 | 0.98 | $10^{-10}$ | 0.008 | 0.0 | **2000** | 0.5 | 3.2368 | 19 |
| 0.87 | 0.98 | $10^{-10}$ | 0.008 | 0.0 | 1000 | **0.0** | 3.2698 | 20 |
| 0.87 | 0.98 | $10^{-10}$ | 0.008 | 0.0 | 1000 | **0.5** | 3.2350 | 21 |
| 0.87 | 0.98 | $10^{-10}$ | 0.008 | 0.0 | 1000 | **1.0** | 3.3125 | 22 |

*Table 26.* Hyperparameter ablation for AdamW on 130M on $1\times$ Chinchilla Data (BF16 weights + FP8 compute)

| $\beta_1$ | $\beta_2$ | $\epsilon$ | $\eta$ | $g_{\text{norm}}$ | warmup | wd | Loss | Link |
|---|---|---|---|---|---|---|---|---|
| 0.95 | 0.98 | $10^{-10}$ | 0.008 | 1.0 | 2000 | 0.5 | 3.1610 | 1 |
| **0.87** | 0.95 | $10^{-10}$ | 0.004 | 1.0 | 1000 | 0.5 | 4.3639 | 2 |
| **0.90** | 0.95 | $10^{-10}$ | 0.004 | 1.0 | 1000 | 0.5 | 4.1246 | 3 |
| **0.95** | 0.95 | $10^{-10}$ | 0.004 | 1.0 | 1000 | 0.5 | 3.2482 | 4 |
| **0.98** | 0.95 | $10^{-10}$ | 0.004 | 1.0 | 1000 | 0.5 | 4.5928 | 5 |
| 0.95 | **0.90** | $10^{-10}$ | 0.004 | 1.0 | 1000 | 0.5 | 3.3018 | 6 |
| 0.95 | **0.95** | $10^{-10}$ | 0.004 | 1.0 | 1000 | 0.5 | 3.2482 | 7 |
| 0.95 | **0.98** | $10^{-10}$ | 0.004 | 1.0 | 1000 | 0.5 | 3.2254 | 8 |
| 0.95 | 0.98 | $10^{-10}$ | 0.004 | 1.0 | 1000 | 0.5 | 3.2254 | 9 |
| 0.95 | 0.98 | $10^{-20}$ | 0.004 | 1.0 | 1000 | 0.5 | 3.2348 | 10 |
| 0.95 | 0.98 | $10^{-10}$ | **0.004** | 1.0 | 1000 | 0.5 | 3.2254 | 11 |
| 0.95 | 0.98 | $10^{-10}$ | **0.008** | 1.0 | 1000 | 0.5 | 3.2073 | 12 |
| 0.95 | 0.98 | $10^{-10}$ | **0.016** | 1.0 | 1000 | 0.5 | 3.3176 | 13 |
| 0.95 | 0.98 | $10^{-10}$ | 0.008 | **0.0** | 1000 | 0.5 | 3.2335 | 14 |
| 0.95 | 0.98 | $10^{-10}$ | 0.008 | **1.0** | 1000 | 0.5 | 3.2073 | 15 |
| 0.95 | 0.98 | $10^{-10}$ | 0.008 | **2.0** | 1000 | 0.5 | 3.2213 | 16 |
| 0.95 | 0.98 | $10^{-10}$ | 0.008 | 1.0 | **500** | 0.5 | 3.2026 | 17 |
| 0.95 | 0.98 | $10^{-10}$ | 0.008 | 1.0 | **1000** | 0.5 | 3.2073 | 18 |
| 0.95 | 0.98 | $10^{-10}$ | 0.008 | 1.0 | **2000** | 0.5 | 3.1610 | 19 |
| 0.95 | 0.98 | $10^{-10}$ | 0.008 | 1.0 | 2000 | **0.0** | 3.2002 | 20 |
| 0.95 | 0.98 | $10^{-10}$ | 0.008 | 1.0 | 2000 | **0.5** | 3.1610 | 21 |
| 0.95 | 0.98 | $10^{-10}$ | 0.008 | 1.0 | 2000 | **1.0** | 3.2762 | 22 |

*Table 27.* Hyperparameter ablation for AdamW on 350M on $1\times$ Chinchilla Data (BF16 weights + FP8 compute)

| $\beta_1$ | $\beta_2$ | $\epsilon$ | $\eta$ | $g_{\text{norm}}$ | warmup | wd | Loss | Link |
|---|---|---|---|---|---|---|---|---|
| 0.87 | 0.95 | $10^{-10}$ | 0.008 | 1.0 | 1000 | 0.5 | 2.9567 | 1 |
| 0.87 | 0.95 | $10^{-10}$ | 0.008 | 1.0 | 1000 | 0.5 | 2.9567 | 2 |
| **0.87** | 0.95 | $10^{-10}$ | 0.008 | 1.0 | 1000 | 0.5 | 2.9567 | 3 |
| **0.90** | 0.95 | $10^{-10}$ | 0.008 | 1.0 | 1000 | 0.5 | 3.0399 | 4 |
| **0.95** | 0.95 | $10^{-10}$ | 0.008 | 1.0 | 1000 | 0.5 | 3.0124 | 5 |
| **0.98** | 0.95 | $10^{-10}$ | 0.008 | 1.0 | 1000 | 0.5 | 3.0301 | 6 |
| 0.87 | **0.90** | $10^{-10}$ | 0.008 | 1.0 | 1000 | 0.5 | 3.0083 | 7 |
| 0.87 | **0.95** | $10^{-10}$ | 0.008 | 1.0 | 1000 | 0.5 | 2.9567 | 8 |
| 0.87 | **0.98** | $10^{-10}$ | 0.008 | 1.0 | 1000 | 0.5 | 2.9732 | 9 |
| 0.87 | 0.95 | $10^{-10}$ | 0.008 | 1.0 | 1000 | 0.5 | 2.9567 | 10 |
| 0.87 | 0.95 | $10^{-20}$ | 0.008 | 1.0 | 1000 | 0.5 | 3.0040 | 11 |
| 0.87 | 0.95 | $10^{-10}$ | **0.004** | 1.0 | 1000 | 0.5 | 6.4934 | 12 |
| 0.87 | 0.95 | $10^{-10}$ | **0.008** | 1.0 | 1000 | 0.5 | 2.9567 | 13 |
| 0.87 | 0.95 | $10^{-10}$ | **0.016** | 1.0 | 1000 | 0.5 | 3.0674 | 14 |
| 0.87 | 0.95 | $10^{-10}$ | 0.008 | **0.0** | 1000 | 0.5 | 3.0391 | 15 |
| 0.87 | 0.95 | $10^{-10}$ | 0.008 | **1.0** | 1000 | 0.5 | 2.9567 | 16 |
| 0.87 | 0.95 | $10^{-10}$ | 0.008 | **2.0** | 1000 | 0.5 | 3.0270 | 17 |
| 0.87 | 0.95 | $10^{-10}$ | 0.008 | 1.0 | **1000** | 0.5 | 2.9567 | 18 |
| 0.87 | 0.95 | $10^{-10}$ | 0.008 | 1.0 | **2000** | 0.5 | 2.9870 | 19 |
| 0.87 | 0.95 | $10^{-10}$ | 0.008 | 1.0 | 1000 | **0.0** | 6.7490 | 20 |
| 0.87 | 0.95 | $10^{-10}$ | 0.008 | 1.0 | 1000 | **0.5** | 2.9567 | 21 |
| 0.87 | 0.95 | $10^{-10}$ | 0.008 | 1.0 | 1000 | **1.0** | 3.0470 | 22 |

## I.4. Sweeping Results for M+Adam (BF16 weights + FP8 compute)

*Table 28.* Hyperparameter ablation for M+Adam on 60M on $1\times$ Chinchilla Data (BF16 weights + FP8 compute)

| $\beta_1$ | $\beta_2$ | $\epsilon$ | $\eta_m$ | $\eta_a$ | warmup | $\lambda_m$ | $\lambda_e$ | Loss | Link |
|---|---|---|---|---|---|---|---|---|---|
| 0.87 | 0.98 | $10^{-10}$ | 0.004 | $8 \times 10^{-5}$ | 1000 | 0.1 | 0.01 | 3.3742 | 1 |
| **0.87** | 0.98 | $10^{-20}$ | 0.004 | $8 \times 10^{-5}$ | 1000 | 0.1 | 0.01 | 3.3774 | 2 |
| **0.90** | 0.98 | $10^{-20}$ | 0.004 | $8 \times 10^{-5}$ | 1000 | 0.1 | 0.01 | 3.3834 | 3 |
| **0.95** | 0.98 | $10^{-20}$ | 0.004 | $8 \times 10^{-5}$ | 1000 | 0.1 | 0.01 | 3.4155 | 4 |
| **0.98** | 0.98 | $10^{-20}$ | 0.004 | $8 \times 10^{-5}$ | 1000 | 0.1 | 0.01 | 3.5928 | 5 |
| 0.87 | **0.90** | $10^{-20}$ | 0.004 | $8 \times 10^{-5}$ | 1000 | 0.1 | 0.01 | 3.4089 | 6 |
| 0.87 | **0.95** | $10^{-20}$ | 0.004 | $8 \times 10^{-5}$ | 1000 | 0.1 | 0.01 | 3.3876 | 7 |
| 0.87 | **0.98** | $10^{-20}$ | 0.004 | $8 \times 10^{-5}$ | 1000 | 0.1 | 0.01 | 3.3774 | 8 |
| 0.87 | 0.98 | $10^{-20}$ | 0.004 | $8 \times 10^{-5}$ | 1000 | 0.1 | 0.01 | 3.3774 | 9 |
| 0.87 | 0.98 | $10^{-10}$ | 0.004 | $8 \times 10^{-5}$ | 1000 | 0.1 | 0.01 | 3.3742 | 10 |
| 0.87 | 0.98 | $10^{-10}$ | **0.004** | $8 \times 10^{-5}$ | 1000 | 0.1 | 0.01 | 3.3742 | 11 |
| 0.87 | 0.98 | $10^{-10}$ | **0.008** | $8 \times 10^{-5}$ | 1000 | 0.1 | 0.01 | 3.3761 | 12 |
| 0.87 | 0.98 | $10^{-10}$ | **0.016** | $8 \times 10^{-5}$ | 1000 | 0.1 | 0.01 | 3.4254 | 13 |
| 0.87 | 0.98 | $10^{-10}$ | 0.004 | $8 \times 10^{-5}$ | 1000 | 0.1 | 0.01 | 3.3742 | 14 |
| 0.87 | 0.98 | $10^{-10}$ | 0.004 | $1.6 \times 10^{-4}$ | 1000 | 0.1 | 0.01 | 3.3772 | 15 |
| 0.87 | 0.98 | $10^{-10}$ | 0.004 | $3.2 \times 10^{-4}$ | 1000 | 0.1 | 0.01 | 3.3750 | 16 |
| 0.87 | 0.98 | $10^{-10}$ | 0.004 | $8 \times 10^{-5}$ | **500** | 0.1 | 0.01 | 3.3877 | 17 |
| 0.87 | 0.98 | $10^{-10}$ | 0.004 | $8 \times 10^{-5}$ | **1000** | 0.1 | 0.01 | 3.3742 | 18 |
| 0.87 | 0.98 | $10^{-10}$ | 0.004 | $8 \times 10^{-5}$ | **2000** | 0.1 | 0.01 | 3.3732 | 19 |
| 0.87 | 0.98 | $10^{-10}$ | 0.004 | $8 \times 10^{-5}$ | 1000 | **0.0** | 0.01 | 3.4136 | 20 |
| 0.87 | 0.98 | $10^{-10}$ | 0.004 | $8 \times 10^{-5}$ | 1000 | **0.1** | 0.01 | 3.3742 | 21 |
| 0.87 | 0.98 | $10^{-10}$ | 0.004 | $8 \times 10^{-5}$ | 1000 | **0.2** | 0.01 | 3.3785 | 22 |
| 0.87 | 0.98 | $10^{-10}$ | 0.004 | $8 \times 10^{-5}$ | 1000 | 0.1 | **0.0** | 3.3726 | 23 |
| 0.87 | 0.98 | $10^{-10}$ | 0.004 | $8 \times 10^{-5}$ | 1000 | 0.1 | **0.02** | 3.3725 | 24 |

*Table 29.* Hyperparameter ablation for M+Adam on 60M on $2\times$ Chinchilla Data (BF16 weights + FP8 compute)

| $\beta_1$ | $\beta_2$ | $\epsilon$ | $\eta_m$ | $\eta_a$ | warmup | $\lambda_m$ | $\lambda_e$ | Loss | Link |
|---|---|---|---|---|---|---|---|---|---|
| 0.87 | 0.98 | $10^{-10}$ | 0.004 | $8 \times 10^{-5}$ | 1000 | 0.1 | 0.01 | 3.2920 | 1 |
| **0.87** | 0.98 | $10^{-20}$ | 0.004 | $8 \times 10^{-5}$ | 1000 | 0.1 | 0.01 | 3.2961 | 2 |
| **0.90** | 0.98 | $10^{-20}$ | 0.004 | $8 \times 10^{-5}$ | 1000 | 0.1 | 0.01 | 3.3021 | 3 |
| **0.95** | 0.98 | $10^{-20}$ | 0.004 | $8 \times 10^{-5}$ | 1000 | 0.1 | 0.01 | 3.3330 | 4 |
| **0.98** | 0.98 | $10^{-20}$ | 0.004 | $8 \times 10^{-5}$ | 1000 | 0.1 | 0.01 | 3.4688 | 5 |
| 0.87 | **0.90** | $10^{-20}$ | 0.004 | $8 \times 10^{-5}$ | 1000 | 0.1 | 0.01 | 3.3323 | 6 |
| 0.87 | **0.95** | $10^{-20}$ | 0.004 | $8 \times 10^{-5}$ | 1000 | 0.1 | 0.01 | 3.3084 | 7 |
| 0.87 | **0.98** | $10^{-20}$ | 0.004 | $8 \times 10^{-5}$ | 1000 | 0.1 | 0.01 | 3.2961 | 8 |
| 0.87 | 0.98 | $10^{-20}$ | 0.004 | $8 \times 10^{-5}$ | 1000 | 0.1 | 0.01 | 3.2961 | 9 |
| 0.87 | 0.98 | $10^{-10}$ | 0.004 | $8 \times 10^{-5}$ | 1000 | 0.1 | 0.01 | 3.2920 | 10 |
| 0.87 | 0.98 | $10^{-10}$ | **0.004** | $8 \times 10^{-5}$ | 1000 | 0.1 | 0.01 | 3.2920 | 11 |
| 0.87 | 0.98 | $10^{-10}$ | **0.008** | $8 \times 10^{-5}$ | 1000 | 0.1 | 0.01 | 3.2970 | 12 |
| 0.87 | 0.98 | $10^{-10}$ | **0.016** | $8 \times 10^{-5}$ | 1000 | 0.1 | 0.01 | 3.3445 | 13 |
| 0.87 | 0.98 | $10^{-10}$ | 0.004 | $8 \times 10^{-5}$ | 1000 | 0.1 | 0.01 | 3.2920 | 14 |
| 0.87 | 0.98 | $10^{-10}$ | 0.004 | $1.6 \times 10^{-4}$ | 1000 | 0.1 | 0.01 | 3.2963 | 15 |
| 0.87 | 0.98 | $10^{-10}$ | 0.004 | $3.2 \times 10^{-4}$ | 1000 | 0.1 | 0.01 | 3.2943 | 16 |
| 0.87 | 0.98 | $10^{-10}$ | 0.004 | $8 \times 10^{-5}$ | **500** | 0.1 | 0.01 | 3.3035 | 17 |
| 0.87 | 0.98 | $10^{-10}$ | 0.004 | $8 \times 10^{-5}$ | **1000** | 0.1 | 0.01 | 3.2920 | 18 |
| 0.87 | 0.98 | $10^{-10}$ | 0.004 | $8 \times 10^{-5}$ | **2000** | 0.1 | 0.01 | 3.2930 | 19 |
| 0.87 | 0.98 | $10^{-10}$ | 0.004 | $8 \times 10^{-5}$ | 1000 | **0.0** | 0.01 | 3.3403 | 20 |
| 0.87 | 0.98 | $10^{-10}$ | 0.004 | $8 \times 10^{-5}$ | 1000 | **0.2** | 0.01 | 3.3014 | 21 |
| 0.87 | 0.98 | $10^{-10}$ | 0.004 | $8 \times 10^{-5}$ | 1000 | 0.1 | **0.0** | 3.2916 | 22 |
| 0.87 | 0.98 | $10^{-10}$ | 0.004 | $8 \times 10^{-5}$ | 1000 | 0.1 | **0.02** | 3.2925 | 23 |

*Table 30.* Hyperparameter ablation for M+Adam on 60M on $4\times$ Chinchilla Data (BF16 weights + FP8 compute)

| $\beta_1$ | $\beta_2$ | $\epsilon$ | $\eta_m$ | $\eta_a$ | warmup | $\lambda_m$ | $\lambda_e$ | Loss | Link |
|---|---|---|---|---|---|---|---|---|---|
| 0.87 | 0.98 | $10^{-20}$ | 0.004 | $8 \times 10^{-5}$ | 1000 | 0.1 | 0.01 | 3.2336 | 1 |
| **0.87** | 0.98 | $10^{-20}$ | 0.004 | $8 \times 10^{-5}$ | 1000 | 0.1 | 0.01 | 3.2336 | 2 |
| **0.90** | 0.98 | $10^{-20}$ | 0.004 | $8 \times 10^{-5}$ | 1000 | 0.1 | 0.01 | 3.2404 | 3 |
| **0.95** | 0.98 | $10^{-20}$ | 0.004 | $8 \times 10^{-5}$ | 1000 | 0.1 | 0.01 | 3.2692 | 4 |
| **0.98** | 0.98 | $10^{-20}$ | 0.004 | $8 \times 10^{-5}$ | 1000 | 0.1 | 0.01 | 3.3810 | 5 |
| 0.87 | **0.90** | $10^{-20}$ | 0.004 | $8 \times 10^{-5}$ | 1000 | 0.1 | 0.01 | 3.2770 | 6 |
| 0.87 | **0.95** | $10^{-20}$ | 0.004 | $8 \times 10^{-5}$ | 1000 | 0.1 | 0.01 | 3.2474 | 7 |
| 0.87 | **0.98** | $10^{-20}$ | 0.004 | $8 \times 10^{-5}$ | 1000 | 0.1 | 0.01 | 3.2336 | 8 |
| 0.87 | 0.98 | $10^{-20}$ | 0.004 | $8 \times 10^{-5}$ | 1000 | 0.1 | 0.01 | 3.2336 | 9 |
| 0.87 | 0.98 | $10^{-10}$ | 0.004 | $8 \times 10^{-5}$ | 1000 | 0.1 | 0.01 | 3.2307 | 10 |
| 0.87 | 0.98 | $10^{-20}$ | **0.004** | $8 \times 10^{-5}$ | 1000 | 0.1 | 0.01 | 3.2336 | 11 |
| 0.87 | 0.98 | $10^{-20}$ | **0.008** | $8 \times 10^{-5}$ | 1000 | 0.1 | 0.01 | 3.2400 | 12 |
| 0.87 | 0.98 | $10^{-20}$ | **0.016** | $8 \times 10^{-5}$ | 1000 | 0.1 | 0.01 | 3.2768 | 13 |
| 0.87 | 0.98 | $10^{-20}$ | 0.004 | $8 \times 10^{-5}$ | 1000 | 0.1 | 0.01 | 3.2336 | 14 |
| 0.87 | 0.98 | $10^{-20}$ | 0.004 | $1.6 \times 10^{-4}$ | 1000 | 0.1 | 0.01 | 3.2327 | 15 |
| 0.87 | 0.98 | $10^{-20}$ | 0.004 | $3.2 \times 10^{-4}$ | 1000 | 0.1 | 0.01 | 3.2342 | 16 |
| 0.87 | 0.98 | $10^{-20}$ | 0.004 | $8 \times 10^{-5}$ | **500** | 0.1 | 0.01 | 3.2385 | 17 |
| 0.87 | 0.98 | $10^{-20}$ | 0.004 | $8 \times 10^{-5}$ | **1000** | 0.1 | 0.01 | 3.2336 | 18 |
| 0.87 | 0.98 | $10^{-20}$ | 0.004 | $8 \times 10^{-5}$ | **2000** | 0.1 | 0.01 | 3.2335 | 19 |
| 0.87 | 0.98 | $10^{-20}$ | 0.004 | $8 \times 10^{-5}$ | 1000 | **0.0** | 0.01 | 3.2862 | 20 |
| 0.87 | 0.98 | $10^{-20}$ | 0.004 | $8 \times 10^{-5}$ | 1000 | **0.1** | 0.01 | 3.2336 | 21 |
| 0.87 | 0.98 | $10^{-20}$ | 0.004 | $8 \times 10^{-5}$ | 1000 | **0.2** | 0.01 | 3.2460 | 22 |
| 0.87 | 0.98 | $10^{-20}$ | 0.004 | $8 \times 10^{-5}$ | 1000 | 0.1 | **0.0** | 3.2329 | 23 |
| 0.87 | 0.98 | $10^{-20}$ | 0.004 | $8 \times 10^{-5}$ | 1000 | 0.1 | **0.01** | 3.2336 | 24 |
| 0.87 | 0.98 | $10^{-20}$ | 0.004 | $8 \times 10^{-5}$ | 1000 | 0.1 | **0.02** | 3.2332 | 25 |

*Table 31.* Hyperparameter ablation for M+Adam on 60M on 8× Chinchilla Data (BF16 weights + FP8 compute)

| $\beta_1$ | $\beta_2$ | $\epsilon$ | $\eta_m$ | $\eta_a$ | warmup | $\lambda_m$ | $\lambda_e$ | Loss | Link |
|---|---|---|---|---|---|---|---|---|---|
| 0.87 | 0.98 | $10^{-20}$ | 0.004 | $8 \times 10^{-5}$ | 1000 | 0.1 | 0.01 | 3.1879 | 1 |
| **0.87** | 0.98 | $10^{-20}$ | 0.004 | $8 \times 10^{-5}$ | 1000 | 0.1 | 0.01 | 3.1879 | 2 |
| **0.90** | 0.98 | $10^{-20}$ | 0.004 | $8 \times 10^{-5}$ | 1000 | 0.1 | 0.01 | 3.1937 | 3 |
| **0.95** | 0.98 | $10^{-20}$ | 0.004 | $8 \times 10^{-5}$ | 1000 | 0.1 | 0.01 | 3.2195 | 4 |
| **0.98** | 0.98 | $10^{-20}$ | 0.004 | $8 \times 10^{-5}$ | 1000 | 0.1 | 0.01 | 3.3184 | 5 |
| 0.87 | **0.90** | $10^{-20}$ | 0.004 | $8 \times 10^{-5}$ | 1000 | 0.1 | 0.01 | 3.2380 | 6 |
| 0.87 | **0.95** | $10^{-20}$ | 0.004 | $8 \times 10^{-5}$ | 1000 | 0.1 | 0.01 | 3.2024 | 7 |
| 0.87 | **0.98** | $10^{-20}$ | 0.004 | $8 \times 10^{-5}$ | 1000 | 0.1 | 0.01 | 3.1879 | 8 |
| 0.87 | 0.98 | $10^{-20}$ | 0.004 | $8 \times 10^{-5}$ | 1000 | 0.1 | 0.01 | 3.1879 | 9 |
| 0.87 | 0.98 | $10^{-10}$ | 0.004 | $8 \times 10^{-5}$ | 1000 | 0.1 | 0.01 | 3.1849 | 10 |
| 0.87 | 0.98 | $10^{-20}$ | **0.004** | $8 \times 10^{-5}$ | 1000 | 0.1 | 0.01 | 3.1879 | 11 |
| 0.87 | 0.98 | $10^{-20}$ | **0.008** | $8 \times 10^{-5}$ | 1000 | 0.1 | 0.01 | 3.1982 | 12 |
| 0.87 | 0.98 | $10^{-20}$ | **0.016** | $8 \times 10^{-5}$ | 1000 | 0.1 | 0.01 | 3.2311 | 13 |
| 0.87 | 0.98 | $10^{-20}$ | 0.004 | $8 \times 10^{-5}$ | 1000 | 0.1 | 0.01 | 3.1879 | 14 |
| 0.87 | 0.98 | $10^{-20}$ | 0.004 | $1.6 \times 10^{-4}$ | 1000 | 0.1 | 0.01 | 3.1868 | 15 |
| 0.87 | 0.98 | $10^{-20}$ | 0.004 | $3.2 \times 10^{-4}$ | 1000 | 0.1 | 0.01 | 3.1896 | 16 |
| 0.87 | 0.98 | $10^{-20}$ | 0.004 | $8 \times 10^{-5}$ | **500** | 0.1 | 0.01 | 3.1903 | 17 |
| 0.87 | 0.98 | $10^{-20}$ | 0.004 | $8 \times 10^{-5}$ | **1000** | 0.1 | 0.01 | 3.1879 | 18 |
| 0.87 | 0.98 | $10^{-20}$ | 0.004 | $8 \times 10^{-5}$ | **2000** | 0.1 | 0.01 | 3.1890 | 19 |
| 0.87 | 0.98 | $10^{-20}$ | 0.004 | $8 \times 10^{-5}$ | 1000 | **0.0** | 0.01 | 3.2468 | 20 |
| 0.87 | 0.98 | $10^{-20}$ | 0.004 | $8 \times 10^{-5}$ | 1000 | **0.1** | 0.01 | 3.1879 | 21 |
| 0.87 | 0.98 | $10^{-20}$ | 0.004 | $8 \times 10^{-5}$ | 1000 | **0.2** | 0.01 | 3.2063 | 22 |
| 0.87 | 0.98 | $10^{-20}$ | 0.004 | $8 \times 10^{-5}$ | 1000 | 0.1 | **0.0** | 3.1866 | 23 |
| 0.87 | 0.98 | $10^{-20}$ | 0.004 | $8 \times 10^{-5}$ | 1000 | 0.1 | **0.02** | 3.1882 | 24 |

*Table 32.* Hyperparameter ablation for M+Adam on 130M on 1× Chinchilla Data (BF16 weights + FP8 compute)

| $\beta_1$ | $\beta_2$ | $\epsilon$ | $\eta_m$ | $\eta_a$ | warmup | $\lambda_m$ | $\lambda_e$ | Loss | Link |
|---|---|---|---|---|---|---|---|---|---|
| 0.87 | 0.98 | $10^{-20}$ | 0.004 | $3.2 \times 10^{-4}$ | 2000 | 0.1 | 0.01 | 3.0972 | 1 |
| **0.87** | 0.98 | $10^{-20}$ | 0.004 | $8 \times 10^{-5}$ | 1000 | 0.1 | 0.01 | 3.1410 | 2 |
| **0.90** | 0.98 | $10^{-20}$ | 0.004 | $8 \times 10^{-5}$ | 1000 | 0.1 | 0.01 | 3.1856 | 3 |
| **0.95** | 0.98 | $10^{-20}$ | 0.004 | $8 \times 10^{-5}$ | 1000 | 0.1 | 0.01 | 3.1700 | 4 |
| **0.98** | 0.98 | $10^{-20}$ | 0.004 | $8 \times 10^{-5}$ | 1000 | 0.1 | 0.01 | 4.1064 | 5 |
| 0.87 | **0.90** | $10^{-20}$ | 0.004 | $8 \times 10^{-5}$ | 1000 | 0.1 | 0.01 | 3.1679 | 6 |
| 0.87 | **0.95** | $10^{-20}$ | 0.004 | $8 \times 10^{-5}$ | 1000 | 0.1 | 0.01 | 3.1448 | 7 |
| 0.87 | **0.98** | $10^{-20}$ | 0.004 | $8 \times 10^{-5}$ | 1000 | 0.1 | 0.01 | 3.1410 | 8 |
| 0.87 | 0.98 | $10^{-20}$ | 0.004 | $8 \times 10^{-5}$ | 1000 | 0.1 | 0.01 | 3.1410 | 9 |
| 0.87 | 0.98 | $10^{-10}$ | 0.004 | $8 \times 10^{-5}$ | 1000 | 0.1 | 0.01 | 3.1384 | 10 |
| 0.87 | 0.98 | $10^{-20}$ | **0.004** | $8 \times 10^{-5}$ | 1000 | 0.1 | 0.01 | 3.1410 | 11 |
| 0.87 | 0.98 | $10^{-20}$ | **0.008** | $8 \times 10^{-5}$ | 1000 | 0.1 | 0.01 | 3.1989 | 12 |
| 0.87 | 0.98 | $10^{-20}$ | **0.016** | $8 \times 10^{-5}$ | 1000 | 0.1 | 0.01 | 7.1386 | 13 |
| 0.87 | 0.98 | $10^{-20}$ | 0.004 | $8 \times 10^{-5}$ | 1000 | 0.1 | 0.01 | 3.1410 | 14 |
| 0.87 | 0.98 | $10^{-20}$ | 0.004 | $1.6 \times 10^{-4}$ | 1000 | 0.1 | 0.01 | 3.1325 | 15 |
| 0.87 | 0.98 | $10^{-20}$ | 0.004 | $3.2 \times 10^{-4}$ | 1000 | 0.1 | 0.01 | 3.1187 | 16 |
| 0.87 | 0.98 | $10^{-20}$ | 0.004 | $3.2 \times 10^{-4}$ | **500** | 0.1 | 0.01 | 3.1535 | 17 |
| 0.87 | 0.98 | $10^{-20}$ | 0.004 | $3.2 \times 10^{-4}$ | **1000** | 0.1 | 0.01 | 3.1187 | 18 |
| 0.87 | 0.98 | $10^{-20}$ | 0.004 | $3.2 \times 10^{-4}$ | **2000** | 0.1 | 0.01 | 3.0972 | 19 |
| 0.87 | 0.98 | $10^{-20}$ | 0.004 | $3.2 \times 10^{-4}$ | 2000 | **0.0** | 0.01 | 3.9269 | 20 |
| 0.87 | 0.98 | $10^{-20}$ | 0.004 | $3.2 \times 10^{-4}$ | 2000 | **0.1** | 0.01 | 3.0972 | 21 |
| 0.87 | 0.98 | $10^{-20}$ | 0.004 | $3.2 \times 10^{-4}$ | 2000 | **0.2** | 0.01 | 3.1054 | 22 |
| 0.87 | 0.98 | $10^{-20}$ | 0.004 | $3.2 \times 10^{-4}$ | 2000 | 0.1 | **0.0** | 3.0977 | 23 |
| 0.87 | 0.98 | $10^{-20}$ | 0.004 | $3.2 \times 10^{-4}$ | 2000 | 0.1 | **0.01** | 3.0972 | 24 |
| 0.87 | 0.98 | $10^{-20}$ | 0.004 | $3.2 \times 10^{-4}$ | 2000 | 0.1 | **0.02** | 3.0980 | 25 |

*Table 33.* Hyperparameter ablation for M+Adam on 350M on $1\times$ Chinchilla Data (BF16 weights + FP8 compute)

| $\beta_1$ | $\beta_2$ | $\epsilon$ | $\eta_m$ | $\eta_a$ | warmup | $\lambda_m$ | $\lambda_e$ | Loss | Link |
|---|---|---|---|---|---|---|---|---|---|
| 0.90 | 0.98 | $10^{-10}$ | 0.004 | $8\times10^{-5}$ | 1000 | 0.1 | 0.01 | 2.8851 | 1 |
| **0.87** | 0.98 | $10^{-20}$ | 0.004 | $8\times10^{-5}$ | 1000 | 0.1 | 0.01 | 2.9296 | 2 |
| **0.90** | 0.98 | $10^{-20}$ | 0.004 | $8\times10^{-5}$ | 1000 | 0.1 | 0.01 | 2.9029 | 3 |
| **0.95** | 0.98 | $10^{-20}$ | 0.004 | $8\times10^{-5}$ | 1000 | 0.1 | 0.01 | 4.2848 | 4 |
| **0.98** | 0.98 | $10^{-20}$ | 0.004 | $8\times10^{-5}$ | 1000 | 0.1 | 0.01 | 4.3719 | 5 |
| 0.90 | **0.90** | $10^{-20}$ | 0.004 | $8\times10^{-5}$ | 1000 | 0.1 | 0.01 | 5.1978 | 6 |
| 0.90 | **0.95** | $10^{-20}$ | 0.004 | $8\times10^{-5}$ | 1000 | 0.1 | 0.01 | 2.9882 | 7 |
| 0.90 | **0.98** | $10^{-20}$ | 0.004 | $8\times10^{-5}$ | 1000 | 0.1 | 0.01 | 2.9029 | 8 |
| 0.90 | 0.98 | $10^{-20}$ | 0.004 | $8\times10^{-5}$ | 1000 | 0.1 | 0.01 | 2.9029 | 9 |
| 0.90 | 0.98 | $10^{-10}$ | 0.004 | $8\times10^{-5}$ | 1000 | 0.1 | 0.01 | 2.8851 | 10 |
| 0.90 | 0.98 | $10^{-10}$ | **0.004** | $8\times10^{-5}$ | 1000 | 0.1 | 0.01 | 2.8851 | 11 |
| 0.90 | 0.98 | $10^{-10}$ | **0.008** | $8\times10^{-5}$ | 1000 | 0.1 | 0.01 | 5.7470 | 12 |
| 0.90 | 0.98 | $10^{-10}$ | **0.016** | $8\times10^{-5}$ | 1000 | 0.1 | 0.01 | 6.7298 | 13 |
| 0.90 | 0.98 | $10^{-10}$ | 0.004 | $8\times10^{-5}$ | 1000 | 0.1 | 0.01 | 2.8851 | 14 |
| 0.90 | 0.98 | $10^{-10}$ | 0.004 | $1.6\times10^{-4}$ | 1000 | 0.1 | 0.01 | 2.8967 | 15 |
| 0.90 | 0.98 | $10^{-10}$ | 0.004 | $3.2\times10^{-4}$ | 1000 | 0.1 | 0.01 | 2.9485 | 16 |
| 0.90 | 0.98 | $10^{-10}$ | 0.004 | $8\times10^{-5}$ | **1000** | 0.1 | 0.01 | 2.8851 | 17 |
| 0.90 | 0.98 | $10^{-10}$ | 0.004 | $8\times10^{-5}$ | **2000** | 0.1 | 0.01 | 4.4545 | 18 |
| 0.90 | 0.98 | $10^{-10}$ | 0.004 | $8\times10^{-5}$ | 1000 | **0.0** | 0.01 | 6.2177 | 19 |
| 0.90 | 0.98 | $10^{-10}$ | 0.004 | $8\times10^{-5}$ | 1000 | **0.1** | 0.01 | 2.8851 | 20 |
| 0.90 | 0.98 | $10^{-10}$ | 0.004 | $8\times10^{-5}$ | 1000 | **0.2** | 0.01 | 6.4175 | 21 |
| 0.90 | 0.98 | $10^{-10}$ | 0.004 | $8\times10^{-5}$ | 1000 | 0.1 | **0.0** | 2.8983 | 22 |
| 0.90 | 0.98 | $10^{-10}$ | 0.004 | $8\times10^{-5}$ | 1000 | 0.1 | **0.01** | 2.8851 | 23 |
| 0.90 | 0.98 | $10^{-10}$ | 0.004 | $8\times10^{-5}$ | 1000 | 0.1 | **0.02** | 6.0207 | 24 |

## I.5. Hyperparameter Ablation in Phase II

*Table 34.* Hyperparameter ablation for AdamW on 130M on $2\times$ Chinchilla Data (BF16 weights + BF16 compute)

| $\beta_1$ | $\beta_2$ | $\epsilon$ | $\eta$ | $g_{\text{norm}}$ | warmup | wd | Loss | Link |
|---|---|---|---|---|---|---|---|---|
| 0.95 | 0.98 | $10^{-10}$ | 0.008 | 1.0 | 2000 | 0.5 | 3.0918 | 1 |
| **0.87** | 0.95 | $10^{-10}$ | 0.004 | 1.0 | 1000 | 0.5 | 4.5454 | 2 |
| **0.95** | 0.95 | $10^{-10}$ | 0.004 | 1.0 | 1000 | 0.5 | 3.1778 | 3 |
| 0.95 | **0.95** | $10^{-10}$ | 0.004 | 1.0 | 1000 | 0.5 | 3.1778 | 4 |
| 0.95 | **0.98** | $10^{-10}$ | 0.004 | 1.0 | 1000 | 0.5 | 3.1489 | 5 |
| 0.95 | 0.98 | $10^{-10}$ | **0.004** | 1.0 | 1000 | 0.5 | 3.1489 | 6 |
| 0.95 | 0.98 | $10^{-10}$ | **0.008** | 1.0 | 1000 | 0.5 | 3.1287 | 7 |
| 0.95 | 0.98 | $10^{-10}$ | 0.008 | **1.0** | 1000 | 0.5 | 3.1287 | 8 |
| 0.95 | 0.98 | $10^{-10}$ | 0.008 | **2.0** | 1000 | 0.5 | 3.1806 | 9 |
| 0.95 | 0.98 | $10^{-10}$ | 0.008 | 1.0 | **1000** | 0.5 | 3.1287 | 10 |
| 0.95 | 0.98 | $10^{-10}$ | 0.008 | 1.0 | **2000** | 0.5 | 3.0918 | 11 |

*Table 35.* Hyperparameter ablation for AdamW on 130M on 4× Chinchilla Data (BF16 weights + BF16 compute)

| $\beta_1$ | $\beta_2$ | $\epsilon$ | $\eta$ | $g_{\text{norm}}$ | warmup | wd | Loss | Link |
|---|---|---|---|---|---|---|---|---|
| 0.95 | 0.98 | $10^{-10}$ | 0.004 | 1.0 | 2000 | 0.5 | 3.0604 | 1 |
| **0.87** | 0.95 | $10^{-10}$ | 0.004 | 1.0 | 1000 | 0.5 | 5.6002 | 2 |
| **0.95** | 0.95 | $10^{-10}$ | 0.004 | 1.0 | 1000 | 0.5 | 3.1320 | 3 |
| 0.95 | **0.95** | $10^{-10}$ | 0.004 | 1.0 | 1000 | 0.5 | 3.1320 | 4 |
| 0.95 | **0.98** | $10^{-10}$ | 0.004 | 1.0 | 1000 | 0.5 | 3.0920 | 5 |
| 0.95 | 0.98 | $10^{-10}$ | **0.004** | 1.0 | 1000 | 0.5 | 3.0920 | 6 |
| 0.95 | 0.98 | $10^{-10}$ | **0.008** | 1.0 | 1000 | 0.5 | 3.1000 | 7 |
| 0.95 | 0.98 | $10^{-10}$ | 0.004 | **1.0** | 1000 | 0.5 | 3.0920 | 8 |
| 0.95 | 0.98 | $10^{-10}$ | 0.004 | **2.0** | 1000 | 0.5 | 3.0967 | 9 |
| 0.95 | 0.98 | $10^{-10}$ | 0.004 | 1.0 | **1000** | 0.5 | 3.0920 | 10 |
| 0.95 | 0.98 | $10^{-10}$ | 0.004 | 1.0 | **2000** | 0.5 | 3.0604 | 11 |

*Table 36.* Hyperparameter ablation for AdamW on 130M on 8× Chinchilla Data (BF16 weights + BF16 compute)

| $\beta_1$ | $\beta_2$ | $\epsilon$ | $\eta$ | $g_{\text{norm}}$ | warmup | wd | Loss | Link |
|---|---|---|---|---|---|---|---|---|
| 0.95 | 0.98 | $10^{-10}$ | 0.004 | 1.0 | 2000 | 0.5 | 3.0183 | 1 |
| **0.87** | 0.95 | $10^{-10}$ | 0.004 | 1.0 | 1000 | 0.5 | 6.8238 | 2 |
| **0.95** | 0.95 | $10^{-10}$ | 0.004 | 1.0 | 1000 | 0.5 | 3.0942 | 3 |
| 0.95 | **0.95** | $10^{-10}$ | 0.004 | 1.0 | 1000 | 0.5 | 3.0942 | 4 |
| 0.95 | **0.98** | $10^{-10}$ | 0.004 | 1.0 | 1000 | 0.5 | 3.0448 | 5 |
| 0.95 | 0.98 | $10^{-10}$ | **0.004** | 1.0 | 1000 | 0.5 | 3.0448 | 6 |
| 0.95 | 0.98 | $10^{-10}$ | **0.008** | 1.0 | 1000 | 0.5 | 3.0565 | 7 |
| 0.95 | 0.98 | $10^{-10}$ | 0.004 | **1.0** | 1000 | 0.5 | 3.0448 | 8 |
| 0.95 | 0.98 | $10^{-10}$ | 0.004 | **2.0** | 1000 | 0.5 | 3.0490 | 9 |
| 0.95 | 0.98 | $10^{-10}$ | 0.004 | 1.0 | **1000** | 0.5 | 3.0448 | 10 |
| 0.95 | 0.98 | $10^{-10}$ | 0.004 | 1.0 | **2000** | 0.5 | 3.0183 | 11 |
| **0.95** | **0.98** | $10^{-10}$ | **0.004** | **1.0** | **2000** | **0.5** | **3.0183** | 12 |

*Table 37.* Hyperparameter ablation for M+Adam on 130M on 2× Chinchilla Data (BF16 weights + BF16 compute)

| $\beta_1$ | $\beta_2$ | $\epsilon$ | $\eta_m$ | $\eta_a$ | warmup | $\lambda_m$ | $\lambda_e$ | Loss | Link |
|---|---|---|---|---|---|---|---|---|---|
| 0.87 | 0.98 | $10^{-20}$ | 0.004 | $3.2 \times 10^{-4}$ | 2000 | 0.1 | 0.01 | 3.0181 | 1 |
| 0.87 | 0.98 | $10^{-10}$ | 0.004 | $8 \times 10^{-5}$ | 1000 | 0.1 | 0.01 | 3.0470 | 2 |
| 0.87 | 0.98 | $10^{-20}$ | 0.004 | $8 \times 10^{-5}$ | 1000 | 0.1 | 0.01 | 3.0462 | 3 |
| 0.87 | 0.98 | $10^{-20}$ | 0.004 | $3.2 \times 10^{-4}$ | **1000** | 0.1 | 0.01 | 3.0344 | 4 |
| 0.87 | 0.98 | $10^{-20}$ | 0.004 | $3.2 \times 10^{-4}$ | **2000** | 0.1 | 0.01 | 3.0181 | 5 |
| 0.87 | 0.98 | $10^{-20}$ | 0.004 | $3.2 \times 10^{-4}$ | 2000 | 0.1 | **0.0** | 3.0171 | 6 |

*Table 38.* Hyperparameter ablation for M+Adam on 130M on 4× Chinchilla Data (BF16 weights + BF16 compute)

| $\beta_1$ | $\beta_2$ | $\epsilon$ | $\eta_m$ | $\eta_a$ | warmup | $\lambda_m$ | $\lambda_e$ | Loss | Link |
|---|---|---|---|---|---|---|---|---|---|
| 0.87 | 0.98 | $10^{-20}$ | 0.004 | $3.2 \times 10^{-4}$ | 2000 | 0.1 | 0.01 | 2.9602 | 1 |
| 0.87 | 0.98 | $10^{-10}$ | 0.004 | $8 \times 10^{-5}$ | 1000 | 0.1 | 0.01 | 2.9829 | 2 |
| 0.87 | 0.98 | $10^{-20}$ | 0.004 | $8 \times 10^{-5}$ | 1000 | 0.1 | 0.01 | 2.9833 | 3 |
| 0.87 | 0.98 | $10^{-20}$ | 0.004 | $3.2 \times 10^{-4}$ | **1000** | 0.1 | 0.01 | 2.9731 | 4 |
| 0.87 | 0.98 | $10^{-20}$ | 0.004 | $3.2 \times 10^{-4}$ | **2000** | 0.1 | 0.01 | 2.9602 | 5 |
| 0.87 | 0.98 | $10^{-20}$ | 0.004 | $3.2 \times 10^{-4}$ | 2000 | 0.1 | **0.0** | 2.9597 | 6 |

*Table 39.* Hyperparameter ablation for M+Adam on 130M on $8\times$ Chinchilla Data (BF16 weights + BF16 compute)

| $\beta_1$ | $\beta_2$ | $\epsilon$ | $\eta_m$ | $\eta_a$ | warmup | $\lambda_m$ | $\lambda_e$ | Loss | Link |
|---|---|---|---|---|---|---|---|---|---|
| 0.87 | 0.98 | $10^{-20}$ | 0.004 | $3.2 \times 10^{-4}$ | 2000 | 0.1 | 0.01 | 2.9189 | 1 |
| 0.87 | 0.98 | $10^{-10}$ | 0.004 | $8 \times 10^{-5}$ | 1000 | 0.1 | 0.01 | 2.9359 | 2 |
| 0.87 | 0.98 | $10^{-20}$ | 0.004 | $8 \times 10^{-5}$ | 1000 | 0.1 | 0.01 | 2.9370 | 3 |
| 0.87 | 0.98 | $10^{-20}$ | 0.004 | $3.2 \times 10^{-4}$ | **1000** | 0.1 | 0.01 | 2.9284 | 4 |
| 0.87 | 0.98 | $10^{-20}$ | 0.004 | $3.2 \times 10^{-4}$ | **2000** | 0.1 | 0.01 | 2.9189 | 5 |
| 0.87 | 0.98 | $10^{-20}$ | 0.004 | $3.2 \times 10^{-4}$ | 2000 | 0.1 | **0.0** | 2.9188 | 6 |

*Table 40.* Hyperparameter ablation for AdamW on 350M on $2\times$ Chinchilla Data (BF16 weights + BF16 compute)

| $\beta_1$ | $\beta_2$ | $\epsilon$ | $\eta$ | $g_{\text{norm}}$ | warmup | $\lambda$ | Loss | Link |
|---|---|---|---|---|---|---|---|---|
| 0.90 | 0.98 | $10^{-10}$ | 0.008 | 1.0 | 1000 | 0.5 | 2.9571 | 1 |
| **0.87** | 0.98 | $10^{-10}$ | 0.008 | 1.0 | 1000 | 0.5 | 2.9459 | 2 |
| 0.87 | 0.98 | $10^{-10}$ | 0.008 | 1.0 | **2000** | 0.5 | 2.9240 | 3 |
| **0.90** | 0.98 | $10^{-10}$ | 0.008 | 1.0 | 2000 | 0.5 | 2.9003 | 4 |

*Table 41.* Hyperparameter ablation for AdamW on 350M on $4\times$ Chinchilla Data (BF16 weights + BF16 compute)

| $\beta_1$ | $\beta_2$ | $\epsilon$ | $\eta$ | $g_{\text{norm}}$ | warmup | $\lambda$ | Loss | Link |
|---|---|---|---|---|---|---|---|---|
| 0.90 | 0.98 | $10^{-10}$ | 0.008 | 1.0 | 1000 | 0.5 | 2.8997 | 1 |
| **0.87** | 0.98 | $10^{-10}$ | 0.008 | 1.0 | 1000 | 0.5 | 2.8982 | 2 |
| **0.95** | 0.98 | $10^{-10}$ | 0.008 | 1.0 | 1000 | 0.5 | 2.9441 | 3 |
| 0.90 | **0.95** | $10^{-10}$ | 0.008 | 1.0 | 1000 | 0.5 | 2.9785 | 4 |
| 0.90 | 0.98 | $10^{-10}$ | 0.008 | **2.0** | 1000 | 0.5 | 2.8933 | 5 |
| 0.87 | 0.98 | $10^{-10}$ | 0.008 | 1.0 | **2000** | 0.5 | 2.8844 | 6 |
| 0.90 | 0.98 | $10^{-10}$ | 0.008 | 1.0 | **2000** | 0.5 | 2.8709 | 7 |
| 0.90 | 0.98 | $10^{-10}$ | 0.008 | **2.0** | 2000 | 0.5 | 2.8187 | 8 |

*Table 42.* Hyperparameter ablation for AdamW on 350M on $8\times$ Chinchilla Data (BF16 weights + BF16 compute)

| $\beta_1$ | $\beta_2$ | $\epsilon$ | $\eta$ | $g_{\text{norm}}$ | warmup | $\lambda$ | Loss | Link |
|---|---|---|---|---|---|---|---|---|
| 0.90 | 0.98 | $10^{-10}$ | 0.008 | 1.0 | 1000 | 0.5 | 2.8187 | 1 |
| **0.87** | 0.98 | $10^{-10}$ | 0.008 | 1.0 | 1000 | 0.5 | 2.8743 | 2 |
| 0.90 | 0.98 | $10^{-10}$ | 0.008 | **2.0** | **2000** | 0.5 | 2.7696 | 3 |

*Table 43.* Hyperparameter ablation for M+Adam on 350M on $2\times$ Chinchilla Data (BF16 weights + BF16 compute)

| $\beta_1$ | $\beta_2$ | $\epsilon$ | $\eta_m$ | $\eta_a$ | warmup | $\lambda_m$ | $\lambda_e$ | Loss | Link |
|---|---|---|---|---|---|---|---|---|---|
| 0.87 | 0.98 | $10^{-20}$ | 0.004 | $8 \times 10^{-5}$ | 1000 | 0.1 | 0.01 | 2.8575 | 1 |
| 0.87 | 0.98 | $10^{-10}$ | 0.004 | $8 \times 10^{-5}$ | 1000 | 0.1 | 0.01 | 2.8013 | 2 |
| 0.87 | 0.98 | $10^{-10}$ | 0.004 | $3.2 \times 10^{-4}$ | 1000 | 0.1 | 0.01 | 2.8385 | 3 |
| 0.87 | 0.98 | $10^{-10}$ | 0.004 | $8 \times 10^{-5}$ | **2000** | 0.1 | 0.01 | 2.7605 | 4 |
| 0.87 | 0.98 | $10^{-10}$ | 0.004 | $8 \times 10^{-5}$ | 2000 | 0.1 | **0.0** | 2.7733 | 5 |

*Table 44.* Hyperparameter ablation for M+Adam on 350M on $4\times$ Chinchilla Data (BF16 weights + BF16 compute)

| $\beta_1$ | $\beta_2$ | $\epsilon$ | $\eta_m$ | $\eta_a$ | warmup | $\lambda_m$ | $\lambda_e$ | Loss | Link |
|---|---|---|---|---|---|---|---|---|---|
| 0.87 | 0.98 | $10^{-20}$ | 0.004 | $3.2 \times 10^{-4}$ | 2000 | 0.1 | 0.01 | 2.7231 | 1 |
| 0.87 | 0.98 | $10^{-20}$ | 0.004 | $3.2 \times 10^{-4}$ | 2000 | 0.1 | **0.0** | 2.7114 | 2 |

*Table 45.* Hyperparameter ablation for M+Adam on 350M on $8\times$ Chinchilla Data (BF16 weights + BF16 compute)

| $\beta_1$ | $\beta_2$ | $\epsilon$ | $\eta_m$ | $\eta_a$ | warmup | $\lambda_m$ | $\lambda_e$ | Loss | Link |
|---|---|---|---|---|---|---|---|---|---|
| 0.87 | 0.98 | $10^{-20}$ | 0.004 | $3.2 \times 10^{-4}$ | 2000 | 0.1 | 0.01 | 2.7294 | 1 |
| 0.87 | 0.98 | $10^{-20}$ | 0.004 | $3.2 \times 10^{-4}$ | 2000 | 0.1 | **0.0** | 2.6852 | 2 |
| 0.87 | 0.98 | $10^{-10}$ | 0.004 | $8 \times 10^{-5}$ | **1000** | 0.1 | **0.01** | 2.7568 | 3 |

*Table 46.* Hyperparameter ablation for AdamW on 130M on $2\times$ Chinchilla Data (BF16 weights + FP8 compute)

| $\beta_1$ | $\beta_2$ | $\epsilon$ | $\eta$ | $g_{\mathrm{norm}}$ | warmup | wd | Loss | Link |
|---|---|---|---|---|---|---|---|---|
| 0.95 | 0.98 | $10^{-10}$ | 0.008 | 1.0 | 2000 | 0.5 | 3.0937 | 1 |
| **0.95** | 0.98 | $10^{-10}$ | 0.004 | 1.0 | 1000 | 0.5 | 3.1578 | 2 |
| 0.95 | 0.98 | $10^{-10}$ | **0.008** | 1.0 | 1000 | 0.5 | 3.1316 | 3 |
| 0.95 | 0.98 | $10^{-10}$ | 0.008 | **0.0** | 1000 | 0.5 | 3.1888 | 4 |
| 0.95 | 0.98 | $10^{-10}$ | 0.008 | 1.0 | **2000** | 0.5 | 3.0937 | 5 |

*Table 47.* Hyperparameter ablation for AdamW on 130M on $4\times$ Chinchilla Data (BF16 weights + FP8 compute)

| $\beta_1$ | $\beta_2$ | $\epsilon$ | $\eta$ | $g_{\mathrm{norm}}$ | warmup | wd | Loss | Link |
|---|---|---|---|---|---|---|---|---|
| 0.95 | 0.98 | $10^{-10}$ | 0.008 | 1.0 | 2000 | 0.5 | 3.0431 | 1 |
| **0.95** | 0.98 | $10^{-10}$ | 0.004 | 1.0 | 1000 | 0.5 | 3.0998 | 2 |
| 0.95 | 0.98 | $10^{-10}$ | **0.008** | 1.0 | 1000 | 0.5 | 3.0819 | 3 |
| 0.95 | 0.98 | $10^{-10}$ | 0.008 | **0.0** | 1000 | 0.5 | 3.1674 | 4 |
| 0.95 | 0.98 | $10^{-10}$ | 0.008 | 1.0 | **2000** | 0.5 | 3.0431 | 5 |

*Table 48.* Hyperparameter ablation for AdamW on 130M on $8\times$ Chinchilla Data (BF16 weights + FP8 compute)

| $\beta_1$ | $\beta_2$ | $\epsilon$ | $\eta$ | $g_{\mathrm{norm}}$ | warmup | wd | Loss | Link |
|---|---|---|---|---|---|---|---|---|
| 0.95 | 0.98 | $10^{-10}$ | 0.008 | 0.0 | 1000 | 0.5 | 2.9992 | 1 |
| **0.95** | 0.98 | $10^{-10}$ | 0.004 | 1.0 | 1000 | 0.5 | 3.0544 | 2 |
| 0.95 | 0.98 | $10^{-10}$ | **0.008** | 1.0 | 1000 | 0.5 | 3.0205 | 3 |
| 0.95 | 0.98 | $10^{-10}$ | 0.008 | **0.0** | 1000 | 0.5 | 2.9992 | 4 |
| 0.95 | 0.98 | $10^{-10}$ | 0.008 | 0.0 | **2000** | 0.5 | 3.0040 | 5 |

*Table 49.* Hyperparameter ablation for M+Adam on 130M on $2\times$ Chinchilla Data (BF16 weights + FP8 compute)

| $\beta_1$ | $\beta_2$ | $\epsilon$ | $\eta_m$ | $\eta_a$ | warmup | $\lambda_m$ | $\lambda_e$ | Loss | Link |
|---|---|---|---|---|---|---|---|---|---|
| 0.87 | 0.98 | $10^{-20}$ | 0.004 | $3.2 \times 10^{-4}$ | 2000 | 0.1 | 0.01 | 3.0214 | 1 |
| 0.87 | 0.98 | $10^{-20}$ | 0.004 | $8 \times 10^{-5}$ | 1000 | 0.1 | 0.01 | 3.0492 | 2 |
| 0.87 | 0.98 | $10^{-10}$ | 0.004 | $8 \times 10^{-5}$ | 1000 | 0.1 | 0.01 | 3.0468 | 3 |
| 0.87 | 0.98 | $10^{-20}$ | 0.004 | $3.2 \times 10^{-4}$ | 1000 | 0.1 | 0.01 | 3.0362 | 4 |
| 0.87 | 0.98 | $10^{-20}$ | 0.004 | $3.2 \times 10^{-4}$ | **2000** | 0.1 | 0.01 | 3.0214 | 5 |

*Table 50.* Hyperparameter ablation for M+Adam on 130M on $4\times$ Chinchilla Data (BF16 weights + FP8 compute)

| $\beta_1$ | $\beta_2$ | $\epsilon$ | $\eta_m$ | $\eta_a$ | warmup | $\lambda_m$ | $\lambda_e$ | Loss | Link |
|---|---|---|---|---|---|---|---|---|---|
| 0.87 | 0.98 | $10^{-20}$ | 0.004 | $3.2 \times 10^{-4}$ | 2000 | 0.1 | 0.01 | 2.9646 | 1 |
| 0.87 | 0.98 | $10^{-20}$ | 0.004 | $8 \times 10^{-5}$ | 1000 | 0.1 | 0.01 | 2.9851 | 2 |
| 0.87 | 0.98 | $10^{-10}$ | 0.004 | $8 \times 10^{-5}$ | 1000 | 0.1 | 0.01 | 2.9846 | 3 |
| 0.87 | 0.98 | $10^{-20}$ | 0.004 | $3.2 \times 10^{-4}$ | 1000 | 0.1 | 0.01 | 2.9747 | 4 |
| 0.87 | 0.98 | $10^{-20}$ | 0.004 | $3.2 \times 10^{-4}$ | **2000** | 0.1 | 0.01 | 2.9646 | 5 |

*Table 51.* Hyperparameter ablation for M+Adam on 130M on 8× Chinchilla Data (BF16 weights + FP8 compute)

| $\beta_1$ | $\beta_2$ | $\epsilon$ | $\eta_m$ | $\eta_a$ | warmup | $\lambda_m$ | $\lambda_e$ | Loss | Link |
|---|---|---|---|---|---|---|---|---|---|
| 0.87 | 0.98 | $10^{-20}$ | 0.004 | $3.2 \times 10^{-4}$ | 2000 | 0.1 | 0.01 | 2.9238 | 1 |
| 0.87 | 0.98 | $10^{-20}$ | 0.004 | $8 \times 10^{-5}$ | 1000 | 0.1 | 0.01 | 2.9389 | 2 |
| 0.87 | 0.98 | $10^{-10}$ | 0.004 | $8 \times 10^{-5}$ | 1000 | 0.1 | 0.01 | 2.9374 | 3 |
| 0.87 | 0.98 | $10^{-20}$ | 0.004 | $3.2 \times 10^{-4}$ | 1000 | 0.1 | 0.01 | 2.9319 | 4 |
| 0.87 | 0.98 | $10^{-20}$ | 0.004 | $3.2 \times 10^{-4}$ | **2000** | 0.1 | 0.01 | 2.9238 | 5 |

*Table 52.* Hyperparameter ablation for AdamW on 350M on 2× Chinchilla Data (BF16 weights + FP8 compute)

| $\beta_1$ | $\beta_2$ | $\epsilon$ | $\eta$ | $g_{\text{norm}}$ | warmup | $\lambda$ | Loss | Link |
|---|---|---|---|---|---|---|---|---|
| 0.87 | 0.98 | $10^{-10}$ | 0.004 | 1.0 | 1000 | 0.5 | 6.8527 | 1 |
| **0.95** | 0.98 | $10^{-10}$ | 0.004 | 1.0 | 1000 | 0.5 | 6.2821 | 2 |
| 0.95 | 0.98 | $10^{-10}$ | **0.008** | 1.0 | 1000 | 0.5 | 2.9676 | 3 |
| **0.87** | 0.98 | $10^{-10}$ | 0.008 | 1.0 | 1000 | 0.5 | 2.9284 | 4 |
| 0.87 | 0.98 | $10^{-20}$ | 0.008 | 1.0 | 1000 | 0.5 | 2.8664 | 5 |
| **0.95** | 0.98 | $10^{-20}$ | 0.008 | 1.0 | 1000 | 0.5 | 2.9470 | 6 |
| 0.87 | 0.98 | $10^{-20}$ | 0.008 | 1.0 | **2000** | 0.5 | 2.9031 | 7 |

*Table 53.* Hyperparameter ablation for AdamW on 350M on 4× Chinchilla Data (BF16 weights + FP8 compute)

| $\beta_1$ | $\beta_2$ | $\epsilon$ | $\eta$ | $g_{\text{norm}}$ | warmup | $\lambda$ | Loss | Link |
|---|---|---|---|---|---|---|---|---|
| 0.87 | 0.98 | $10^{-10}$ | 0.004 | 1.0 | 1000 | 0.5 | 6.9507 | 1 |
| 0.87 | 0.98 | $10^{-10}$ | **0.008** | 1.0 | 1000 | 0.5 | 2.8563 | 2 |
| **0.95** | 0.98 | $10^{-10}$ | 0.008 | 1.0 | 1000 | 0.5 | 2.8070 | 3 |

*Table 54.* Hyperparameter ablation for AdamW on 350M on 8× Chinchilla Data (BF16 weights + FP8 compute)

| $\beta_1$ | $\beta_2$ | $\epsilon$ | $\eta$ | $g_{\text{norm}}$ | warmup | $\lambda$ | Loss | Link |
|---|---|---|---|---|---|---|---|---|
| 0.87 | 0.95 | $10^{-10}$ | 0.008 | 1.0 | 1000 | 0.5 | 2.8489 | 1 |
| 0.87 | **0.98** | $10^{-10}$ | 0.008 | 1.0 | 1000 | 0.5 | 2.8232 | 2 |
| **0.95** | 0.98 | $10^{-10}$ | 0.008 | 1.0 | 1000 | 0.5 | 2.8006 | 3 |
| 0.95 | 0.98 | $10^{-10}$ | 0.008 | 1.0 | **2000** | 0.5 | 2.7823 | 4 |
| 0.95 | 0.98 | $10^{-20}$ | 0.008 | 1.0 | 2000 | 0.5 | 2.7663 | 5 |

*Table 55.* Hyperparameter ablation for M+Adam on 350M on 2× Chinchilla Data (BF16 weights + FP8 compute)

| $\beta_1$ | $\beta_2$ | $\epsilon$ | $\eta_m$ | $\eta_a$ | warmup | $\lambda_m$ | $\lambda_e$ | Loss | Link |
|---|---|---|---|---|---|---|---|---|---|
| 0.87 | 0.98 | $10^{-20}$ | 0.004 | $8 \times 10^{-5}$ | 1000 | 0.1 | 0.01 | 2.8449 | 1 |
| 0.87 | 0.98 | $10^{-10}$ | 0.004 | $8 \times 10^{-5}$ | 1000 | 0.1 | 0.01 | 2.8337 | 2 |
| 0.87 | 0.98 | $10^{-10}$ | 0.004 | $8 \times 10^{-5}$ | 1000 | 0.1 | 0.01 | 2.8190 | 3 |

*Table 56.* Hyperparameter ablation for M+Adam on 350M on 4× Chinchilla Data (BF16 weights + FP8 compute)

| $\beta_1$ | $\beta_2$ | $\epsilon$ | $\eta_m$ | $\eta_a$ | warmup | $\lambda_m$ | $\lambda_e$ | Loss | Link |
|---|---|---|---|---|---|---|---|---|---|
| 0.90 | 0.98 | $10^{-10}$ | 0.004 | $8 \times 10^{-5}$ | 1000 | 0.1 | 0.01 | 2.7954 | 1 |
| **0.87** | 0.98 | $10^{-10}$ | 0.004 | $8 \times 10^{-5}$ | 1000 | 0.1 | 0.01 | 3.4313 | 2 |
| 0.87 | 0.98 | $10^{-20}$ | 0.004 | $8 \times 10^{-5}$ | 1000 | 0.1 | 0.01 | 2.7648 | 3 |
| 0.87 | 0.98 | $10^{-20}$ | 0.004 | $8 \times 10^{-5}$ | 1000 | 0.1 | **0.00** | 3.9468 | 4 |

*Table 57.* Hyperparameter ablation for M+Adam on 350M on 8× Chinchilla Data (BF16 weights + FP8 compute)

| $\beta_1$ | $\beta_2$ | $\epsilon$ | $\eta_m$ | $\eta_a$ | warmup | $\lambda_m$ | $\lambda_e$ | Loss | Link |
|------|------|------------|-------|-----------------|--------|------|------|--------|------|
| 0.90 | 0.98 | $10^{-10}$ | 0.004 | $8 \times 10^{-5}$ | 1000 | 0.1 | 0.01 | 6.5851 | 1 |
| 0.90 | 0.98 | $10^{-10}$ | 0.004 | $8 \times 10^{-5}$ | **2000** | 0.1 | **0.00** | 2.7378 | 2 |
| 0.90 | 0.98 | $10^{-10}$ | **0.002** | $8 \times 10^{-5}$ | 1000 | 0.1 | 0.00 | 2.6963 | 3 |

## I.6. Hyperparameter Ablation of FP8 master weights with FP8 compute

*Table 58.* Hyperparameter ablation for AdamW on 60M on 1× Chinchilla Data (FP8 weights + FP8 compute)

| $\beta_1$ | $\beta_2$ | $\epsilon$ | $\eta$ | $g_{\text{norm}}$ | warmup | wd | Loss | Link |
|------|------|------------|--------|-------|--------|-----|--------|------|
| 0.87 | 0.95 | $10^{-10}$ | 0.004 | 1.0 | 1000 | 0.5 | 3.4519 | 1 |
| **0.87** | 0.95 | $10^{-10}$ | 0.004 | 1.0 | 1000 | 0.5 | 3.4519 | 2 |
| **0.90** | 0.95 | $10^{-10}$ | 0.004 | 1.0 | 1000 | 0.5 | 3.4631 | 3 |
| **0.95** | 0.95 | $10^{-10}$ | 0.004 | 1.0 | 1000 | 0.5 | 3.5713 | 4 |
| **0.98** | 0.95 | $10^{-10}$ | 0.004 | 1.0 | 1000 | 0.5 | 3.8539 | 5 |
| 0.87 | **0.90** | $10^{-10}$ | 0.004 | 1.0 | 1000 | 0.5 | 3.4692 | 6 |
| 0.87 | **0.95** | $10^{-10}$ | 0.004 | 1.0 | 1000 | 0.5 | 3.4519 | 7 |
| 0.87 | **0.98** | $10^{-10}$ | 0.004 | 1.0 | 1000 | 0.5 | 3.4878 | 8 |
| 0.87 | 0.95 | $10^{-10}$ | 0.004 | 1.0 | 1000 | 0.5 | 3.4519 | 9 |
| 0.87 | 0.95 | $10^{-20}$ | 0.004 | 1.0 | 1000 | 0.5 | 3.4508 | 10 |
| 0.87 | 0.95 | $10^{-10}$ | **0.004** | 1.0 | 1000 | 0.5 | 3.4519 | 11 |
| 0.87 | 0.95 | $10^{-10}$ | **0.008** | 1.0 | 1000 | 0.5 | 3.5245 | 12 |
| 0.87 | 0.95 | $10^{-10}$ | **0.016** | 1.0 | 1000 | 0.5 | 3.5528 | 13 |
| 0.87 | 0.95 | $10^{-10}$ | 0.004 | **0.0** | 1000 | 0.5 | 3.4559 | 14 |
| 0.87 | 0.95 | $10^{-10}$ | 0.004 | **1.0** | 1000 | 0.5 | 3.4519 | 15 |
| 0.87 | 0.95 | $10^{-10}$ | 0.004 | **2.0** | 1000 | 0.5 | 3.4524 | 16 |
| 0.87 | 0.95 | $10^{-10}$ | 0.004 | 1.0 | **1000** | 0.5 | 3.4519 | 17 |
| 0.87 | 0.95 | $10^{-10}$ | 0.004 | 1.0 | **2000** | 0.5 | 3.4537 | 18 |
| 0.87 | 0.95 | $10^{-10}$ | 0.004 | 1.0 | 1000 | **0.0** | 3.4575 | 19 |
| 0.87 | 0.95 | $10^{-10}$ | 0.004 | 1.0 | 1000 | **0.5** | 3.4519 | 20 |
| 0.87 | 0.95 | $10^{-10}$ | 0.004 | 1.0 | 1000 | **1.0** | 3.7919 | 21 |

*Table 59.* Hyperparameter ablation for AdamW on 130M on 1× Chinchilla Data (FP8 weights + FP8 compute)

| $\beta_1$ | $\beta_2$ | $\epsilon$ | $\eta$ | $g_{\text{norm}}$ | warmup | wd | Loss | Link |
|---|---|---|---|---|---|---|---|---|
| 0.90 | 0.98 | $10^{-10}$ | 0.004 | 1.0 | 1000 | 0.5 | 3.2175 | 1 |
| **0.87** | 0.95 | $10^{-10}$ | 0.004 | 1.0 | 1000 | 0.5 | 3.2257 | 2 |
| **0.90** | 0.95 | $10^{-10}$ | 0.004 | 1.0 | 1000 | 0.5 | 3.2264 | 3 |
| **0.95** | 0.95 | $10^{-10}$ | 0.004 | 1.0 | 1000 | 0.5 | 3.4394 | 4 |
| **0.98** | 0.95 | $10^{-10}$ | 0.004 | 1.0 | 1000 | 0.5 | 4.2699 | 5 |
| 0.90 | **0.90** | $10^{-10}$ | 0.004 | 1.0 | 1000 | 0.5 | 3.2543 | 6 |
| 0.90 | **0.95** | $10^{-10}$ | 0.004 | 1.0 | 1000 | 0.5 | 3.2264 | 7 |
| 0.90 | **0.98** | $10^{-10}$ | 0.004 | 1.0 | 1000 | 0.5 | 3.2175 | 8 |
| 0.90 | 0.98 | $10^{-10}$ | 0.004 | 1.0 | 1000 | 0.5 | 3.2175 | 9 |
| 0.90 | 0.98 | $10^{-20}$ | 0.004 | 1.0 | 1000 | 0.5 | 3.2214 | 10 |
| 0.90 | 0.98 | $10^{-10}$ | **0.004** | 1.0 | 1000 | 0.5 | 3.2175 | 11 |
| 0.90 | 0.98 | $10^{-10}$ | **0.008** | 1.0 | 1000 | 0.5 | 3.3212 | 12 |
| 0.90 | 0.98 | $10^{-10}$ | **0.016** | 1.0 | 1000 | 0.5 | 3.3830 | 13 |
| 0.90 | 0.98 | $10^{-10}$ | 0.004 | **0.0** | 1000 | 0.5 | 3.2432 | 14 |
| 0.90 | 0.98 | $10^{-10}$ | 0.004 | **1.0** | 1000 | 0.5 | 3.2175 | 15 |
| 0.90 | 0.98 | $10^{-10}$ | 0.004 | **2.0** | 1000 | 0.5 | 3.2255 | 16 |
| 0.90 | 0.98 | $10^{-10}$ | 0.004 | 1.0 | **1000** | 0.5 | 3.2175 | 17 |
| 0.90 | 0.98 | $10^{-10}$ | 0.004 | 1.0 | **2000** | 0.5 | 3.2166 | 18 |
| 0.90 | 0.98 | $10^{-10}$ | 0.004 | 1.0 | 1000 | **0.0** | 3.2195 | 19 |
| 0.90 | 0.98 | $10^{-10}$ | 0.004 | 1.0 | 1000 | **0.5** | 3.2175 | 20 |
| 0.90 | 0.98 | $10^{-10}$ | 0.004 | 1.0 | 1000 | **1.0** | 3.4953 | 21 |

*Table 60.* Hyperparameter ablation for AdamW on 350M on 1× Chinchilla Data (FP8 weights + FP8 compute)

| $\beta_1$ | $\beta_2$ | $\epsilon$ | $\eta$ | $g_{\text{norm}}$ | warmup | $\lambda$ | Loss | Link |
|---|---|---|---|---|---|---|---|---|
| 0.90 | 0.95 | $10^{-20}$ | 0.008 | 1.0 | 2000 | 0.5 | 3.0761 | 1 |
| **0.87** | 0.95 | $10^{-10}$ | 0.008 | 1.0 | 1000 | 0.5 | 3.1546 | 2 |
| **0.90** | 0.95 | $10^{-10}$ | 0.008 | 1.0 | 1000 | 0.5 | 3.1134 | 3 |
| **0.95** | 0.95 | $10^{-10}$ | 0.008 | 1.0 | 1000 | 0.5 | 3.2110 | 4 |
| **0.98** | 0.95 | $10^{-10}$ | 0.008 | 1.0 | 1000 | 0.5 | 3.1655 | 5 |
| 0.90 | **0.90** | $10^{-10}$ | 0.008 | 1.0 | 1000 | 0.5 | 3.1345 | 6 |
| 0.90 | **0.95** | $10^{-10}$ | 0.008 | 1.0 | 1000 | 0.5 | 3.1134 | 7 |
| 0.90 | **0.98** | $10^{-10}$ | 0.008 | 1.0 | 1000 | 0.5 | 3.1157 | 8 |
| 0.90 | 0.95 | $10^{-10}$ | 0.008 | 1.0 | 1000 | 0.5 | 3.1134 | 9 |
| 0.90 | 0.95 | $10^{-20}$ | 0.008 | 1.0 | 1000 | 0.5 | 3.0836 | 10 |
| 0.90 | 0.95 | $10^{-20}$ | **0.004** | 1.0 | 1000 | 0.5 | 3.1237 | 11 |
| 0.90 | 0.95 | $10^{-20}$ | **0.008** | 1.0 | 1000 | 0.5 | 3.0836 | 12 |
| 0.90 | 0.95 | $10^{-20}$ | **0.016** | 1.0 | 1000 | 0.5 | 3.1297 | 13 |
| 0.90 | 0.95 | $10^{-20}$ | 0.008 | **0.0** | 1000 | 0.5 | 3.1288 | 14 |
| 0.90 | 0.95 | $10^{-20}$ | 0.008 | **1.0** | 1000 | 0.5 | 3.0836 | 15 |
| 0.90 | 0.95 | $10^{-20}$ | 0.008 | **2.0** | 1000 | 0.5 | 3.1023 | 16 |
| 0.90 | 0.95 | $10^{-20}$ | 0.008 | 1.0 | **1000** | 0.5 | 3.0836 | 17 |
| 0.90 | 0.95 | $10^{-20}$ | 0.008 | 1.0 | **2000** | 0.5 | 3.0761 | 18 |
| 0.90 | 0.95 | $10^{-20}$ | 0.008 | 1.0 | 2000 | **0.0** | 3.0981 | 19 |
| 0.90 | 0.95 | $10^{-20}$ | 0.008 | 1.0 | 2000 | **0.5** | 3.0761 | 20 |
| 0.90 | 0.95 | $10^{-20}$ | 0.008 | 1.0 | 2000 | **1.0** | 5.1046 | 21 |

*Table 61.* Hyperparameter ablation for M+Adam on 60M on 1× Chinchilla Data (FP8 weights + FP8 compute)

| $\beta_1$ | $\beta_2$ | $\epsilon$ | $\eta_m$ | $\eta_a$ | warmup | $\lambda_m$ | $\lambda_e$ | Loss | Link |
|---|---|---|---|---|---|---|---|---|---|
| 0.87 | 0.98 | $10^{-20}$ | 0.004 | $8 \times 10^{-5}$ | 2000 | 0.1 | 0.01 | 3.4172 | 1 |
| 0.87 | 0.98 | $10^{-10}$ | 0.004 | $8 \times 10^{-5}$ | 1000 | 0.1 | 0.01 | 3.4282 | 2 |
| **0.87** | 0.98 | $10^{-10}$ | 0.004 | $8 \times 10^{-5}$ | 1000 | 0.1 | 0.01 | 3.4282 | 3 |
| **0.90** | 0.98 | $10^{-10}$ | 0.004 | $8 \times 10^{-5}$ | 1000 | 0.1 | 0.01 | 3.4426 | 4 |
| **0.95** | 0.98 | $10^{-10}$ | 0.004 | $8 \times 10^{-5}$ | 1000 | 0.1 | 0.01 | 3.5539 | 5 |
| **0.98** | 0.98 | $10^{-10}$ | 0.004 | $8 \times 10^{-5}$ | 1000 | 0.1 | 0.01 | 3.7322 | 6 |
| 0.87 | **0.90** | $10^{-10}$ | 0.004 | $8 \times 10^{-5}$ | 1000 | 0.1 | 0.01 | 3.4673 | 7 |
| 0.87 | **0.95** | $10^{-10}$ | 0.004 | $8 \times 10^{-5}$ | 1000 | 0.1 | 0.01 | 3.4376 | 8 |
| 0.87 | **0.98** | $10^{-10}$ | 0.004 | $8 \times 10^{-5}$ | 1000 | 0.1 | 0.01 | 3.4282 | 9 |
| 0.87 | 0.98 | $10^{-10}$ | 0.004 | $8 \times 10^{-5}$ | 1000 | 0.1 | 0.01 | 3.4282 | 10 |
| 0.87 | 0.98 | $10^{-20}$ | 0.004 | $8 \times 10^{-5}$ | 1000 | 0.1 | 0.01 | 3.4233 | 11 |
| 0.87 | 0.98 | $10^{-20}$ | **0.004** | $8 \times 10^{-5}$ | 1000 | 0.1 | 0.01 | 3.4233 | 12 |
| 0.87 | 0.98 | $10^{-20}$ | **0.008** | $8 \times 10^{-5}$ | 1000 | 0.1 | 0.01 | 3.4930 | 13 |
| 0.87 | 0.98 | $10^{-20}$ | **0.016** | $8 \times 10^{-5}$ | 1000 | 0.1 | 0.01 | 3.5640 | 14 |
| 0.87 | 0.98 | $10^{-20}$ | 0.004 | $8 \times 10^{-5}$ | 1000 | 0.1 | 0.01 | 3.4233 | 15 |
| 0.87 | 0.98 | $10^{-20}$ | 0.004 | $1.6 \times 10^{-4}$ | 1000 | 0.1 | 0.01 | 3.4211 | 16 |
| 0.87 | 0.98 | $10^{-20}$ | 0.004 | $3.2 \times 10^{-4}$ | 1000 | 0.1 | 0.01 | 3.4245 | 17 |
| 0.87 | 0.98 | $10^{-20}$ | 0.004 | $8 \times 10^{-5}$ | **1000** | 0.1 | 0.01 | 3.4233 | 18 |
| 0.87 | 0.98 | $10^{-20}$ | 0.004 | $8 \times 10^{-5}$ | **2000** | 0.1 | 0.01 | 3.4172 | 19 |
| 0.87 | 0.98 | $10^{-20}$ | 0.004 | $8 \times 10^{-5}$ | 2000 | **0.0** | 0.01 | 3.4204 | 20 |
| 0.87 | 0.98 | $10^{-20}$ | 0.004 | $8 \times 10^{-5}$ | 2000 | **0.1** | 0.01 | 3.4172 | 21 |
| 0.87 | 0.98 | $10^{-20}$ | 0.004 | $8 \times 10^{-5}$ | 2000 | **0.2** | 0.01 | 3.4263 | 22 |
| 0.87 | 0.98 | $10^{-20}$ | 0.004 | $8 \times 10^{-5}$ | 2000 | 0.1 | **0.0** | 3.4172 | 23 |
| 0.87 | 0.98 | $10^{-20}$ | 0.004 | $8 \times 10^{-5}$ | 2000 | 0.1 | **0.01** | 3.4172 | 24 |
| 0.87 | 0.98 | $10^{-20}$ | 0.004 | $8 \times 10^{-5}$ | 2000 | 0.1 | **0.02** | 3.4171 | 25 |

*Table 62.* Hyperparameter ablation for M+Adam on 60M on 4× Chinchilla Data (FP8 weights + FP8 compute)

| $\beta_1$ | $\beta_2$ | $\epsilon$ | $\eta_m$ | $\eta_a$ | warmup | $\lambda_m$ | $\lambda_e$ | Loss | Link |
|---|---|---|---|---|---|---|---|---|---|
| 0.87 | 0.98 | $10^{-10}$ | 0.004 | $8 \times 10^{-5}$ | 2000 | 0.1 | 0.01 | 3.3085 | 1 |
| 0.87 | 0.98 | $10^{-10}$ | 0.004 | $8 \times 10^{-5}$ | 1000 | 0.1 | 0.01 | 3.3127 | 2 |
| **0.87** | 0.98 | $10^{-10}$ | 0.004 | $8 \times 10^{-5}$ | 1000 | 0.1 | 0.01 | 3.3127 | 3 |
| **0.90** | 0.98 | $10^{-10}$ | 0.004 | $8 \times 10^{-5}$ | 1000 | 0.1 | 0.01 | 3.3165 | 4 |
| **0.95** | 0.98 | $10^{-10}$ | 0.004 | $8 \times 10^{-5}$ | 1000 | 0.1 | 0.01 | 3.4342 | 5 |
| **0.98** | 0.98 | $10^{-10}$ | 0.004 | $8 \times 10^{-5}$ | 1000 | 0.1 | 0.01 | 3.5365 | 6 |
| 0.87 | **0.90** | $10^{-10}$ | 0.004 | $8 \times 10^{-5}$ | 1000 | 0.1 | 0.01 | 3.3564 | 7 |
| 0.87 | **0.95** | $10^{-10}$ | 0.004 | $8 \times 10^{-5}$ | 1000 | 0.1 | 0.01 | 3.3207 | 8 |
| 0.87 | **0.98** | $10^{-10}$ | 0.004 | $8 \times 10^{-5}$ | 1000 | 0.1 | 0.01 | 3.3127 | 9 |
| 0.87 | 0.98 | $10^{-10}$ | 0.004 | $8 \times 10^{-5}$ | 1000 | 0.1 | 0.01 | 3.3127 | 10 |
| 0.87 | 0.98 | $10^{-20}$ | 0.004 | $8 \times 10^{-5}$ | 1000 | 0.1 | 0.01 | 3.3108 | 11 |
| 0.87 | 0.98 | $10^{-10}$ | **0.004** | $8 \times 10^{-5}$ | 1000 | 0.1 | 0.01 | 3.3127 | 12 |
| 0.87 | 0.98 | $10^{-10}$ | **0.008** | $8 \times 10^{-5}$ | 1000 | 0.1 | 0.01 | 3.3260 | 13 |
| 0.87 | 0.98 | $10^{-10}$ | **0.016** | $8 \times 10^{-5}$ | 1000 | 0.1 | 0.01 | 3.3361 | 14 |
| 0.87 | 0.98 | $10^{-10}$ | 0.004 | $8 \times 10^{-5}$ | 1000 | 0.1 | 0.01 | 3.3127 | 15 |
| 0.87 | 0.98 | $10^{-10}$ | 0.004 | $1.6 \times 10^{-4}$ | 1000 | 0.1 | 0.01 | 3.3116 | 16 |
| 0.87 | 0.98 | $10^{-10}$ | 0.004 | $3.2 \times 10^{-4}$ | 1000 | 0.1 | 0.01 | 3.3295 | 17 |
| 0.87 | 0.98 | $10^{-10}$ | 0.004 | $8 \times 10^{-5}$ | **1000** | 0.1 | 0.01 | 3.3127 | 18 |
| 0.87 | 0.98 | $10^{-10}$ | 0.004 | $8 \times 10^{-5}$ | **2000** | 0.1 | 0.01 | 3.3085 | 19 |
| 0.87 | 0.98 | $10^{-10}$ | 0.004 | $8 \times 10^{-5}$ | 2000 | **0.0** | 0.01 | 3.3177 | 20 |
| 0.87 | 0.98 | $10^{-10}$ | 0.004 | $8 \times 10^{-5}$ | 2000 | **0.1** | 0.01 | 3.3085 | 21 |
| 0.87 | 0.98 | $10^{-10}$ | 0.004 | $8 \times 10^{-5}$ | 2000 | **0.2** | 0.01 | 3.3310 | 22 |
| 0.87 | 0.98 | $10^{-10}$ | 0.004 | $8 \times 10^{-5}$ | 2000 | 0.1 | **0.0** | 3.3069 | 23 |
| 0.87 | 0.98 | $10^{-10}$ | 0.004 | $8 \times 10^{-5}$ | 2000 | 0.1 | **0.01** | 3.3085 | 24 |

*Table 63.* Hyperparameter ablation for M+Adam on 60M on $8\times$ Chinchilla Data (FP8 weights + FP8 compute)

| $\beta_1$ | $\beta_2$ | $\epsilon$ | $\eta_m$ | $\eta_a$ | warmup | $\lambda_m$ | $\lambda_e$ | Loss | Link |
|---|---|---|---|---|---|---|---|---|---|
| 0.90 | 0.98 | $10^{-10}$ | 0.008 | $8\times10^{-5}$ | 1000 | 0.0 | 0.01 | 3.2437 | 1 |
| 0.87 | 0.98 | $10^{-10}$ | 0.004 | $8\times10^{-5}$ | 1000 | 0.1 | 0.01 | 3.2835 | 2 |
| **0.87** | 0.98 | $10^{-10}$ | 0.004 | $8\times10^{-5}$ | 1000 | 0.1 | 0.01 | 3.2835 | 3 |
| **0.90** | 0.98 | $10^{-10}$ | 0.004 | $8\times10^{-5}$ | 1000 | 0.1 | 0.01 | 3.2790 | 4 |
| **0.95** | 0.98 | $10^{-10}$ | 0.004 | $8\times10^{-5}$ | 1000 | 0.1 | 0.01 | 3.3879 | 5 |
| **0.98** | 0.98 | $10^{-10}$ | 0.004 | $8\times10^{-5}$ | 1000 | 0.1 | 0.01 | 3.5009 | 6 |
| 0.90 | **0.90** | $10^{-10}$ | 0.004 | $8\times10^{-5}$ | 1000 | 0.1 | 0.01 | 3.3509 | 7 |
| 0.90 | **0.95** | $10^{-10}$ | 0.004 | $8\times10^{-5}$ | 1000 | 0.1 | 0.01 | 3.3031 | 8 |
| 0.90 | **0.98** | $10^{-10}$ | 0.004 | $8\times10^{-5}$ | 1000 | 0.1 | 0.01 | 3.2790 | 9 |
| 0.90 | 0.98 | $10^{-10}$ | 0.004 | $8\times10^{-5}$ | 1000 | 0.1 | 0.01 | 3.2790 | 10 |
| 0.90 | 0.98 | $10^{-20}$ | 0.004 | $8\times10^{-5}$ | 1000 | 0.1 | 0.01 | 3.2802 | 11 |
| 0.90 | 0.98 | $10^{-10}$ | **0.004** | $8\times10^{-5}$ | 1000 | 0.1 | 0.01 | 3.2790 | 12 |
| 0.90 | 0.98 | $10^{-10}$ | **0.008** | $8\times10^{-5}$ | 1000 | 0.1 | 0.01 | 3.2468 | 13 |
| 0.90 | 0.98 | $10^{-10}$ | **0.016** | $8\times10^{-5}$ | 1000 | 0.1 | 0.01 | 3.2681 | 14 |
| 0.90 | 0.98 | $10^{-10}$ | 0.008 | $8\times10^{-5}$ | 1000 | 0.1 | 0.01 | 3.2468 | 15 |
| 0.90 | 0.98 | $10^{-10}$ | 0.008 | $1.6\times10^{-4}$ | 1000 | 0.1 | 0.01 | 3.2469 | 16 |
| 0.90 | 0.98 | $10^{-10}$ | 0.008 | $3.2\times10^{-4}$ | 1000 | 0.1 | 0.01 | 3.2527 | 17 |
| 0.90 | 0.98 | $10^{-10}$ | 0.008 | $8\times10^{-5}$ | **1000** | 0.1 | 0.01 | 3.2468 | 18 |
| 0.90 | 0.98 | $10^{-10}$ | 0.008 | $8\times10^{-5}$ | **2000** | 0.1 | 0.01 | 3.2449 | 19 |
| 0.90 | 0.98 | $10^{-10}$ | 0.008 | $8\times10^{-5}$ | 1000 | **0.0** | 0.01 | 3.2437 | 20 |
| 0.90 | 0.98 | $10^{-10}$ | 0.008 | $8\times10^{-5}$ | 1000 | **0.1** | 0.01 | 3.2468 | 21 |
| 0.90 | 0.98 | $10^{-10}$ | 0.008 | $8\times10^{-5}$ | 1000 | **0.2** | 0.01 | 3.2781 | 22 |
| 0.90 | 0.98 | $10^{-10}$ | 0.008 | $8\times10^{-5}$ | 1000 | 0.0 | **0.0** | 3.2447 | 23 |
| 0.90 | 0.98 | $10^{-10}$ | 0.008 | $8\times10^{-5}$ | 1000 | 0.0 | **0.01** | 3.2437 | 24 |
| 0.90 | 0.98 | $10^{-10}$ | 0.008 | $8\times10^{-5}$ | 1000 | 0.0 | **0.02** | 3.2447 | 25 |

*Table 64.* Hyperparameter ablation for M+Adam on 130M on $1\times$ Chinchilla Data (FP8 weights + FP8 compute)

| $\beta_1$ | $\beta_2$ | $\epsilon$ | $\eta_m$ | $\eta_a$ | warmup | $\lambda_m$ | $\lambda_e$ | Loss | Link |
|---|---|---|---|---|---|---|---|---|---|
| 0.90 | 0.98 | $10^{-10}$ | 0.004 | $1.6\times10^{-4}$ | 1000 | 0.1 | 0.01 | 3.1898 | 1 |
| 0.87 | 0.98 | $10^{-10}$ | 0.004 | $3.2\times10^{-4}$ | 1000 | 0.1 | 0.01 | 3.2076 | 2 |
| **0.87** | 0.98 | $10^{-10}$ | 0.004 | $3.2\times10^{-4}$ | 1000 | 0.1 | 0.01 | 3.2076 | 3 |
| **0.90** | 0.98 | $10^{-10}$ | 0.004 | $3.2\times10^{-4}$ | 1000 | 0.1 | 0.01 | 3.2037 | 4 |
| **0.95** | 0.98 | $10^{-10}$ | 0.004 | $3.2\times10^{-4}$ | 1000 | 0.1 | 0.01 | 3.2789 | 5 |
| **0.98** | 0.98 | $10^{-10}$ | 0.004 | $3.2\times10^{-4}$ | 1000 | 0.1 | 0.01 | 3.3689 | 6 |
| 0.90 | **0.90** | $10^{-10}$ | 0.004 | $3.2\times10^{-4}$ | 1000 | 0.1 | 0.01 | 3.2513 | 7 |
| 0.90 | **0.95** | $10^{-10}$ | 0.004 | $3.2\times10^{-4}$ | 1000 | 0.1 | 0.01 | 3.2153 | 8 |
| 0.90 | **0.98** | $10^{-10}$ | 0.004 | $3.2\times10^{-4}$ | 1000 | 0.1 | 0.01 | 3.2037 | 9 |
| 0.90 | 0.98 | $10^{-10}$ | 0.004 | $3.2\times10^{-4}$ | 1000 | 0.1 | 0.01 | 3.2037 | 10 |
| 0.90 | 0.98 | $10^{-20}$ | 0.004 | $3.2\times10^{-4}$ | 1000 | 0.1 | 0.01 | 3.2030 | 11 |
| 0.90 | 0.98 | $10^{-10}$ | **0.004** | $3.2\times10^{-4}$ | 1000 | 0.1 | 0.01 | 3.2037 | 12 |
| 0.90 | 0.98 | $10^{-10}$ | **0.008** | $3.2\times10^{-4}$ | 1000 | 0.1 | 0.01 | 3.2495 | 13 |
| 0.90 | 0.98 | $10^{-10}$ | **0.016** | $3.2\times10^{-4}$ | 1000 | 0.1 | 0.01 | 3.3385 | 14 |
| 0.90 | 0.98 | $10^{-10}$ | 0.004 | $8\times10^{-5}$ | 1000 | 0.1 | 0.01 | 3.1901 | 15 |
| 0.90 | 0.98 | $10^{-10}$ | 0.004 | $1.6\times10^{-4}$ | 1000 | 0.1 | 0.01 | 3.1898 | 16 |
| 0.90 | 0.98 | $10^{-10}$ | 0.004 | $3.2\times10^{-4}$ | 1000 | 0.1 | 0.01 | 3.2037 | 17 |
| 0.90 | 0.98 | $10^{-10}$ | 0.004 | $1.6\times10^{-4}$ | **1000** | 0.1 | 0.01 | 3.1898 | 18 |
| 0.90 | 0.98 | $10^{-10}$ | 0.004 | $1.6\times10^{-4}$ | **2000** | 0.1 | 0.01 | 3.1874 | 19 |
| 0.90 | 0.98 | $10^{-10}$ | 0.004 | $1.6\times10^{-4}$ | 1000 | **0.0** | 0.01 | 3.1959 | 20 |
| 0.90 | 0.98 | $10^{-10}$ | 0.004 | $1.6\times10^{-4}$ | 1000 | **0.1** | 0.01 | 3.1898 | 21 |
| 0.90 | 0.98 | $10^{-10}$ | 0.004 | $1.6\times10^{-4}$ | 1000 | **0.2** | 0.01 | 3.2075 | 22 |
| 0.90 | 0.98 | $10^{-10}$ | 0.004 | $1.6\times10^{-4}$ | 1000 | 0.1 | **0.0** | 3.1913 | 23 |
| 0.90 | 0.98 | $10^{-10}$ | 0.004 | $1.6\times10^{-4}$ | 1000 | 0.1 | **0.01** | 3.1898 | 24 |
| 0.90 | 0.98 | $10^{-10}$ | 0.004 | $1.6\times10^{-4}$ | 1000 | 0.1 | **0.02** | 3.1949 | 25 |

*Table 65.* Hyperparameter ablation for M+Adam on 350M on $1\times$ Chinchilla Data (FP8 weights + FP8 compute)

| $\beta_1$ | $\beta_2$ | $\epsilon$ | $\eta_m$ | $\eta_a$ | warmup | $\lambda_m$ | $\lambda_e$ | Loss | Link |
|---|---|---|---|---|---|---|---|---|---|
| 0.90 | 0.95 | $10^{-10}$ | 0.004 | $8 \times 10^{-5}$ | 1000 | 0.1 | 0.01 | 2.9650 | 1 |
| 0.87 | 0.98 | $10^{-10}$ | 0.004 | $3.2 \times 10^{-4}$ | 1000 | 0.1 | 0.01 | 3.0310 | 2 |
| **0.87** | 0.98 | $10^{-10}$ | 0.004 | $3.2 \times 10^{-4}$ | 1000 | 0.1 | 0.01 | 3.0310 | 3 |
| **0.90** | 0.98 | $10^{-10}$ | 0.004 | $3.2 \times 10^{-4}$ | 1000 | 0.1 | 0.01 | 2.9871 | 4 |
| **0.95** | 0.98 | $10^{-10}$ | 0.004 | $3.2 \times 10^{-4}$ | 1000 | 0.1 | 0.01 | 3.0151 | 5 |
| **0.98** | 0.98 | $10^{-10}$ | 0.004 | $3.2 \times 10^{-4}$ | 1000 | 0.1 | 0.01 | 3.1459 | 6 |
| 0.90 | **0.90** | $10^{-10}$ | 0.004 | $3.2 \times 10^{-4}$ | 1000 | 0.1 | 0.01 | 3.0175 | 7 |
| 0.90 | **0.95** | $10^{-10}$ | 0.004 | $3.2 \times 10^{-4}$ | 1000 | 0.1 | 0.01 | 2.9818 | 8 |
| 0.90 | **0.98** | $10^{-10}$ | 0.004 | $3.2 \times 10^{-4}$ | 1000 | 0.1 | 0.01 | 2.9871 | 9 |
| 0.90 | 0.95 | $10^{-10}$ | 0.004 | $3.2 \times 10^{-4}$ | 1000 | 0.1 | 0.01 | 2.9818 | 10 |
| 0.90 | 0.95 | $10^{-20}$ | 0.004 | $3.2 \times 10^{-4}$ | 1000 | 0.1 | 0.01 | 3.0006 | 11 |
| 0.90 | 0.95 | $10^{-10}$ | **0.004** | $3.2 \times 10^{-4}$ | 1000 | 0.1 | 0.01 | 2.9818 | 12 |
| 0.90 | 0.95 | $10^{-10}$ | **0.008** | $3.2 \times 10^{-4}$ | 1000 | 0.1 | 0.01 | 3.0581 | 13 |
| 0.90 | 0.95 | $10^{-10}$ | **0.016** | $3.2 \times 10^{-4}$ | 1000 | 0.1 | 0.01 | 3.1782 | 14 |
| 0.90 | 0.95 | $10^{-10}$ | 0.004 | $8 \times 10^{-5}$ | 1000 | 0.1 | 0.01 | 2.9650 | 15 |
| 0.90 | 0.95 | $10^{-10}$ | 0.004 | $1.6 \times 10^{-4}$ | 1000 | 0.1 | 0.01 | 2.9675 | 16 |
| 0.90 | 0.95 | $10^{-10}$ | 0.004 | $3.2 \times 10^{-4}$ | 1000 | 0.1 | 0.01 | 2.9818 | 17 |
| 0.90 | 0.95 | $10^{-10}$ | 0.004 | $8 \times 10^{-5}$ | **1000** | 0.1 | 0.01 | 2.9650 | 18 |
| 0.90 | 0.95 | $10^{-10}$ | 0.004 | $8 \times 10^{-5}$ | **2000** | 0.1 | 0.01 | 2.9633 | 19 |
| 0.90 | 0.95 | $10^{-10}$ | 0.004 | $8 \times 10^{-5}$ | 1000 | **0.0** | 0.01 | 2.9620 | 20 |
| 0.90 | 0.95 | $10^{-10}$ | 0.004 | $8 \times 10^{-5}$ | 1000 | **0.1** | 0.01 | 2.9650 | 21 |
| 0.90 | 0.95 | $10^{-10}$ | 0.004 | $8 \times 10^{-5}$ | 1000 | **0.2** | 0.01 | 2.9900 | 22 |
| 0.90 | 0.95 | $10^{-10}$ | 0.004 | $8 \times 10^{-5}$ | 1000 | 0.1 | **0.0** | 2.9640 | 23 |
| 0.90 | 0.95 | $10^{-10}$ | 0.004 | $8 \times 10^{-5}$ | 1000 | 0.1 | **0.01** | 2.9650 | 24 |
| 0.90 | 0.95 | $10^{-10}$ | 0.004 | $8 \times 10^{-5}$ | 1000 | 0.1 | **0.02** | 2.9652 | 25 |

*Table 66.* Hyperparameter ablation for AdamW on 60M on $1\times$ Chinchilla Data (FP32 weights + TF32 compute)

| $\beta_1$ | $\beta_2$ | $\epsilon$ | $\eta$ | $g_{\text{norm}}$ | warmup | $\lambda$ | Loss | Link |
|---|---|---|---|---|---|---|---|---|
| 0.90 | 0.98 | $10^{-10}$ | 0.004 | 1.0 | 1000 | 0.0 | 3.3689 | 1 |
| 0.90 | 0.95 | $10^{-10}$ | 0.004 | 1.0 | 1000 | 0.5 | 3.4191 | 2 |
| **0.87** | 0.95 | $10^{-10}$ | 0.004 | 1.0 | 1000 | 0.5 | 3.4211 | 3 |
| **0.90** | 0.95 | $10^{-10}$ | 0.004 | 1.0 | 1000 | 0.5 | 3.4191 | 4 |
| **0.95** | 0.95 | $10^{-10}$ | 0.004 | 1.0 | 1000 | 0.5 | 3.4289 | 5 |
| **0.98** | 0.95 | $10^{-10}$ | 0.004 | 1.0 | 1000 | 0.5 | 3.4664 | 6 |
| 0.90 | **0.90** | $10^{-10}$ | 0.004 | 1.0 | 1000 | 0.5 | 3.4257 | 7 |
| 0.90 | **0.95** | $10^{-10}$ | 0.004 | 1.0 | 1000 | 0.5 | 3.4191 | 8 |
| 0.90 | **0.98** | $10^{-10}$ | 0.004 | 1.0 | 1000 | 0.5 | 3.4157 | 9 |
| 0.90 | 0.98 | $10^{-10}$ | 0.004 | 1.0 | 1000 | 0.5 | 3.4157 | 10 |
| 0.90 | 0.98 | $10^{-20}$ | 0.004 | 1.0 | 1000 | 0.5 | 3.4154 | 11 |
| 0.90 | 0.98 | $10^{-10}$ | **0.004** | 1.0 | 1000 | 0.5 | 3.4157 | 12 |
| 0.90 | 0.98 | $10^{-10}$ | **0.008** | 1.0 | 1000 | 0.5 | 3.4562 | 13 |
| 0.90 | 0.98 | $10^{-10}$ | **0.016** | 1.0 | 1000 | 0.5 | 3.5441 | 14 |
| 0.90 | 0.98 | $10^{-10}$ | 0.004 | **0.0** | 1000 | 0.5 | 3.4176 | 15 |
| 0.90 | 0.98 | $10^{-10}$ | 0.004 | **1.0** | 1000 | 0.5 | 3.4157 | 16 |
| 0.90 | 0.98 | $10^{-10}$ | 0.004 | **2.0** | 1000 | 0.5 | 3.4169 | 17 |
| 0.90 | 0.98 | $10^{-10}$ | 0.004 | 1.0 | **500** | 0.5 | 3.4191 | 18 |
| 0.90 | 0.98 | $10^{-10}$ | 0.004 | 1.0 | **1000** | 0.5 | 3.4157 | 19 |
| 0.90 | 0.98 | $10^{-10}$ | 0.004 | 1.0 | **2000** | 0.5 | 3.4182 | 20 |
| 0.90 | 0.98 | $10^{-10}$ | 0.004 | 1.0 | 1000 | **0.0** | 3.3689 | 21 |
| 0.90 | 0.98 | $10^{-10}$ | 0.004 | 1.0 | 1000 | **0.5** | 3.4157 | 22 |
| 0.90 | 0.98 | $10^{-10}$ | 0.004 | 1.0 | 1000 | **1.0** | 3.5522 | 23 |

*Table 67.* Hyperparameter ablation for AdamW on 130M on $1\times$ Chinchilla Data (FP32 weights + TF32 compute)

| $\beta_1$ | $\beta_2$ | $\epsilon$ | $\eta$ | $g_{\text{norm}}$ | warmup | $\lambda$ | Loss | Link |
|---|---|---|---|---|---|---|---|---|
| 0.90 | 0.95 | $10^{-10}$ | 0.004 | 1.0 | 1000 | 0.0 | 3.1186 | 1 |
| 0.90 | 0.95 | $10^{-10}$ | 0.004 | 1.0 | 1000 | 0.5 | 3.1611 | 2 |
| **0.87** | 0.95 | $10^{-10}$ | 0.004 | 1.0 | 1000 | 0.5 | 3.1628 | 3 |
| **0.90** | 0.95 | $10^{-10}$ | 0.004 | 1.0 | 1000 | 0.5 | 3.1611 | 4 |
| **0.95** | 0.95 | $10^{-10}$ | 0.004 | 1.0 | 1000 | 0.5 | 3.1747 | 5 |
| **0.98** | 0.95 | $10^{-10}$ | 0.004 | 1.0 | 1000 | 0.5 | 3.1913 | 6 |
| 0.90 | **0.90** | $10^{-10}$ | 0.004 | 1.0 | 1000 | 0.5 | 3.1676 | 7 |
| 0.90 | **0.95** | $10^{-10}$ | 0.004 | 1.0 | 1000 | 0.5 | 3.1611 | 8 |
| 0.90 | **0.98** | $10^{-10}$ | 0.004 | 1.0 | 1000 | 0.5 | 3.1589 | 9 |
| 0.90 | 0.95 | $10^{-10}$ | 0.004 | 1.0 | 1000 | 0.5 | 3.1611 | 10 |
| 0.90 | 0.95 | $10^{-20}$ | 0.004 | 1.0 | 1000 | 0.5 | 3.1610 | 11 |
| 0.90 | 0.95 | $10^{-10}$ | **0.004** | 1.0 | 1000 | 0.5 | 3.1611 | 12 |
| 0.90 | 0.95 | $10^{-10}$ | **0.008** | 1.0 | 1000 | 0.5 | 3.2240 | 13 |
| 0.90 | 0.95 | $10^{-10}$ | **0.016** | 1.0 | 1000 | 0.5 | 3.3167 | 14 |
| 0.90 | 0.95 | $10^{-10}$ | 0.004 | **0.0** | 1000 | 0.5 | 3.1643 | 15 |
| 0.90 | 0.95 | $10^{-10}$ | 0.004 | **1.0** | 1000 | 0.5 | 3.1611 | 16 |
| 0.90 | 0.95 | $10^{-10}$ | 0.004 | **2.0** | 1000 | 0.5 | 3.1643 | 17 |
| 0.90 | 0.95 | $10^{-10}$ | 0.004 | 1.0 | **500** | 0.5 | 3.1673 | 18 |
| 0.90 | 0.95 | $10^{-10}$ | 0.004 | 1.0 | **1000** | 0.5 | 3.1611 | 19 |
| 0.90 | 0.95 | $10^{-10}$ | 0.004 | 1.0 | **2000** | 0.5 | 3.1610 | 20 |
| 0.90 | 0.95 | $10^{-10}$ | 0.004 | 1.0 | 1000 | **0.0** | 3.1186 | 21 |
| 0.90 | 0.95 | $10^{-10}$ | 0.004 | 1.0 | 1000 | **0.5** | 3.1611 | 22 |
| 0.90 | 0.95 | $10^{-10}$ | 0.004 | 1.0 | 1000 | **1.0** | 3.3078 | 23 |

*Table 68.* Hyperparameter ablation for AdamW + SR on 60M on $2\times$ Chinchilla Data (BF16 weights + BF16 compute)

| $\beta_1$ | $\beta_2$ | $\epsilon$ | $\eta$ | $g_{\text{norm}}$ | warmup | $\lambda$ | Loss | Link |
|---|---|---|---|---|---|---|---|---|
| 0.90 | 0.98 | $10^{-10}$ | 0.004 | 1.0 | 2000 | 0.2 | 3.2849 | 1 |
| 0.90 | 0.95 | $10^{-10}$ | 0.004 | 1.0 | 1000 | 0.2 | 3.2958 | 2 |
| **0.87** | 0.95 | $10^{-10}$ | 0.004 | 1.0 | 1000 | 0.2 | 3.2930 | 3 |
| **0.90** | 0.95 | $10^{-10}$ | 0.004 | 1.0 | 1000 | 0.2 | 3.2958 | 4 |
| **0.95** | 0.95 | $10^{-10}$ | 0.004 | 1.0 | 1000 | 0.2 | 3.3001 | 5 |
| **0.98** | 0.95 | $10^{-10}$ | 0.004 | 1.0 | 1000 | 0.2 | 3.3318 | 6 |
| 0.90 | **0.90** | $10^{-10}$ | 0.004 | 1.0 | 1000 | 0.2 | 3.3027 | 7 |
| 0.90 | **0.95** | $10^{-10}$ | 0.004 | 1.0 | 1000 | 0.2 | 3.2958 | 8 |
| 0.90 | **0.98** | $10^{-10}$ | 0.004 | 1.0 | 1000 | 0.2 | 3.2884 | 9 |
| 0.90 | 0.98 | $10^{-10}$ | 0.004 | 1.0 | 1000 | 0.2 | 3.2884 | 10 |
| 0.90 | 0.98 | $10^{-20}$ | 0.004 | 1.0 | 1000 | 0.2 | 3.2881 | 11 |
| 0.90 | 0.98 | $10^{-10}$ | **0.004** | 1.0 | 1000 | 0.2 | 3.2884 | 12 |
| 0.90 | 0.98 | $10^{-10}$ | **0.008** | 1.0 | 1000 | 0.2 | 3.3099 | 13 |
| 0.90 | 0.98 | $10^{-10}$ | **0.016** | 1.0 | 1000 | 0.2 | 3.3581 | 14 |
| 0.90 | 0.98 | $10^{-10}$ | 0.004 | **0.0** | 1000 | 0.2 | 3.2896 | 15 |
| 0.90 | 0.98 | $10^{-10}$ | 0.004 | **1.0** | 1000 | 0.2 | 3.2884 | 16 |
| 0.90 | 0.98 | $10^{-10}$ | 0.004 | **2.0** | 1000 | 0.2 | 3.2883 | 17 |
| 0.90 | 0.98 | $10^{-10}$ | 0.004 | 1.0 | **1000** | 0.2 | 3.2884 | 18 |
| 0.90 | 0.98 | $10^{-10}$ | 0.004 | 1.0 | **2000** | 0.2 | 3.2849 | 19 |
| 0.90 | 0.98 | $10^{-10}$ | 0.004 | 1.0 | 2000 | **0.0** | 3.2871 | 20 |
| 0.90 | 0.98 | $10^{-10}$ | 0.004 | 1.0 | 2000 | **0.2** | 3.2849 | 21 |
| 0.90 | 0.98 | $10^{-10}$ | 0.004 | 1.0 | 2000 | **0.4** | 3.3285 | 22 |

*Table 69.* Hyperparameter ablation for AdamW + SR on 60M on $4\times$ Chinchilla Data (BF16 weights + BF16 compute)

| $\beta_1$ | $\beta_2$ | $\epsilon$ | $\eta$ | $g_{\text{norm}}$ | warmup | $\lambda$ | Loss | Link |
|------|------|------------|--------|-------|--------|------|--------|------|
| 0.90 | 0.98 | $10^{-10}$ | 0.004 | 1.0 | 2000 | 0.2 | 3.2290 | 1 |
| 0.90 | 0.95 | $10^{-10}$ | 0.004 | 1.0 | 1000 | 0.2 | 3.2399 | 2 |
| **0.87** | 0.95 | $10^{-10}$ | 0.004 | 1.0 | 1000 | 0.2 | 3.2390 | 3 |
| **0.90** | 0.95 | $10^{-10}$ | 0.004 | 1.0 | 1000 | 0.2 | 3.2399 | 4 |
| **0.95** | 0.95 | $10^{-10}$ | 0.004 | 1.0 | 1000 | 0.2 | 3.2393 | 5 |
| **0.98** | 0.95 | $10^{-10}$ | 0.004 | 1.0 | 1000 | 0.2 | 3.2535 | 6 |
| 0.90 | **0.90** | $10^{-10}$ | 0.004 | 1.0 | 1000 | 0.2 | 3.2475 | 7 |
| 0.90 | **0.95** | $10^{-10}$ | 0.004 | 1.0 | 1000 | 0.2 | 3.2399 | 8 |
| 0.90 | **0.98** | $10^{-10}$ | 0.004 | 1.0 | 1000 | 0.2 | 3.2325 | 9 |
| 0.90 | 0.98 | $10^{-10}$ | 0.004 | 1.0 | 1000 | 0.2 | 3.2325 | 10 |
| 0.90 | 0.98 | $10^{-20}$ | 0.004 | 1.0 | 1000 | 0.2 | 3.2320 | 11 |
| 0.90 | 0.98 | $10^{-10}$ | **0.004** | 1.0 | 1000 | 0.2 | 3.2325 | 12 |
| 0.90 | 0.98 | $10^{-10}$ | **0.008** | 1.0 | 1000 | 0.2 | 3.2549 | 13 |
| 0.90 | 0.98 | $10^{-10}$ | **0.016** | 1.0 | 1000 | 0.2 | 3.3015 | 14 |
| 0.90 | 0.98 | $10^{-10}$ | 0.004 | **0.0** | 1000 | 0.2 | 3.2327 | 15 |
| 0.90 | 0.98 | $10^{-10}$ | 0.004 | **1.0** | 1000 | 0.2 | 3.2325 | 16 |
| 0.90 | 0.98 | $10^{-10}$ | 0.004 | **2.0** | 1000 | 0.2 | 3.2321 | 17 |
| 0.90 | 0.98 | $10^{-10}$ | 0.004 | 1.0 | **1000** | 0.2 | 3.2325 | 18 |
| 0.90 | 0.98 | $10^{-10}$ | 0.004 | 1.0 | **2000** | 0.2 | 3.2290 | 19 |
| 0.90 | 0.98 | $10^{-10}$ | 0.004 | 1.0 | 2000 | **0.0** | 3.2286 | 20 |
| 0.90 | 0.98 | $10^{-10}$ | 0.004 | 1.0 | 2000 | **0.2** | 3.2290 | 21 |
| 0.90 | 0.98 | $10^{-10}$ | 0.004 | 1.0 | 2000 | **0.4** | 3.2788 | 22 |

*Table 70.* Hyperparameter ablation for AdamW + SR on 60M on $8\times$ Chinchilla Data (BF16 weights + BF16 compute)

| $\beta_1$ | $\beta_2$ | $\epsilon$ | $\eta$ | $g_{\text{norm}}$ | warmup | $\lambda$ | Loss | Link |
|------|------|------------|--------|-------|--------|------|--------|------|
| 0.90 | 0.98 | $10^{-10}$ | 0.004 | 1.0 | 2000 | 0.2 | 3.1874 | 1 |
| 0.90 | 0.95 | $10^{-10}$ | 0.004 | 1.0 | 1000 | 0.2 | 3.1996 | 2 |
| **0.87** | 0.95 | $10^{-10}$ | 0.004 | 1.0 | 1000 | 0.2 | 3.1997 | 3 |
| **0.90** | 0.95 | $10^{-10}$ | 0.004 | 1.0 | 1000 | 0.2 | 3.1996 | 4 |
| **0.95** | 0.95 | $10^{-10}$ | 0.004 | 1.0 | 1000 | 0.2 | 3.1969 | 5 |
| **0.98** | 0.95 | $10^{-10}$ | 0.004 | 1.0 | 1000 | 0.2 | 3.2028 | 6 |
| 0.90 | **0.90** | $10^{-10}$ | 0.004 | 1.0 | 1000 | 0.2 | 3.2051 | 7 |
| 0.90 | **0.95** | $10^{-10}$ | 0.004 | 1.0 | 1000 | 0.2 | 3.1996 | 8 |
| 0.90 | **0.98** | $10^{-10}$ | 0.004 | 1.0 | 1000 | 0.2 | 3.1915 | 9 |
| 0.90 | 0.98 | $10^{-10}$ | 0.004 | 1.0 | 1000 | 0.2 | 3.1915 | 10 |
| 0.90 | 0.98 | $10^{-20}$ | 0.004 | 1.0 | 1000 | 0.2 | 3.1915 | 11 |
| 0.90 | 0.98 | $10^{-10}$ | **0.004** | 1.0 | 1000 | 0.2 | 3.1915 | 12 |
| 0.90 | 0.98 | $10^{-10}$ | **0.008** | 1.0 | 1000 | 0.2 | 3.2119 | 13 |
| 0.90 | 0.98 | $10^{-10}$ | **0.016** | 1.0 | 1000 | 0.2 | 3.2557 | 14 |
| 0.90 | 0.98 | $10^{-10}$ | 0.004 | **0.0** | 1000 | 0.2 | 3.1916 | 15 |
| 0.90 | 0.98 | $10^{-10}$ | 0.004 | **1.0** | 1000 | 0.2 | 3.1915 | 16 |
| 0.90 | 0.98 | $10^{-10}$ | 0.004 | **2.0** | 1000 | 0.2 | 3.1908 | 17 |
| 0.90 | 0.98 | $10^{-10}$ | 0.004 | 1.0 | **1000** | 0.2 | 3.1915 | 18 |
| 0.90 | 0.98 | $10^{-10}$ | 0.004 | 1.0 | **2000** | 0.2 | 3.1874 | 19 |
| 0.90 | 0.98 | $10^{-10}$ | 0.004 | 1.0 | 2000 | **0.0** | 3.1854 | 20 |
| 0.90 | 0.98 | $10^{-10}$ | 0.004 | 1.0 | 2000 | **0.2** | 3.1874 | 21 |
| 0.90 | 0.98 | $10^{-10}$ | 0.004 | 1.0 | 2000 | **0.4** | 3.2436 | 22 |

