# OpenReview forum: "M+Adam: Low-Precision Training via Additive–Multiplicative Optimization"
_ICML.cc/2026/Conference — ICML 2026 regular_

### Official Review · Reviewer_ojsS · 2026-03-09

**Soundness:** 3
**Presentation:** 3
**Significance:** 2
**Originality:** 3
**Overall Recommendation:** 3
**Confidence:** 3

**Summary:**

This paper introduces M+Adam, an optimizer for low-precision training. The method decomposes floating-point weights into mantissa and exponent components ($w = m \cdot 2^e$), updating the mantissa additively (like Adam) and the exponent multiplicatively (like Madam). The approach aims to address issues where standard additive updates vanish due to coarse resolution at large magnitudes, while purely multiplicative updates struggle near zero. Theoretical results include a monotone descent guarantee and a quantization-survival condition. Empirically, the paper evaluates M+Adam against AdamW on LLaMA-style language models (up to 350M parameters).

**Compliance With Llm Reviewing Policy:**

Affirmed.

**Final Justification:**

I appreciate the rebuttal, but I keep my score. Since the approach relies on double moments and frequent log and exp computations, the practical overhead kind of defeats the purpose of a low-precision optimizer.

**Key Questions For Authors:**

See Weaknesses.

**Limitations:**

See Weaknesses.

**Strengths And Weaknesses:**

### **Strengths**
- **Format-aligned Design**: The decomposition of weight parameters into mantissa for refinement and exponent for scaling directly utilizes the underlying floating-point architecture.
- **Insightful Diagnostic Experiments**: The toy matrix-fitting study clearly isolates failure modes such as sign-flipping and zero-revival, effectively demonstrating the limitations of purely additive or multiplicative updates.
- **Systematic Hyperparameter Methodology**: The use of a documented coordinate-descent protocol for optimizer comparison ensures that results are based on well-tuned versions of both the proposed method and the baseline.

---

### **Weaknesses**
- **Memory Overhead Contradicts Motivation**: The approach aims to save memory by removing FP32 master weights. However, requiring separate first- and second-moment buffers for both $m$ and $e$ results in a increase in persistent memory overhead compared to standard FP32-master AdamW (Table 7).
- **Limited scale and training horizon**: Low-precision instabilities are most critical at large scales and long durations. The current evaluation caps at 350M parameters (Table 3), failing to demonstrate if advantages persist at 1B+ scales [1,2]. Additionally, the FP8-FP8 regime is only validated at 1× Chinchilla (Table 2), failing to demonstrate if the stability persists over longer horizons (unlike the 8× horizon shown for BF16).
- **Incomplete Baselines**: The paper misses critical recent work such as [3] and does not compare against INT-based or 8-bit/4-bit quantized-state optimizers [4,5]. Discussions also omit recent theoretical and benchmarking perspectives [2,6,7].
- **Unclear Justification for Vanishing Updates**: The argument that Adam's additive step size ($m / \sqrt{v} = \Theta(1)$) vanishes due to coarse resolution requires more empirical evidence. The paper lacks an ablation sweeping significantly larger learning rates ($\eta$) for Adam to prove that preconditioning cannot overcome the precision ceiling.
- **Theoretical Limitations**: Theorem 4.1 applies to standard Gradient Descent rather than Adam with momentum. Theorem 4.2 only considers round-to-nearest (RN) and excludes Stochastic Rounding (SR), nor does it provide the necessary lower-bounds on $\eta$ for standard Adam to succeed.
- **Unexplained Hyperparameter Sensitivity**: As shown in Table 4, M+Adam displays sensitivity patterns (e.g., towards $\epsilon$ but not momentum) that differ markedly from AdamW, yet no theoretical or empirical explanation is provided for these shifts.

[1] Fantastic Pretraining Optimizers and Where to Find Them

[2] Benchmarking Optimizers for Large Language Model Pretraining

[3] Scaling FP8 training to trillion-token LLMs

[4] 8-bit Optimizers via Block-wise Quantization

[5] Memory Efficient Optimizers with 4-bit States

[6] Pushing the Limits of Low-Bit Optimizers: A Focus on EMA Dynamics

[7] A Convergence Analysis of Adaptive Optimizers under Floating-point Quantization

---

> ### Author Rebuttal · Authors · 2026-03-31
>
> We thank the reviewer for the careful reading and constructive feedback. We are glad that the format-aligned design, toy diagnostics, and systematic hyperparameter methodology were found valuable. We address the main concerns below.
> During rebuttal, in response to reviewer requests, we ran and report new 1B scale-extension experiments (Table 6; GeFu, v862, ojsS), throughput / systems measurements (Tables 1–3; GeFu), stronger baselines with AdamW+SR and AdamW+Kahan (Tables 4 and 6; oXdN, ojsS), and selected downstream evaluations (Table 5; oXdN). These are summarized in the following anonymous supplementary document: https://github.com/xyxyxyxy1/icmlre/blob/main/rebuttal_supplementary.pdf
>
> **(1) Memory overhead vs. motivation.**
> We refer to our response to Reviewer oXdN, point (3)
>
> **(2) Limited scale and training horizon.**
> We agree that evaluation beyond 350M is important. At the same time, our current study is broader than a single small-scale comparison: we evaluate 60M / 130M / 350M models, multiple precision regimes, and up to 8x Chinchilla budgets with explicit coordinate-descent tuning rather than lightly tuned baselines. Within this regime, M+Adam consistently improves over AdamW, and the gains persist at longer horizons rather than appearing only in short runs.
> To address scale more directly, we also ran 1B follow-up experiments during rebuttal using configurations extrapolated from the 60M–350M tuning study. The current preliminary validation perplexities are:
> 1B, BF16/BF16: AdamW 15.6, M+Adam 14.1
>
> 1B, BF16/FP8: AdamW 16.3, M+Adam 15.7
> These are rebuttal-time results with extrapolated configurations rather than full 1B sweeps, so we do not present them as definitive large-scale evidence. However, they provide encouraging support that the advantage extends beyond 350M.
> We also acknowledge that FP8/FP8 is currently only evaluated at 1x Chinchilla. In our setup, persistent FP8 weights must be simulated in software, which makes long-horizon FP8/FP8 runs substantially more expensive than the BF16-based settings. We are extending these experiments and will include the longer-horizon FP8/FP8 results in the revision as they complete.
>
> **(3) Incomplete baselines and related work.**
> We thank the reviewer for pointing out these references. We agree that they should be discussed more explicitly, and we will expand the related-work section accordingly. Our work is aimed at a different question from low-bit optimizer-state methods such as [4,5]. Those methods primarily target optimizer-state compression, whereas our focus is on optimization stability when weights themselves are stored and updated persistently in low precision without FP32 master weights. Similarly, [3] focuses on scaling FP8 training pipelines, including architectural and systems considerations, whereas our contribution is at the optimizer-update level: replacing a purely additive update with a mantissa/exponent decomposition better matched to floating-point geometry. We agree that benchmarking/theory papers such as [2,6,7] are highly relevant for positioning our claims, and we will incorporate them more carefully in the revision.
>
> **(4) Empirical support for the vanishing-update explanation.**
> We thank the reviewer for this suggestion. We would like to clarify that larger-learning-rate tests for AdamW in low precision are already included in the appendix. The sweep tables contain runs at substantially larger learning rates than the tuned baseline (e.g., for 60M BF16/BF16: 0.004, 0.008, 0.016), and the sensitivity plots show the same trend across 60M / 130M / 350M and all precision settings. The conclusion is consistent: increasing AdamW’s learning rate beyond the tuned region does not remove the degradation caused by coarse low-precision resolution; it generally worsens validation loss and sometimes causes instability. We will point the reader more explicitly to these appendix results in the revision and surface this ablation more prominently in the main text.
>
> **(5) Theoretical limitations**
> We refer to our response to Reviewer v862, point (1)
>
> **(6) Hyperparameter sensitivity.**
> We do not view the different sensitivity pattern of M+Adam relative to AdamW as unexplained, but as a consequence of optimizing in a different coordinate system. AdamW uses a single additive update in weight space, whereas M+Adam separates mantissa updates for local refinement from exponent updates for scale adjustment. Since these branches play different roles, they need not inherit the same sensitivity pattern as AdamW. In particular, weaker sensitivity to momentum is plausible once scale adjustment is handled explicitly through the exponent branch, while $\epsilon$ remains important because it controls the denominator floor in adaptive normalization. We will add this explanation in the revision.

---

> > ### Author Rebuttal · Reviewer_ojsS · 2026-04-03
> >
> > Thanks for the detailed rebuttal and the supplementary 1B experiments. While the response clarifies several questions, my main concern regarding system-level efficiency remains. The memory and time overhead introduced by the double moments and frequent reparameterize/recombine operations is non-trivial (>30% overhead even at the 350M scale) and will likely become a bottleneck as model sizes scale up. For practical low-precision optimization, finding a way to efficiently leverage modern hardware and techniques to achieve the "best of three worlds" (update sensitivity, memory consumption, and compute efficiency) remains a crucial and unavoidable challenge.

---

> > > ### Author Response · Authors · 2026-04-03
> > >
> > > Thank you for the follow-up and for engaging with the supplementary results.
> > >
> > > We agree that system-level efficiency, that jointly balances update sensitivity, memory, and compute, is an important and non-trivial challenge for practical low-precision training. At the same time, our goal in this work is more focused: to show that optimizer design aligned with floating-point structure can recover stability and effectiveness under persistent low-precision weights, rather than to fully optimize systems efficiency.
> > >
> > > Regarding the observed overhead, we note that the current implementation is a research prototype built with standard PyTorch operators, without fused kernels or hardware-specific optimization. As also reflected in the supplementary results, the relative overhead is already substantially smaller at 1B scale (~6–9% - see Table 1 in the supplementary link) than at 350M. Additionally, we expect further reductions with standard systems optimizations (e.g., fusion or custom kernels).
> > >
> > > We thus view the current trade-off as a first step in demonstrating that stable low-precision training without FP32 master weights is achievable.
> > >
> > > We hope this clarifies the intended scope and helps address the remaining concern.

---

### Official Review · Reviewer_oXdN · 2026-03-11

**Soundness:** 3
**Presentation:** 3
**Significance:** 2
**Originality:** 2
**Overall Recommendation:** 3
**Confidence:** 4

**Summary:**

This paper introduces M+ADAM, a new low-precision training optimizer that decomposes parameters into a mantissa and exponent pair, computing the gradient for each part during every update step. The authors analyze four failure modes and theoretically explain why Adam fails with additive operations under low precision, while proving that M+ADAM can overcome these limitations. In their experiments, a toy model demonstrates that M+ADAM successfully navigates these four failure modes by combining additive and multiplicative updates. Furthermore, across different model sizes (from 60M to 350M parameters), the approach achieves lower perplexity during the pretraining stage using pure low-precision training.

**Compliance With Llm Reviewing Policy:**

Affirmed.

**Final Justification:**

Considering the memory overhead of the proposed approach compared to Kahan summation, I will maintain my current score.

**Key Questions For Authors:**

Could the authors include Kahan summation (e.g., https://optimi.benjaminwarner.dev/kahan_summation/) as an additional experimental baseline?

**Limitations:**

yes

**Strengths And Weaknesses:**

Strength:

1. This paper explicitly analyzes four distinct failure modes encountered during low-precision training. Furthermore, the authors establish a specific mathematical boundary, the critical magnitude threshold ($M_{crit}$), to formally prove exactly when Adam's purely additive updates vanish and fail.

2. The authors propose a novel approach to stabilize low-precision training by decomposing the network weights into a mantissa and an exponent.

Weakness:

1. The baselines presented in the scaling curves (Figure 4) rely solely on standard AdamW in low precision. Since standard AdamW is known to fail in this regime, it serves as a weak baseline. To strengthen the empirical claims, Figure 4 should incorporate stronger baselines, such as AdamW with Stochastic Rounding (which is currently restricted to Table 3 ) and AdamW with Kahan summation.

2. The empirical evaluation currently relies strictly on validation perplexity. The paper lacks comparisons on downstream tasks.

3. The proposed M+ADAM optimizer requires maintaining separate optimizer states (first and second moments) for both the mantissa and the exponent. As the authors acknowledge, this overhead can fully offset the memory savings gained from removing FP32 master weights. There is no memory advantage over standard mixed-precision training.

4. Missing Kahan Summation Baseline: The paper completely omits Kahan summation (compensated summation) as an experimental baseline. Kahan summation utilizes a low-precision (e.g., BF16) compensation weight to accumulate small updates, effectively mitigating the additive failure mode while maintaining a pure low-precision training pipeline.

---

> ### Author Rebuttal · Authors · 2026-03-31
>
> We thank the reviewer for the helpful feedback. We agree that additional baselines and broader evaluation further strengthen the paper, and we address each concern below.
>
> During rebuttal, in response to reviewer requests, we ran and report new 1B scale-extension experiments (Table 6; GeFu, v862, ojsS), throughput / systems measurements (Tables 1–3; GeFu), stronger baselines with AdamW+SR and AdamW+Kahan (Tables 4 and 6; oXdN, ojsS), and selected downstream evaluations (Table 5; oXdN). These are summarized in the following anonymous supplementary document: https://github.com/xyxyxyxy1/icmlre/blob/main/rebuttal_supplementary.pdf
>
> **(1 and 4) Additional baselines.**
> We agree that AdamW alone is not a sufficient reference for Figure 4. Accordingly, we have already added AdamW+SR to the BF16/BF16 scaling comparisons, and will include the extended results for the remaining settings in the revised manuscript. M+Adam continues to outperform AdamW+SR at every model size and training budget in this setting. For example, at 350M, AdamW+SR obtains perplexities 18.72 / 17.66 / 16.62 / 15.13 at 1x / 2x / 4x / 8x Chinchilla, whereas M+Adam obtains 16.61 / 15.81 / 15.05 / 14.66. The same pattern holds at 60M and 130M. We consider this important because SR is a strong low-precision baseline, yet M+Adam still improves over it.
>
> We provide the full BF16/BF16 scaling table, now including AdamW+SR, in Table 4 of the supplementary note.
>
> **Kahan**
> We also agree that Kahan summation is a meaningful baseline, since it directly targets the accumulation of small low-precision updates. We therefore ran AdamW+Kahan at matched BF16/BF16, 1x Chinchilla settings for 60M / 130M / 350M. The resulting validation perplexities are 29.275 / 22.859 / 18.758, compared to 29.035 / 22.032 / 16.607 for M+Adam under the same settings. Thus, Kahan improves over standard AdamW, but M+Adam remains better across all three scales. For the full comparison table (including AdamW and AdamW+SR perplexities), please refer to Table 6, and for timings to Table 3, in the supplementary link. We note that Kahan incurred a substantial throughput penalty (roughly 2x slower end-to-end on H200 gpus), which we did not have time to investigate further.
>
> SR and Kahan are both meaningful and strong additive baselines, but they act differently from M+Adam. SR changes how additive updates are rounded onto the low-precision grid, while Kahan improves the accumulation of small additive updates through compensated summation. In contrast, M+Adam changes the update parameterization itself, combining additive mantissa refinement with multiplicative exponent adjustment. As mentioned in Future work line 404, we also view combining M+Adam with SR or Kahan as a meaningful follow-up direction.
>
> **(2) Downstream evaluation.**
> We agree that validation perplexity should not be the only metric. At the same time, perplexity remains the standard primary metric for autoregressive pretraining and is widely used for optimizer comparisons in LLM pretraining. To strengthen the evaluation, we also measured downstream performance for the 350M, 1x Chinchilla checkpoints using the EleutherAI lm-evaluation-harness on HellaSwag (Zellers et al., 2019), PIQA (Bisk et al., 2020), WinoGrande (Sakaguchi et al., 2021), and ARC-Challenge (Clark et al., 2018). We report the mean of the taskwise preferred scores.
>
> Across all three precision regimes, M+Adam achieves the best average preferred score: in BF16/BF16, AdamW = 0.3913, AdamW+SR = 0.3929, and M+Adam = 0.3944; in BF16/FP8, AdamW = 0.3855 and M+Adam = 0.3861; and in FP8/FP8, AdamW = 0.3825 and M+Adam = 0.3834. While the per-task gaps are naturally small at this scale, the overall downstream ranking is consistent with the perplexity results. Full per-task results are provided in Table 5 of the supplementary note.
>
> **(3) Memory overhead vs. standard mixed precision.**
> We agree that the current M+Adam formulation does not provide a memory advantage over standard mixed-precision AdamW, and we already state this explicitly in the paper. In particular, maintaining separate optimizer states for the mantissa and exponent can offset the memory saved by removing FP32 master weights. Our contribution in this work is therefore not lower optimizer-state memory, but improved stability for persistent low-precision weight updates without FP32 master weights. Several prior works, including those discussed in the paper, instead target optimizer-state memory directly, for example, by storing moments in lower precision or by using low-rank or sparse optimizer states. We view these directions as complementary rather than competing: M+Adam changes the update rule to better match low-precision weight geometry, whereas those methods reduce the storage cost of the optimizer states. Combining such memory-efficient optimizer-state techniques with M+Adam is a natural next step, but is left for future work. We will clarify this distinction more explicitly in the revision.

---

> > ### Author Rebuttal · Reviewer_oXdN · 2026-04-03
> >
> > Thank you for your response. I still have a few questions:
> >
> > 1). For the downstream evaluation, what is the performance of Kahan?
> >
> > 2). Regarding memory overhead, the proposed approach appears less efficient than Kahan, which only requires a BF16 compensation while keeping other components in low precision. What is the main advantage of the proposed method over Kahan?

---

> > > ### Author Response · Authors · 2026-04-07
> > >
> > > Thank you for the follow-up questions. We agree these are important clarifications and appreciate the opportunity to address them directly.
> > >
> > > **(1) Downstream performance of Kahan.**
> > >
> > > We have now evaluated AdamW+Kahan on the same downstream harness for the 350M, 1x Chinchilla, BF16/BF16 setting. It achieves an average preferred score of 0.3928 (HellaSwag 0.2772, PIQA 0.5292, WinoGrande 0.5113, ARC-Challenge 0.2534). For comparison, under the same setting AdamW = 0.3913, AdamW+SR = 0.3929, and M+ADAM = 0.3944. Thus, Kahan is a meaningful and competitive baseline. It improves over AdamW, but M+Adam performs better overall in this comparison.
> > >
> > > **(2) Comparison to Kahan: memory and main advantage.**
> > >
> > > We agree that Kahan is attractive when the goal is specifically to improve accumulation of small additive updates with additional states. In that sense, M+Adam is not more memory-efficient than Kahan, and we do not want to claim otherwise.
> > >
> > > However, the intended advantage of M+Adam is different. Kahan keeps the standard additive update rule and mainly mitigates one specific failure mode: the loss of small additive updates. In contrast, M+ADAM changes the update parameterization itself, combining additive mantissa refinement with multiplicative exponent adjustment. The goal is therefore broader: improving robustness under persistent low-precision weight storage, not only preserving small updates.
> > > This difference is reflected in pretraining. At 1x Chinchilla, BF16/BF16, M+Adam outperforms AdamW+Kahan at all three scales: 60M: 29.275 vs. 29.035, 130M: 22.859 vs. 22.032, 350M: 18.758 vs. 16.607.
> > >
> > > Regarding systems cost, we did observe higher overhead for Kahan in our current implementation. However, we do not yet fully understand whether this slowdown is intrinsic to Kahan or implementation-specific, so we do not want to over-interpret the absolute timing numbers. Still, in the experiments we ran, Kahan was less effective than M+Adam and did not show a clear efficiency advantage over alternatives such as stochastic rounding in our setup.

---

### Official Review · Reviewer_v862 · 2026-03-12

**Soundness:** 3
**Presentation:** 3
**Significance:** 3
**Originality:** 3
**Overall Recommendation:** 4
**Confidence:** 3

**Summary:**

The paper proposes M+ADAM, which decomposes weights into mantissa and exponent, applies Adam-style additive updates to the mantissa and Madam-style multiplicative updates to the exponent, and argues that this combination better survives quantization under BF16/FP8 training. The paper supports this idea with toy examples illustrating complementary failure modes of additive vs. multiplicative updates, a monotone-descent style analysis for the mantissa–exponent formulation under smoothness assumptions, and LLaMA-style pretraining experiments at 60M, 130M, and 350M parameters under BF16/BF16, BF16/FP8, and simulated FP8/FP8 settings, where M+ADAM consistently improves over AdamW and in some cases approaches FP32-master-weight performance.

**Compliance With Llm Reviewing Policy:**

Affirmed.

**Final Justification:**

My concerns have been adequately addressed in rebuttal, so I maintain my weak accept judgment.

**Key Questions For Authors:**

1. **How large is the gap between the theory and the actual training algorithm?**
   The main descent guarantee is established for sequential, deterministic, real-valued updates, while the practical method uses parallel updates, momentum states, clipping, and stochastic gradients. Could the authors explain more explicitly why these differences should not change the main conclusions of the paper?

2. **Why do the experiments stop at 350M parameters?**
   AdamW is often outperformed by alternative optimizers at smaller scales, yet it remains the default choice overall. I therefore think it is important to test at least one larger order of magnitude, e.g., around 1B parameters. Alternatively, the authors could combine the method with approaches such as GaLore or other optimizer-state compression techniques to enable evaluation at larger scale. The paper itself also mentions billion-scale models and integration with compressed optimizer states as a natural next step.

**Limitations:**

yes

**Strengths And Weaknesses:**

**Soundness:**
The main strength is that the paper is motivated by a concrete numerical issue: additive updates can vanish under coarse low-precision formats, while purely multiplicative updates have complementary problems near zero. The mantissa/exponent decomposition is a reasonable way to address this, and the toy experiments clearly support the intuition. The empirical results are also fairly broad within the chosen regime, covering multiple precisions, model sizes, and compute budgets. That said, the main theoretical guarantee is proved for sequential, deterministic, real-valued updates, whereas the practical algorithm uses parallel updates, momentum states, clipping, and stochastic gradients. I would like a clearer argument for why this gap does not materially affect the paper’s conclusions.

**Presentation:**
The paper is generally clear and easy to follow. The problem setup, motivation, method, and experiments are aligned, and the toy examples are especially helpful for understanding the proposed design. The experimental protocol is also more careful than many optimizer papers, since the authors explicitly describe a coordinate-descent hyperparameter tuning procedure. Still, the relationship between the theory and the final implemented algorithm could be explained more directly in the main paper rather than being left somewhat implicit.

**Significance:**
The topic is important because fully low-precision training without FP32 master weights is practically relevant, and the paper offers a useful optimizer-level perspective on that problem. The results are promising, especially the gains over AdamW across 1–8× Chinchilla budgets and in the challenging FP8 setting. However, the scale of evaluation stops at 350M, so it is still unclear whether the same conclusions hold at the more decision-relevant scale of around 1B parameters or beyond.

**Originality:**
I find the core idea reasonably original. Separating mantissa and exponent and assigning different update rules to each is a neat optimizer design that is more specific than a generic additive/multiplicative hybrid. The paper also does a good job of connecting this design to floating-point structure rather than presenting it as a purely heuristic modification.

---

> ### Author Rebuttal · Authors · 2026-03-31
>
> We thank the reviewer for the careful and balanced assessment. We are glad the reviewer found the motivation, toy analysis, and empirical protocol convincing. We agree that the current submission should more clearly distinguish between the scope of the theory and the scope of the empirical claims.
>
> During rebuttal, in response to reviewer requests, we ran and report new 1B scale-extension experiments (Table 6; GeFu, v862, ojsS), throughput / systems measurements (Tables 1–3; GeFu), stronger baselines with AdamW+SR and AdamW+Kahan (Tables 4 and 6; oXdN, ojsS), and selected downstream evaluations (Table 5; oXdN). These are summarized in the following anonymous supplementary document: https://github.com/xyxyxyxy1/icmlre/blob/main/rebuttal_supplementary.pdf
>
> **(1) Gap between theory and the implemented optimizer.**
> We agree that Theorem 4.1 analyzes a simplified sequential, deterministic, real-valued mantissa-exponent update rather than the full implemented optimizer with parallel updates, Adam-style preconditioning, clipping, and stochastic gradients. We should have made this scope clearer in the main text. At the same time, the paper already points to the relevant bridge: as noted around line 1020, the appendix shows that when the same gradient information is used, the parallel update satisfies the same descent conclusion for sufficiently small step sizes. Thus, the theorem is intended as an idealized analysis of the same underlying mantissa-exponent mechanism, not as a literal convergence theorem for the final stochastic implementation. We nevertheless believe this analysis remains relevant to the practical method because the differences above do not remove the low-precision geometry that motivates M+Adam: additive updates can vanish at large magnitudes, while multiplicative scale adjustments behave differently near zero and across exponent bins.
> Parallel vs. sequential application changes the ordering of the two branches, but not the fact that the optimizer combines local additive refinement with scale-aware multiplicative adjustment.
>
>  The remaining practical ingredients are not covered by the theorem, and we will make that limitation explicit in the revision. More broadly, the main conclusions of the paper are empirical: the full implemented optimizer is evaluated across model sizes, precision regimes, and training budgets, and consistently improves over AdamW in the low-precision settings of interest.
>
> **(2) Why stop at 350M parameters?**
>  We agree that evaluation beyond 350M parameters is important. At the same time, our current study was designed for strict and reliable optimizer comparison: we used a systematic coordinate-descent hyperparameter search and reported full sweeps rather than relying on lightly tuned baselines. We also evaluated across three model sizes (60M/130M/350M), multiple precision regimes, and training horizons from 1x,2x,4x, and 8x Chinchilla. Within this studied regime, M+Adam consistently outperforms AdamW, and the gains persist at longer horizons rather than appearing only in short runs.
>
> To address the reviewer’s scale concern more directly, we also ran *1B-parameter follow-up* experiments during rebuttal using a configuration extrapolated from the 60M-350M tuning study. Under matched BF16/BF16 settings, we currently observe validation perplexity 15.6 for AdamW versus 14.1 for M+Adam, which we view as promising preliminary evidence that the advantage persists beyond 350M. We emphasize, however, that this is a single rebuttal-time 1B configuration rather than a full 1B sweep. Full hyperparameter sweeps at this scale are typically prohibitively expensive, so we present this result as preliminary scale-extension evidence rather than definitive large-scale validation. For the revised manuscript, we are extending these 1B experiments to the remaining setups, including BF16/FP8 and additional baseline comparisons such as AdamW+SR.

---

> > ### Author Rebuttal · Reviewer_v862 · 2026-04-03
> >
> > Thank you for the reply; I will maintain my positive rating.

---

> > > ### Author Response · Authors · 2026-04-04
> > >
> > > Thank you. We are glad that your concerns have been fully addressed and appreciate your careful evaluation of the paper.

---

### Official Review · Reviewer_GeFu · 2026-03-16

**Soundness:** 2
**Presentation:** 2
**Significance:** 3
**Originality:** 3
**Overall Recommendation:** 3
**Confidence:** 4

**Summary:**

This paper proposes a low-precision variant of the Adam optimizer by redesigning the updates for the mantissa and exponent separately under a floating-point representation framework. The authors provide theoretical analysis showing that the objective value can decrease stably under gradient-based optimization. Empirically, experiments on models with up to 350M parameters demonstrate that the proposed method outperforms AdamW.

**Compliance With Llm Reviewing Policy:**

Affirmed.

**Final Justification:**

The algorithm is only for GPU memory saving, but it introduces more computations. However, low precision should introduce some computational benefit. Compared with CPU offload, which is the most GPU memory-saving method, M+ADAM does not seem to be better than this.

**Key Questions For Authors:**

1. Could the authors provide further discussion or analysis of the convergence behavior of the actual proposed algorithm?

2. Could the authors report the computational overhead introduced by the proposed update scheme?

3. Can the authors provide additional evidence of the method’s effectiveness at larger scales?

**Limitations:**

Yes

**Strengths And Weaknesses:**

**Strengths**

1. The paper identifies an important challenge in low-precision optimization, namely that quantities at different scales may require different update strategies. Motivated by this observation, the authors design separate update rules for the exponent and mantissa, which is a novel and well-motivated idea.

2. The empirical results are promising: on models with up to 350M parameters, the proposed method (M-Adam) consistently outperforms AdamW, suggesting its practical effectiveness in the targeted low-precision setting.


**Weaknesses**

1. The theoretical analysis is not fully aligned with the actual algorithm. In the proof, the exponent is updated first, and the updated exponent is then used to update the mantissa. In contrast, the algorithm appears to update the exponent and mantissa simultaneously. Moreover, the theory is established for standard gradient descent, whereas the practical setting involves stochastic gradients and an Adam-style optimizer. As a result, it remains unclear to what extent the theoretical results are representative of the convergence behavior of the proposed method in practice.

2. Updating the mantissa and exponent separately may require a new computational paradigm. If implemented directly on existing hardware primitives, this design could introduce additional overhead and slow down optimizer updates. However, the paper does not report the systems cost of the proposed optimizer or quantify how much training throughput is affected in practice.

3. The experimental scale is still limited. Although the results on models up to 350M parameters are encouraging, this scale may not be sufficient to validate the method for large-scale training. In practice, many optimization methods perform well at the 350M scale but encounter difficulties when scaled to much larger models. Therefore, the current experiments do not yet establish the robustness of the method in truly large-model settings.

---

> ### Author Rebuttal · Authors · 2026-03-31
>
> We thank the reviewer for the careful reading and constructive feedback.
> During rebuttal, in response to reviewer requests, we ran and reported new 1B scale-extension experiments (Table 6; GeFu, v862, ojsS), throughput / systems measurements (Tables 1-3; GeFu), stronger baselines with AdamW+SR and AdamW+Kahan (Tables 4 and 6; oXdN, ojsS), and selected downstream evaluations (Table 5; oXdN). These are summarized in the following anonymous supplementary document: https://github.com/xyxyxyxy1/icmlre/blob/main/rebuttal_supplementary.pdf
>
> **(1) Theory / algorithm alignment.**
> We agree that Theorem 4.1 does not analyze the full implemented optimizer. The theorem studies a simplified sequential, deterministic, real-valued mantissa-exponent update, whereas the practical method uses parallel mantissa/exponent updates together with Adam-style preconditioning, clipping, and stochastic gradients. We should have made this scope clearer in the main paper. At the same time, the paper already points to the relevant bridge: as noted around line 1020, the appendix shows that when the same gradient information is used, the parallel update satisfies the same descent conclusion for sufficiently small step sizes. Thus, the sequential analysis is intended as an idealized analysis of the same underlying mantissa-exponent mechanism, not as a separate theoretical object unrelated to the implementation. The additional practical ingredients, especially Adam-style moments and stochastic gradients, are not covered by the theorem, and we will state this limitation explicitly in the revision. Our intent is therefore not to present Theorem 4.1 as a literal convergence theorem for the full implementation but as supporting analysis for the core update structure, while the paper’s main claims about the practical optimizer are empirical.
>
> **(2) Systems cost / throughput.**
> We agree that the paper should have reported systems costs explicitly. To address this, we extracted e2e update times from the existing W&B logs for the submitted runs; the full measurements are provided in Tables 1–3 of the supplementary note. M+ADAM does introduce measurable overhead, but the results do not suggest that it requires a qualitatively different training pipeline. At 350M in FP8/FP8, AdamW takes 0.359 s/update and M+ADAM 0.384 s/update (+6.98%). In BF16/BF16 at 350M, AdamW takes 0.212 s/update, M+ADAM 0.285 (+34.6%), and AdamW+SR 0.304 (+43.6%). At 1B, the overhead is smaller: 3.893 vs. 4.232 s/update in BF16/FP8 (+8.72%) and 3.726 vs. 3.957 in BF16/BF16 (+6.21%).
> We note that the larger overhead at 350M BF16/BF16 appears to be at least partly system-dependent. In our current prototype, optimizer-side memory traffic and kernel overhead are less amortized at 350M than at 1B, where the relative overhead is substantially smaller. BF16/FP8 can also be slower per step than BF16/BF16 in the current stack because FP8 GEMMs require additional quantize/dequantize operations. In addition, the FP8/FP8 timings are conservative, since persistent FP8 weights are currently simulated in software rather than supported natively in PyTorch.
> These timings come from our current research prototype built with standard PyTorch operators rather than a fused CUDA/Triton implementation, so they should be interpreted as measured prototype costs rather than an optimized lower bound. We will report the full timing tables in the revision and in Tables 1–3 of the supplementary note.
>
> **(3) Experimental scale.**
> We agree that evaluation beyond 350M parameters is important, and this scale does not yet match frontier LLM training. At the same time, our current study was designed for strict and reliable optimizer comparison: we used a systematic coordinate-descent hparam search and reported full sweeps rather than relying on lightly tuned baselines. We also evaluated across three model sizes (60M/130M/350M), multiple precision regimes, and training horizons from 1x to 8x Chinchilla. Within this studied regime, M+ADAM consistently outperforms AdamW, and the gains persist at longer horizons rather than appearing only in short runs.
> To address the reviewer’s scale concern more directly, we also ran 1B-parameter follow-up experiments during rebuttal using a configuration extrapolated from the 60M-350M tuning study. Under matched BF16/BF16 settings, we currently observe validation perplexity 15.6 for AdamW versus 14.1 for M+ADAM, which we view as promising preliminary evidence that the advantage persists beyond 350M. We emphasize, however, that this is a single rebuttal-time 1B configuration rather than a full 1B sweep, as full hyperparameter sweeps at this scale are typically prohibitively expensive. For the revised manuscript, we are extending these 1B experiments to the remaining setups, including BF16/FP8 and additional baseline comparisons such as AdamW+SR.
>
> [1] benchmarking optimizers - https://arxiv.org/abs/2509.01440
>
> [2] fantastik pretraining optimizers - https://arxiv.org/abs/2509.02046

---

> > ### Author Rebuttal · Reviewer_GeFu · 2026-04-01
> >
> > Thanks for the detailed response. If M+Adam can not save time but can save memory, it is worth comparing the algorithm with CPU-offload type of algorithms, which move the entire optimizer into the CPU while introducing some overhead. At least M+Adam should be better than CPU offload.

---

> > > ### Author Response · Authors · 2026-04-03
> > >
> > > Thank you for the follow-up and for engaging with our rebuttal.
> > >
> > > We would first like to clarify that our paper does not claim that M+ADAM is more memory-efficient than standard mixed-precision training. We explicitly note in the paper that the current implementation can lose the memory savings from removing FP32 master weights because of the additional optimizer states. Our main claim is instead about improved stability and effectiveness under persistent low-precision weight storage, rather than immediate speed or memory gains.
> > >
> > > For this reason, we view standard AdamW as the primary systems baseline in our setting. CPU offloading is a related but somewhat orthogonal trade-off: it can reduce GPU memory usage by moving optimizer state and computation to CPU, but does so at the cost of additional transfer and execution overhead.
> > >
> > > As a concrete reference point, we additionally tested AdamW with DeepSpeed’s built-in ZeRO-Offload path (https://www.deepspeed.ai/tutorials/zero-offload/). In our 350M setup, this CPU-offload baseline is 94.02% slower than standard GPU-resident AdamW.
> > >
> > > We view CPU offloading as a complementary memory-saving systems technique, rather than the main baseline for our contribution. Our goal in this paper is to show that, when updates are decomposed into additive and multiplicative components, low-precision training can closely match training with FP32 master weights. Improving memory and throughput further remains important future work.
> > >
> > > We hope this clarification helps address the remaining concern.

---

### Decision · Program_Chairs · 2026-04-30

**Decision:**

Accept (regular)

**Comment:**

The paper proposes a variant of Adam tailored towards low-precision training of large neural networks by separately updating exponent and mantissa of the weights. Overall, reviewers found the paper to be well-written with encouraging numerical experiments, but raised several concerns:
1. Lack of convergence analysis and the theoretical results not matching the actual implemented algorithm
2. Small scale of the experiments.
3. Several reviewers were concerned about memory overhead compared to regular Adam and another baseline method based on Kahan summation.

Since the paper is mostly empirical, a detailed theoretical analysis is out-of-scope.  The rebuttal provides results for up to 1B models, which clears concerns about small-scale of experiments.  I found the memory overhead concerns to be well-addressed in the rebuttal:  The main point of the paper is to achieve stable training in low-precision, and no claims about memory savings are made in the paper.

I found reviewer's concerns to be addressed, and recommend to accept the paper in case there is remaining space at the conference.